# The neuronal architecture of the mushroom body provides a logic for associative learning

Yoshinori Aso[1]*, Daisuke Hattori[2], Yang Yu[1], Rebecca M Johnston[1], Nirmala A Iyer[1], Teri-TB Ngo[1], Heather Dionne[1], LF Abbott[3,4], Richard Axel[2,3,7], Hiromu Tanimoto[5,6], Gerald M Rubin[1]*

[1]Janelia Research Campus, Howard Hughes Medical Institute, Ashburn, United States; [2]Howard Hughes Medical Institute, Columbia University, New York, United States; [3]Department of Neuroscience, College of Physicians and Surgeons, Columbia University, New York, United States; [4]Department of Physiology and Cellular Biophysics, College of Physicians and Surgeons, Columbia University, New York, United States; [5]Tohuku University Graduate School of Life Sciences, Sendai, Japan; [6]Max-Planck Institute of Neurobiology, Martinsried, Germany; [7]Department of Biochemistry and Molecular Biophysics, College of Physicians and Surgeons, Columbia University, New York, United States

**Abstract** We identified the neurons comprising the *Drosophila* mushroom body (MB), an associative center in invertebrate brains, and provide a comprehensive map describing their potential connections. Each of the 21 MB output neuron (MBON) types elaborates segregated dendritic arbors along the parallel axons of ~2000 Kenyon cells, forming 15 compartments that collectively tile the MB lobes. MBON axons project to five discrete neuropils outside of the MB and three MBON types form a feedforward network in the lobes. Each of the 20 dopaminergic neuron (DAN) types projects axons to one, or at most two, of the MBON compartments. Convergence of DAN axons on compartmentalized Kenyon cell–MBON synapses creates a highly ordered unit that can support learning to impose valence on sensory representations. The elucidation of the complement of neurons of the MB provides a comprehensive anatomical substrate from which one can infer a functional logic of associative olfactory learning and memory.

*For correspondence: asoy@ janelia.hhmi.org (YA); rubing@ janelia.hhmi.org (GMR)

**Competing interests:** The authors declare that no competing interests exist.

**Reviewing editor**: Leslie C Griffith, Brandeis University, United States

## Introduction

Neural representations of the sensory world give rise to appropriate innate or learned behavioral responses. Innate behaviors are observed in naïve animals without prior learning or experience, suggesting that they are mediated by genetically determined neural circuits. Responses to most sensory stimuli, however, are not innate but experience-dependent, allowing an organism to respond appropriately in a variable and uncertain world. Thus, most sensory cues acquire behavioral relevance through learning. In *Drosophila melanogaster*, a number of different forms of learning have been observed in response to sensory stimuli (*Siegel and Hall, 1979*; *Liu et al., 1999*, *2006*; *Masek and Scott, 2010*; *Schnaitmann et al., 2010*; *Ofstad et al., 2011*; *Vogt et al., 2014*). In associative olfactory learning, exposure to an odor (conditioned stimulus, CS) in association with an unconditioned stimulus (US) results in appetitive or aversive memory (*Quinn et al., 1974*; *Tempel et al., 1983*; *Tully and Quinn, 1985*). Olfactory memory formation and retrieval in insects require the mushroom body (MB) (*Heisenberg et al., 1985*; *de Belle and Heisenberg, 1994*,

**eLife digest** One of the key goals of neuroscience is to understand how specific circuits of brain cells enable animals to respond optimally to the constantly changing world around them. Such processes are more easily studied in simpler brains, and the fruit fly—with its small size, short life cycle, and well-developed genetic toolkit—is widely used to study the genes and circuits that underlie learning and behavior.

Fruit flies can learn to approach odors that have previously been paired with food, and also to avoid any odors that have been paired with an electric shock, and a part of the brain called the mushroom body has a central role in this process. When odorant molecules bind to receptors on the fly's antennae, they activate neurons in the antennal lobe of the brain, which in turn activate cells called Kenyon cells within the mushroom body. The Kenyon cells then activate output neurons that convey signals to other parts of the brain.

It is known that relatively few Kenyon cells are activated by any given odor. Moreover, it seems that a given odor activates different sets of Kenyon cells in different flies. Because the association between an odor and the Kenyon cells it activates is unique to each fly, each fly needs to learn through its own experiences what a particular pattern of Kenyon cell activation means.

Aso et al. have now applied sophisticated molecular genetic and anatomical techniques to thousands of different transgenic flies to identify the neurons of the mushroom body. The resulting map reveals that the mushroom body contains roughly 2200 neurons, including seven types of Kenyon cells and 21 types of output cells, as well as 20 types of neurons that use the neurotransmitter dopamine. Moreover, this map provides insights into the circuits that support odor-based learning. It reveals, for example, that the mushroom body can be divided into 15 anatomical compartments that are each defined by the presence of a specific set of output and dopaminergic neuron cell types. Since the dopaminergic neurons help to shape a fly's response to odors on the basis of previous experience, this organization suggests that these compartments may be semi-autonomous information processing units.

In contrast to the rest of the insect brain, the mushroom body has a flexible organization that is similar to that of the mammalian brain. Elucidating the circuits that support associative learning in fruit flies should therefore make it easier to identify the equivalent mechanisms in vertebrate animals.

*Dubnau et al., 2001*; *McGuire et al., 2001*), an associative center in the protocerebrum (*Figure 1* and *Video 1*).

Olfactory perception in the fly is initiated by the binding of an odorant to an ensemble of olfactory sensory neurons in the antennae, resulting in the activation of a distinct and topographically fixed combination of glomeruli in the antennal lobe (*Figure 1A,B*; reviewed in *Vosshall and Stocker (2007)*; *Masse et al. (2009)*). Most antennal lobe projection neurons (PNs) extend dendrites to a single glomerulus and project axons that bifurcate to innervate two brain regions, the lateral horn and the MB (*Stocker et al., 1990*; *Wong et al., 2002*; *Jefferis et al., 2007*). The invariant circuitry of the lateral horn is thought to mediate innate behaviors, whereas the MB translates olfactory sensory information into learned behavioral responses (*Heisenberg et al., 1985*). The PN axons synapse onto the dendrites of the Kenyon cells (KCs) in the MB calyx; the parallel axons of the KCs form the MB lobes. Odors activate sparse subpopulations of KCs distributed across the MB without spatial preference (*Turner et al., 2008*; *Honegger et al., 2011*; *Campbell et al., 2013*). Anatomical and physiological studies reveal that each KC receives on average 6.4 inputs from a random combination of glomeruli; that is, knowledge of a single input to a KC provides no information about the identity of the additional inputs, and connections differ in different flies (*Murthy et al., 2008*; *Caron et al., 2013*; *Gruntman and Turner, 2013*). Thus, the calyx of the MB discards the highly ordered structure of the antennal lobe. A restoration of order must therefore be imposed downstream to link the KC representation to an appropriate behavioral output.

Three classes of KCs extend parallel fibers that form the γ, α′/β′, and α/β lobes of the MB, where they form synapses with a relatively small number of MB output neurons (MBONs; *Figure 1C*) (*Crittenden et al., 1998*; *Ito et al., 1998*; *Strausfeld et al., 2003*; *Lin et al., 2007*; *Tanaka et al., 2008*;

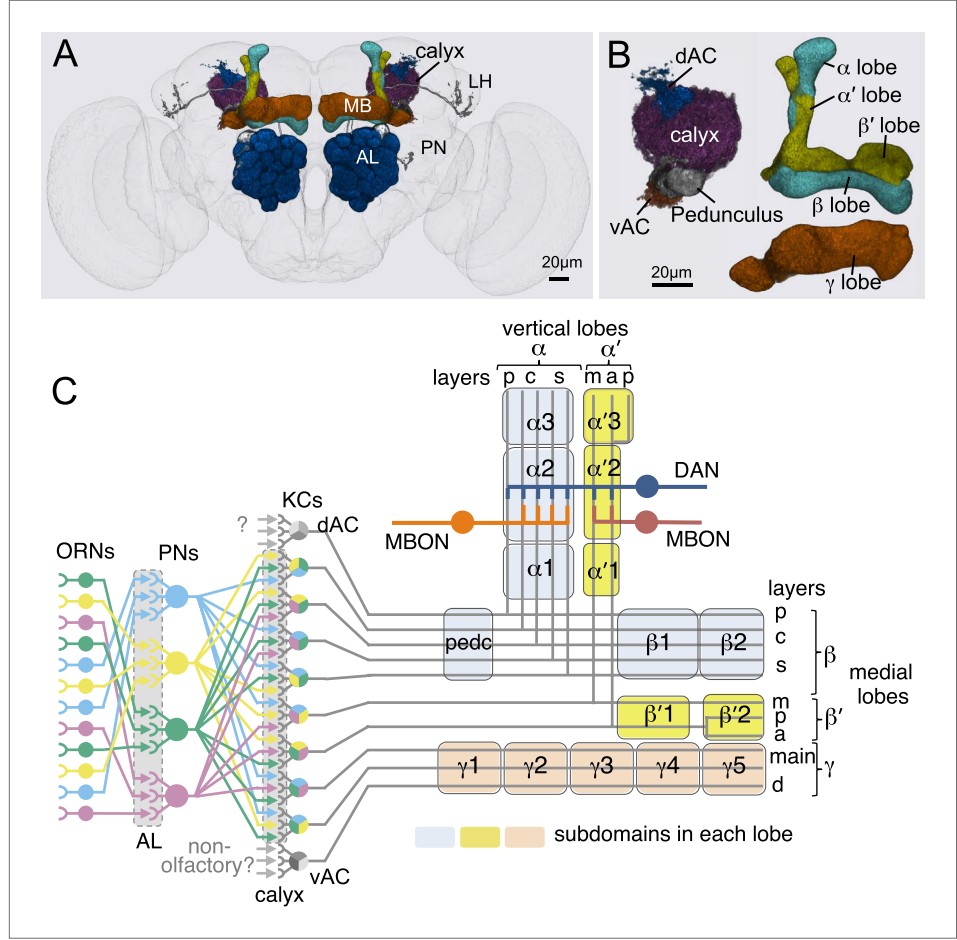

**Figure 1**. Anatomy of olfactory pathways in the adult fly brain. (**A**) An image of the adult female brain showing the antennal lobes (AL) and subregions of the mushroom bodies (MB; see panel **B** for more detail). The image was generated using a 3D image rendering software (FluoRender) (**Wan et al., 2009**; **Wan et al., 2012**). The 51 glomeruli of the AL extend projection neurons (PN) to the calyx of the MB and the lateral horn (LH). There are a total of ~200 PN; 6 from the DL3 glomerulus are shown. See **Video 1** for an introduction to olfactory circuit. (**B**) Subregions within the MB. The γ lobe, calyx, and pedunculus (ped) are displayed separately from other lobes; their normal positions are as shown in panel **A** and **Video 1**. Color-coding is as in panel **A**. See below for a detailed description of dorsal accessory calyx (dAC) and ventral accessory calyx (vAC). (**C**) A schematic representation of the key cellular components and information flow during processing of olfactory inputs to the MB (see text for references and more details). Olfactory receptor neurons expressing the same odorant receptor converge onto a single glomerulus in the AL. A small number (generally 3–4) of PNs from each of the 51 AL glomeruli innervate the MB calyx where they synapse on the dendrites of the ~2000 Kenyon cells (KCs) in a globular structure, the calyx. Each KC exhibits, on average, 6.4 dendritic 'claws' (**Butcher et al., 2012**), and each claw is innervated by a single PN. There is little order in connection patterns of PNs to KCs. The axons of the KCs project in parallel anteriorly through the pedunculus to the lobes, where KCs terminate onto the dendrites of MB output neurons (MBONs). KCs can be categorized into three major classes α/β, α'/β', and γ based on their projection patterns in the lobes (**Crittenden et al., 1998**). The β, β', and γ lobes constitute the medial lobes (also known as horizontal lobes), while the α and α' lobes constitute the vertical lobes. These lobes are separately wrapped by ensheathing glia (**Awasaki et al., 2008**). The α/β and α'/β' neurons bifurcate at the anterior end of the pedunculus and project to both the medial and vertical lobes (**Lee et al., 1999**). The γ neurons project only to the medial lobe. Dendrites of MBONs and terminals of modulatory dopaminergic neurons (DANs) intersect the longitudinal axis of the KC axon bundle, forming 15 subdomains, five each in the α/β, α'/β', and γ lobes (numbered α1, α2, and α3 for subdomains in the α lobe from proximal to distal) (**Tanaka et al., 2008**). Additionally, one MBON and one DAN innervate the core of the distal pedunculus intersecting the α/β KCs (pedc, see below). There are seven types of KCs; five of the seven types have their dendrites in the main calyx, while those of the γd cells form the vAC (**Aso et al., 2009**; **Butcher et al., 2012**) and those of the α/βp cells the dAC (**Tanaka et al., 2008**). The accessory calyces are thought to

*Figure 1. Continued on next page*

*Figure 1. Continued*

receive non-olfactory input since they do not receive input from the PNs from the AL (*Tanaka et al., 2008*). Different KCs occupy distinct layers in the lobes as indicated (p: posterior; c: core; s: surface; m: medial; a: anterior; and d: dorsal). Some MB extrinsic neurons extend processes to only a specific layer within a subdomain, defining elemental subdivisions in the lobes, or 'synaptic units' as proposed by *Tanaka et al. (2008)*.

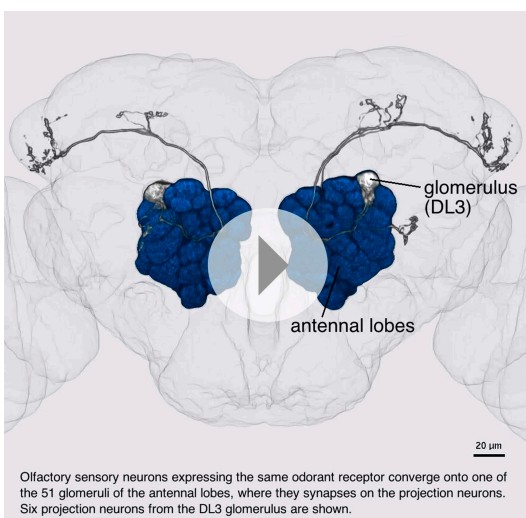

Olfactory sensory neurons expressing the same odorant receptor converge onto one of the 51 glomeruli of the antennal lobes, where they synapses on the projection neurons. Six projection neurons from the DL3 glomerulus are shown.

**Video 1**. Introduction to MB anatomy and the olfactory circuit.

*Busch et al., 2009*). The MBONs have dendrites in the MB lobes and project axons to neuropils outside of the MB. Modulatory input neurons, including dopaminergic neurons (DANs) and octopaminergic neurons (*Nassel and Elekes, 1992*; *Tanaka et al., 2008*; *Busch et al., 2009*; *Mao and Davis, 2009*), also innervate the MB lobes. The MBONs and DANs send their processes to stereotyped locations, defining spatially restricted 'subdomains' in each lobe (*Ito et al., 1998*; *Tanaka et al., 2008*; *Mao and Davis, 2009*; *Pech et al., 2013*). However, these studies did not establish the precise anatomical relationships between the subdomains; knowledge of these relationships will be required to understand the structure and logic of MB circuits.

The DANs are the most prevalent modulatory neurons in the MB and dopamine is thought to act locally to modify KC–MBON synapses (*Aso et al., 2010*; *Waddell, 2013*). In accord with this model, DAN activity is required during learning (*Schwaerzel et al., 2003*; *Aso et al., 2010, 2012*; *Burke et al., 2012*; *Liu et al., 2012*) and exogenous activation of DAN subpopulations can serve as an US in associative learning paradigms (*Schroll et al., 2006*; *Claridge-Chang et al., 2009*; *Aso et al., 2010, 2012*; *Burke et al., 2012*; *Liu et al., 2012*). In addition, D1-like dopamine receptors in the KCs are necessary to form olfactory memories (*Kim et al., 2007*).

Different populations of DANs are activated by USs of different valence; see Figure 1A of the accompanying paper (*Aso et al., 2014*) for summary (*Riemensperger et al., 2005*; *Mao and Davis, 2009*; *Liu et al., 2012*; *Das et al., 2014*). Genetic manipulation has also implicated specific subsets of MBONs in the mediation of learned appetitive and aversive behaviors (*Sejourne et al., 2011*; *Pai et al., 2013*; *Placais et al., 2013*; *Aso et al., 2014*). These experiments implicate the DANs as the source of the learning cue and the MBONs as the mediators of behavioral output. The elucidation of the connections between KCs, DANs, and MBONs should provide insight into a problem shared by invertebrate and vertebrate nervous systems: how is meaning imposed on an unstructured ensemble of neurons and how is imposed valence translated into an appropriate behavioral response?

In this study, we developed new genetic reagents and used them to identify the cell types and projections of the neurons comprising the MB lobes. These data provide insight into the potential connections in the MB and suggest how the MB may mediate learned behaviors. We found that the MB lobes are composed of ~2200 neurons that include 7 KC, 21 MBON, and 20 DAN cell types. The MBONs of a given type exhibit spatially stereotyped dendritic arbors in the MB lobes that form 15 compartments that collectively tile the lobes. Each DAN cell type projects axons to one or at most two of the compartments defined by the MBONs. The alignment of DAN axons with compartmentalized KC–MBON synapses creates an isolated unit for learning that can transform the disordered KC representation into ordered MBON output. The MBON axons project to five discrete neuropils outside of the MB, providing loci for the convergence of all the information necessary for learned associative responses. The elucidation of the full complement of MB neurons and the details of their projections provide an anatomical substrate from which we can infer a functional logic of olfactory learning and memory.

## Results

We designed a genetic approach to examine the architecture of the MB circuit and identified most, if not all, of the neurons innervating the MB lobes. We screened adult brains from 7000 GAL4 lines driven by known enhancers (*Pfeiffer et al., 2008*; *Jenett et al., 2012*) to identify lines containing neurons that innervate the MB lobes. These GAL4 drivers typically label many other neurons, making it difficult to disambiguate the projection patterns of the labeled MB neurons. We therefore identified lines with overlapping expression patterns for the MB neurons and used the split-GAL4 strategy (*Luan et al., 2006*; *Pfeiffer et al., 2010*) to identify lines with more restricted expression in the MB (*Figure 2*). After screening 2500 such intersections, we obtained more than 400 split-GAL4 combinations that had strong expression in the MB neurons. Most split-GAL4 lines that drive expression in MBONs or DANs contain a small number of neurons that share virtually identical morphologies and exhibit bilateral symmetry, and this profile is maintained across different individuals (*Figure 2—figure supplements 3–6* and data not shown). The split-GAL4 lines that label the KC contain far greater cell numbers (75–600) (*Figure 2—figure supplement 2*). These split-GAL4 lines allowed us to classify the MB neurons into cell types (*Figure 3* and *Videos 2–4*). We operationally define a cell type as a single neuron (per hemisphere) or a group of neurons that is not further subdivided in any of the 7000 GAL4 lines. Moreover, neurons within most cell types exhibit indistinguishable morphology. Importantly, we identified on average 12 GAL4 drivers that label the same cell type within the set of 7000 lines, indicating that the screen was near saturation in our large GAL4 collection.

We employed an independent approach, photoactivatable GFP (PA-GFP) tracing (*Patterson and Lippincott-Schwartz, 2002*; *Datta et al., 2008*; *Ruta et al., 2010*), to verify the results of the split-GAL4 experiments and determine whether the neuron types identified in our screen represent the full complement of MBONs and DANs. Photoactivation of the MB labels all neurons that express PA-GFP and project to the MB (*Figure 4A*; for limitations, see 'Materials and methods'). Photoactivation of the MB lobes in flies expressing PA-GFP pan-neuronally (except in the KCs) resulted in labeling of eight individual neurons and five clusters of neuronal cell bodies (*Figure 4*, see 'Materials and methods'). The number and position of these PA-GFP labeled neurons matched well with the cells identified in the split-GAL4 lines.

We performed a more refined analysis by photoactivating individual subdomains of the MB lobes (*Figure 5*). By labeling processes of specific MBONs or DANs, we could decorate individual subdomains of the MB lobes (*Figure 1C* and see below) (*Tanaka et al., 2008*), allowing focal photoactivation and subsequent identification of the full complements of neurons innervating each lobe subdomain. Photoactivation of individual subdomains confirmed the results obtained from the genetic approach (*Figure 5*), but the photoactivation experiments also revealed two MBONs not identified in the split-GAL4 lines. These MBONs were subsequently identified in the VT-GAL4 collection, allowing us to characterize their projections (*Figure 5—figure supplement 1*).

The split-GAL4 approach identified 20 DAN types of the PPL1 and PAM clusters that innervate the MB. We identified about 30% less DANs in the PAM cluster in our collection of split-GAL4 lines compared to the number estimated by PA-GFP and anti-dopamine immunoreactivity (*Figure 5*) (*Liu et al., 2012*). However, these additional DANs exhibit innervation patterns similar to those of the split-GAL4 lines (see below), and therefore we assume that they either represent closely related cell types or that some of our split-GAL4 drivers are stochastic in their expression and fail to label all members of a cell type. Taken together, these data indicate that each cell type defined by our criteria likely represents an irreducible group of equivalent cells, and that the split-GAL4 screen and PA-GFP tracing identified perhaps all of the neurons in the MB lobes.

These complementary analyses allowed us to make a comprehensive list of cell types comprising the MB lobes (*Table 1*). We selected 92 split-GAL4 lines representing the best examples for single cell types as well as combinations of related cell types; these split-GAL4 lines will facilitate further anatomical and functional characterization of the MB cell types (see *Figure 2—figure supplements 1–6*, *Supplementary file 1*, www.janelia.org/split-gal4, and 'Materials and methods'). In this study, we focus on the three major classes of neurons that provide the input and output of the MB lobes: 7 types of KCs, 21 types of MBONs, and 20 types of DANs (*Figure 3* and *Videos 2–4*).

## The Kenyon cells

Each MB contains ~2000 KCs that are sequentially generated from four neuroblasts (*Ito et al., 1997*; *Lee et al., 1999*; *Zhu et al., 2003*; *Lin et al., 2007*). The dendrites of the KCs form the MB calyx and

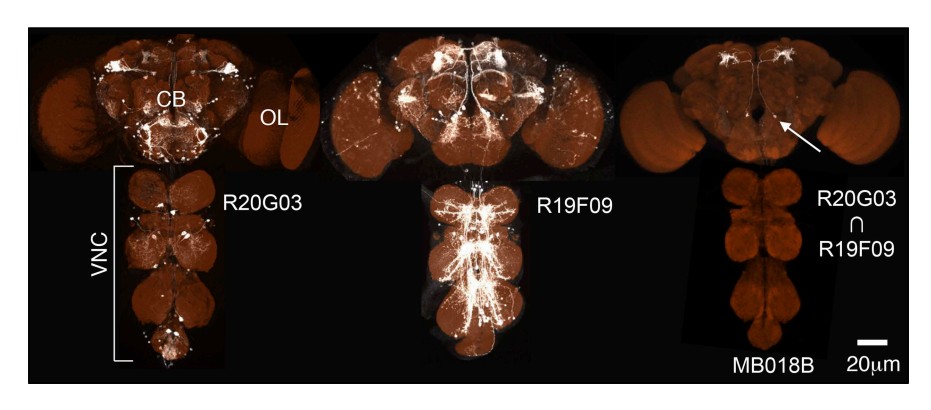

**Figure 2**. Generation of split-GAL4 drivers for the MB neurons. An example of the use of the split-GAL4 approach to generate a driver line specific for MBON-α'2 (see *Table 1* for the naming convention of MBONs and DANs). *R20G03-GAL4* in *attP2* (left) and *R19F09-GAL4* in *attP2* (center) both show expression in MBON-α'2 when crossed to *pJFRC2-10XUAS-IVS-mCD8::GFP* in *attP2* and in many other neurons that differ between the two GAL4 lines. The optic lobes (OL), central brain (CB), and ventral nerve cord (VNC) are indicated. The enhancer fragments from these lines were used to generate the fly line MB018B carrying both *R20G03-p65ADZp* in *attP40* and *R19F09-ZpGAL4DBD* in *attP2* (right). The p65ADZp and ZpGAL4DBD proteins are themselves inactive; the reconstitution of an active GAL4 transcription factor requires heterodimerization that occurs only in cells expressing both proteins (*Luan et al., 2006*; *Pfeiffer et al., 2010*). This approach, therefore, labels cells in which both enhancers are active. The arrow indicates the cell body of one MBON-α'2 cell visualized using *pJFRC225-5xUAS-IVS-myr::smGFP-FLAG* reporter in *VK00005* (white). Neuropils were visualized with nc82 antibody (orange). Genotypes of 92 split-GAL4 lines and the cell types they label are listed in *Supplementary file 1* and raw confocal images are available online (http://www.janelia.org/split-gal4). The expression pattern observed using a split-GAL4 line depends to some extent on the UAS reporter construct used, as illustrated in *Figure 2—figure supplement 1*. Expression patterns of split-GAL4 lines for KCs (*Figure 2—figure supplement 2*), PPL1-cluster DANs (*Figure 2—figure supplement 3*), PAM cluster DANs (*Figure 2—figure supplement 4*), and MBONs (*Figure 2—figure supplement 5*) are shown. We also generated split-GAL4 lines for a variety of other modulatory cell types that project to the MB including serotonergic, GABAergic, octopaminergic, and peptidergic neurons (*Figure 2—figure supplement 6*). We chose lines with minimal off-target expression in neuronal and non-neuronal cells (*Figure 2—figure supplement 7*) to facilitate the use of these lines in future functional analyses to manipulate the activity of individual cell types.

The following figure supplements are available for figure 2:

**Figure supplement 1**. Expression pattern of split-GAL4 drivers with various reporters.

**Figure supplement 2**. Expression patterns of split-GAL4 drivers for KCs.

**Figure supplement 3**. Expression patterns of split-GAL4 drivers for PPL1 cluster DANs.

**Figure supplement 4**. Expression patterns of split-GAL4 drivers for PAM cluster DANs.

**Figure supplement 5**. Expression patterns of split-GAL4 drivers for MBONs.

**Figure supplement 6**. Expression patterns of split-GAL4 drivers for other modulatory input cells.

**Figure supplement 7**. Examples of off-targeted non-neuronal expression.

their parallel axons form the three MB lobes (*Figure 1*) (*Crittenden et al., 1998*). The main calyx primarily receives olfactory input from the antennal lobe, whereas the smaller ventral and dorsal accessory calyces are thought to receive non-olfactory input (see *Figure 1C*) (*Tanaka et al., 2008*; *Butcher et al., 2012*). The KCs have been divided into three classes, γ, α'/β', and α/β, with each class projecting

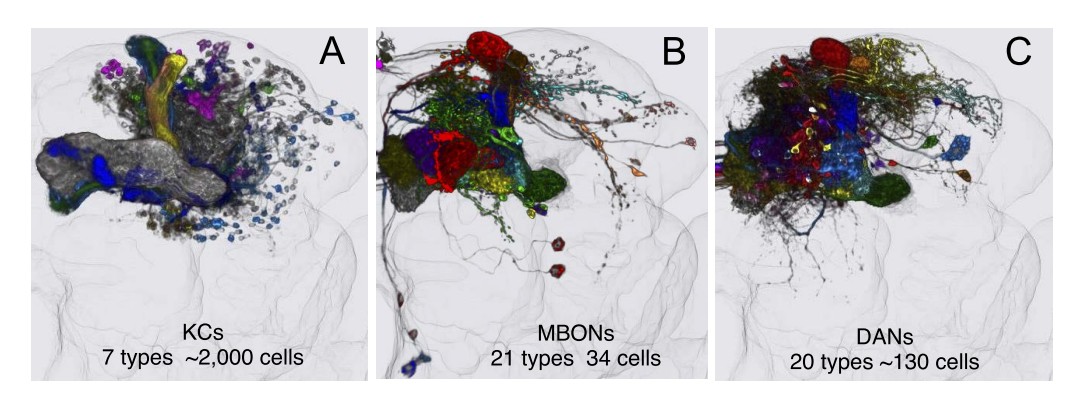

**Figure 3**. MB cell types. Registered images of KCs (**A**; *Video 2*), MBONs (**B**; *Video 3*), and DANs (**C**; *Video 4*). After alignment to the standard brain and segmentation, different colors were assigned to each cell type, while the outlines of the neuropils of the brain are shown in light gray. These images illustrate the overall extent and position of these cell types in the brain; the morphology of individual cell types can be seen in *Videos 2–4*. For MBONs, the two neurons found upon PA-GFP tracing experiments (see below) are not included, thus the image represents 19 different MBON cell types. These are the three major intrinsic and extrinsic neuron types innervating the MB lobes. Additionally, there are two intrinsic neurons (MB-APL and MB-DPM) and 10 extrinsic neuron types that innervate the MB; see *Table 1* for references. The extrinsic neurons with processes in the lobes include two types of octopaminergic neurons (OA-VPM3 and OA-VPM4) and one type of peptidergic neuron (SIFamide). The extrinsic neurons with processes in the calyx include two types of octopaminergic neurons (OA-VPM5 and OA-VUMa2), one type of GABAergic neuron (MB-C1), one serotonergic neuron (MB-CSD), two types of DANs (PPL2ab-DANs), and one neuron with dendritic arbors in the calyx and the proximal pedunculus as well as in the LH (MB-CP1). See *Figure 3—figure supplement 1* for images of some of these cell types.

The following figure supplement is available for figure 3:

**Figure supplement 1**. MBON-calyx and other modulatory neurons.

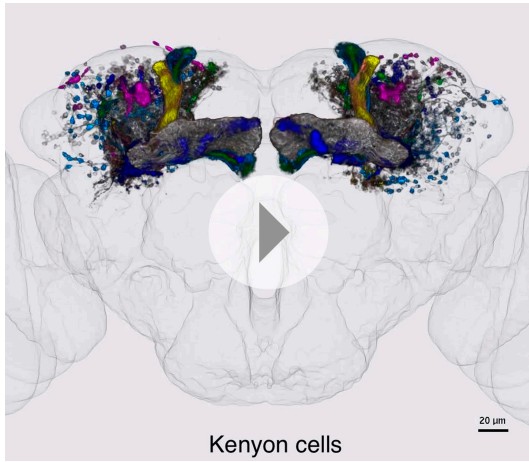

Kenyon cells

**Video 2**. KCs.

axons to the eponymous lobe (*Crittenden et al., 1998*; *Lee et al., 1999*). The split-GAL4 screen and the analysis of the axonal projection patterns of single cells revealed that these three classes of KCs divide into seven cell types (*Figure 3A* and *Video 2*). Each of the four neuroblasts contributes to each of the seven cell types and the dendrites of the KCs generated from the different neuroblasts remain segregated in the main calyx (*Lin et al., 2007*). The parallel axon fibers of each of the seven types of KCs occupy specific layers within the γ, α′/β′, and α/β lobes (*Figures 6 and 7*). Two KC types divide the γ lobe into the main and dorsal (d) layers, two types divide the α′/β′ lobe into the middle (m) and anterior–posterior (ap) layers, and three KC types divide the α/β lobe into the posterior (p), core (c), and surface (s) layers (*Figures 6 and 7*, also see *Figure 1C*). Examination of single cell morphologies suggests that each KC may form *en passant* synapses with target MBONs along the length of its axon, providing each MBON with access to a large number of KC inputs. Five of the seven types of KCs elaborate their dendrites in the main calyx, whereas two types of KCs (γd and α/βp) have dendrites exclusively in the ventral and dorsal accessory calyces, respectively (*Figures 6A and 7A*) (*Lin et al., 2007*; *Tanaka et al., 2008*; *Butcher et al., 2012*).

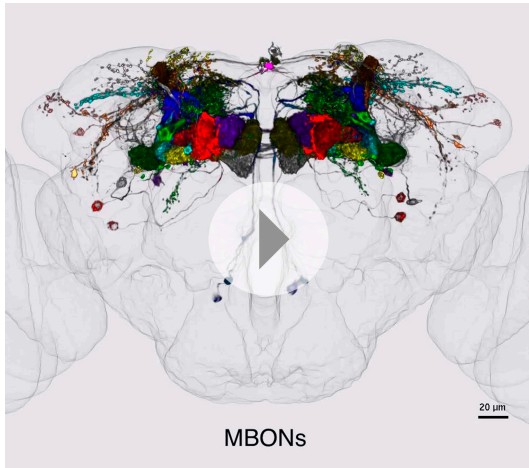

**Video 3**. MBONs.

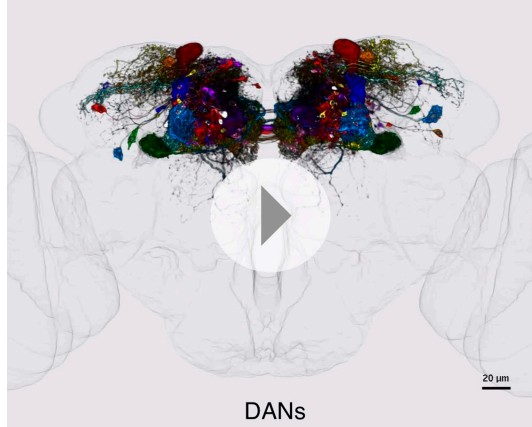

**Video 4**. PAM and PPL1 cluster DANs.

The five KC types (γmain, α′/β′ap, α′/β′m, α/βc, and α/βs) that receive olfactory information are each represented by hundreds of neurons per hemisphere and have their dendrites in the main calyx. Each KC cell type sends axonal projections to a spatially segregated layer in the lobes. The dendritic arbors of each KC type also tend to be found in the same regions of the calyx (*Lin et al., 2007*; *Leiss et al., 2009*), but those dendritic zones are largely overlapping and individual KCs within a given cell type exhibit variable dendritic projection patterns (*Figures 6 and 7*). Moreover, the KCs receive input from an apparently random collection of glomeruli (*Murthy et al., 2008*; *Caron et al., 2013*; *Gruntman and Turner, 2013*). These features are in sharp contrast to most neuronal cell types in the olfactory pathway of the fly that are thought to consist of one to ten neurons that exhibit stereotyped projections (*Yu et al., 2010*), suggesting that their input and output connections are genetically predetermined. These observations suggest a unique function of the KCs in the processing of olfactory information (see 'Discussion').

## The MB output neurons

The MBONs extend dendrites that overlap with the KC axons in the MB lobes and project axons outside the MB. By determining the polarity of each cell type using high-resolution confocal imaging along with an analysis of the expression of the presynaptic reporter synaptotagmin-smGFP-HA (Syt::smGFP-HA; *Figure 8*), we identified 34 MBONs that comprise 21 different cell types (*Table 1*, *Figure 3B*, *Video 3*). We employed immunostaining to identify MBON types as either cholinergic, glutamatergic, or GABAergic (*Figure 9*, *Table 1*). MBONs that use the same neurotransmitter extend dendrites to adjacent regions of the lobes; cholinergic MBONs in the vertical (α and α′) lobes, glutamatergic MBONs in the medial (β, β′, and γ) lobes, and GABAergic MBONs in an area of the lobes at the intersection between these two regions (*Figure 9* and *Video 5*).

Fourteen MBON cell types consist of only one cell per hemisphere, six types contain two cells, and one type eight cells per hemisphere. In split-GAL4 lines with expression in more than one neuron, single-cell resolution was achieved by using the multicolor flp-out strategy (MCFO; Nern et al., in preparation, *Figure 8*). Single cell analysis revealed that each member of an MBON type exhibits indistinguishable morphology as assessed by light microscopy, and these stereotyped projection patterns are invariant across flies (see below for all cell types).

The 21 MBON types elaborate dendritic arbors in insular, segregated domains of the lobes that we call compartments. MBON dendritic arbors within each compartment exhibit little, if any, overlap with arbors in neighboring compartments (*Figure 10*). Computational alignment of the dendritic arbors of each of the MBON types within a single reference brain revealed that these compartments collectively tile the MB lobes with minimal overlap (*Figure 10G,I,K*). The alignment reveals gaps between arbors at four compartment borders; staining of the MB lobes for the presynaptic marker Bruchpilot (*Figure 10—figure supplement 1*) suggests that these gaps represent areas of reduced synaptic density. Two-color labeling experiments confirmed that the dendritic

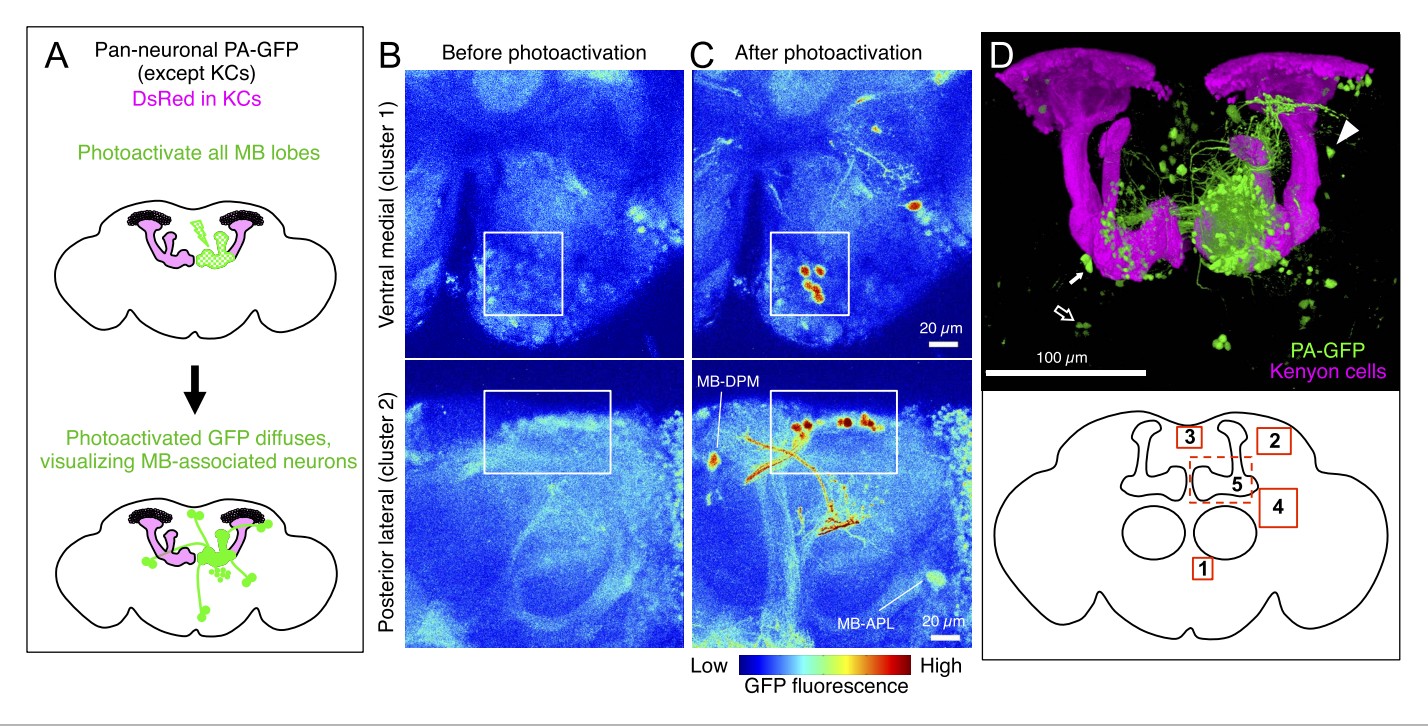

**Figure 4**. Identification of neurons innervating the MB lobes by PA-GFP tracing. (**A**) Schematic representation of the experiment. PA-GFP was expressed pan-neuronally, using a *synaptobrevin-GAL4* driver, except in the KCs where GAL4 activity was suppressed by *MB247-GAL80*. Photoactivation of the MB lobes was guided by a red fluorescent protein expressed broadly in the KCs (*MB247-DsRed*). A two-photon laser was used to achieve three-dimensional precision of the photoactivation. Photoactivation results in an increase in the fluorescence of PA-GFP molecules in the processes of neurons innervating the MB lobes. The photoactivated GFP molecules then diffuse throughout the cytoplasm of these neurons, revealing both cell bodies and processes (see 'Materials and methods' for details and limitations). (**B**) Baseline fluorescence of PA-GFP before photoactivation in the ventral medial (top; cluster 1 in panel **D**) or posterior lateral region of the brain (bottom; cluster 2 in panel **D**). Each panel represents a maximum intensity projection of z-stack images obtained with two-photon microscopy. (**C**) Fluorescence of PA-GFP after photoactivation of the right MB lobes in the same brain regions as in (**B**). The cell bodies as well as the processes of photoactivated neurons are labeled strongly with PA-GFP. The cell bodies of MB-DPM and MB-APL neurons were also visualized. (**D**) MB extrinsic neurons as identified by PA-GFP tracing upon photoactivation of the right MB lobes. Shown here is a three-dimensional reconstruction of a confocal stack. KCs are labeled in magenta (*MB247-DsRed*) and the cell bodies and the processes of photoactivated neurons are shown in green (native PA-GFP fluorescence). Five clusters of cell bodies comprising more than two cells were detected reproducibly (see 'Materials and methods') and are schematically indicated by the boxes (bottom): 1, ventral medial cluster; 2, posterior lateral cluster; 3, dorsal medial cluster; 4, anterior lateral cluster; and 5, delineated by the dashed lines, the anterior medial cluster that includes the PAM-cluster DANs. PA-GFP labeling in the contralateral (i.e., left) hemisphere includes the cell bodies of DANs (the PAM cluster DANs in the anterior medial region and the PPL1 cluster DANs in the posterior lateral region), as they innervate the MB lobes bilaterally. Eight individual neurons were also detected, which include MB-APL and MB-DPM in the ipsilateral hemisphere, and one MBON-β1>α, one MBON-γ4>γ1γ2 (closed arrow), one MBON-γ3, and one MBON-γ3β′1 (open arrow) in the contralateral hemisphere, because these extend dendrites in the contralateral MB lobes (see below). In addition, one cell body ventral to the calyx (arrowhead) and one cell body ventral lateral to the antennal lobe were reproducibly observed, which represent MBONs not identified in the split-GAL4 lines (see *Figure 5G* and *Figure 5—figure supplement 1*, also see 'Materials and methods'). The comparison of number of cells identified in split-GAL4 lines and in the PA-GFP tracing experiments is as follows: cluster name; cell counts by split-GAL4 lines, cell counts by PA-GFP tracing (sample size): (1) Ventral medial; 5, 5.2 ± 0.8 (n = 5). (2) Posterior lateral; 14–15, 15.9 ± 2.1 (n = 9). (3) Dorsal medial; 4, 5.4 ± 1.2 (n = 13). (4) Anterior lateral; 8–10, 5.9 ± 2.0 (n = 8). Note that the split-GAL4 line labeling the MBON-β′1 type, which comprises of 7–9 cells in the anterior lateral cluster (MB078C, see *Table 1*) includes at least three cells that exhibit very sparse dendritic arbors in the MB lobes as examined by multicolor flp-out experiments (data not shown). This was also observed in an independent experiment, in which the β′1 compartment was specifically photoactivated (*Figure 5B*) and may account for the fact that PA-GFP experiments identified fewer cells than the split-GAL4 lines in the anterior lateral cluster. It was not feasible to accurately count the large number of cells in the anterior medial cluster (cluster 5; indicated as a dashed box) comprising mostly the PAM DANs and several MBONs such as MBON-α1 (see *Figure 5* and 'Materials and methods'). Genotype of the fly was *yw/yw; UAS-C3PA-GFP(attP40),MB247-DsRed/UAS-SPA-GFP(attP40),MB247-GAL80; NSyb-GAL4/UAS-C3PA-GFP(attP2),UAS-C3PA-GFP(VK00005),UAS-C3PA-GFP(VK00027)*.

arbors of different MBONs are segregated in spatially stereotyped compartments (*Figure 11A–C*). We observed ensheathing glia at the borders between the MB lobes but not between the MBON compartments in each lobe (*Figure 11J–L*).

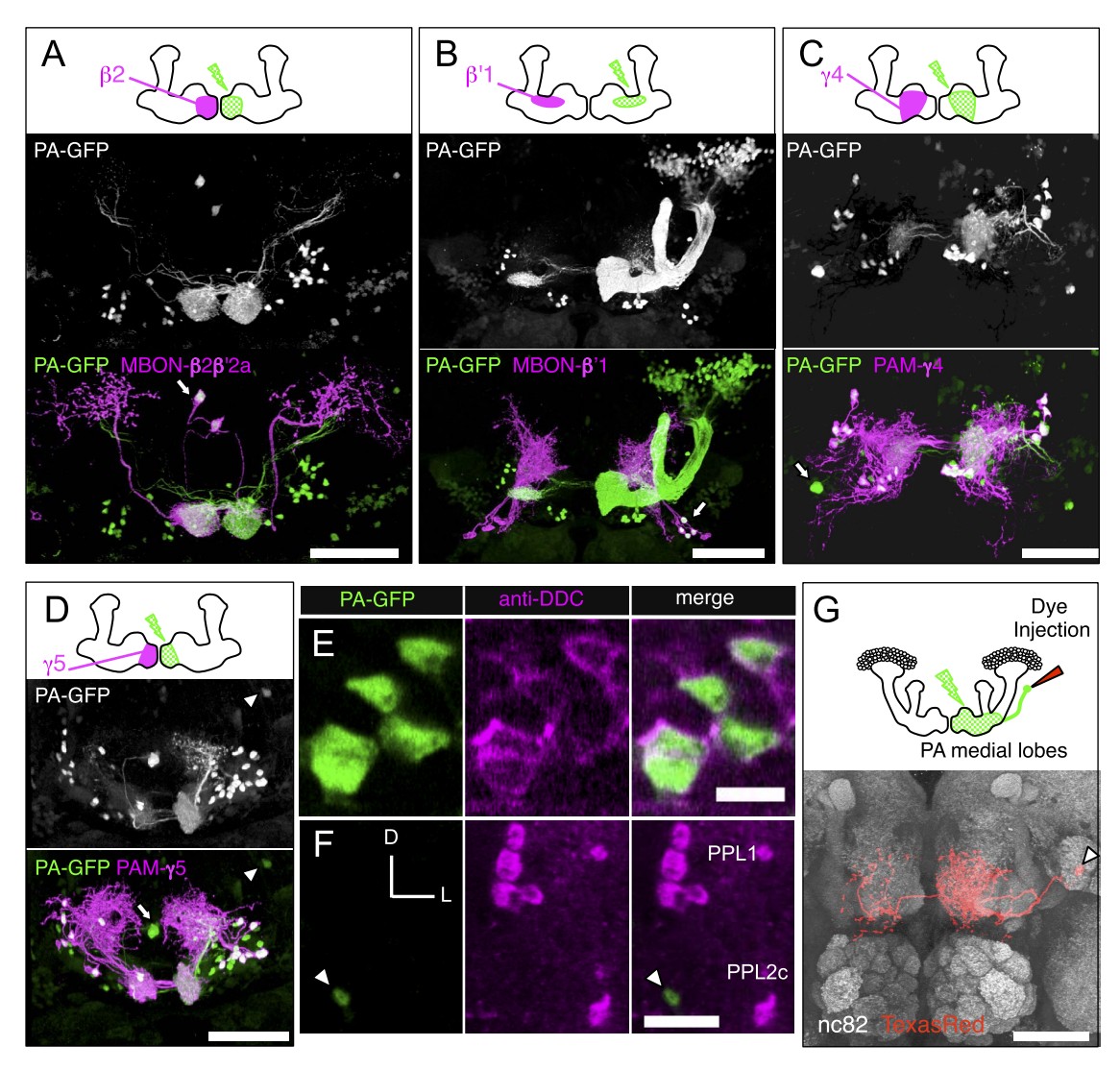

**Figure 5**. Identification of the MB extrinsic neurons innervating each MB lobe compartment by PA-GFP tracing. We photoactivated each of the 15 MB lobe subdomains (**Figure 1C**), or compartments (see below), individually to provide an independent approach to identify the extrinsic neurons associated with each compartment. PA-GFP was expressed pan-neuronally using a *synaptobrevin-QF* driver (see 'Materials and methods') but was suppressed in α/β and γ KCs using *MB247-QS*. Note that the α'/β' KCs were visualized when compartments in the α'/β' lobes were photoactivated (e.g., panel **B**). Different split-GAL4 lines (e.g., MBON lines for panels **A** and **B**, and DAN lines for panels **C** and **D**) were used to drive expression of membrane-targeted red fluorescent protein (myr::tdTomato) to demarcate a specific compartment within the lobes for photoactivation. The photoactivated samples were subsequently immunostained for a marker of dopaminergic cells (tyrosine hydroxylase [not shown] or dopa-decarboxylase [DDC, see panels **E** and **F**]) to classify individual photoactivated neurons as MBONs or DANs (see 'Materials and methods'). (**A–D**) Each panel shows a diagram of the experiment and a three-dimensional reconstruction of confocal images showing DANs and MBONs innervating the photoactivated compartment. Photoactivation was performed unilaterally (i.e., right) as indicated in the diagram. Native PA-GFP fluorescence is shown alone in gray scale in the middle panel and in green in the bottom panel together with myr::tdTomato signals driven by the indicated split-GAL4 in magenta; overlap is white. We identified all of the MBONs found in the split-GAL4 lines by photoactivation of each compartment (arrows and data not shown). We also identified DANs innervating each compartment (for example, myr::tdTomato negative and PA-GFP positive cells in panels **A** and **B**). We noted additional DANs in the PAM cluster that are not labeled by the split-GAL4 lines (green cell bodies in panels **C** and **D**). We therefore counted PAM cluster DANs associated with each medial lobe compartment based on these PA-GFP experiments (see 'Materials and methods'). The comparison of numbers of DANs in the PAM cluster identified in split-GAL4 lines and in the PA-GFP tracing experiments (i.e., photoactivated cells positive for tyrosine hydroxylase or dopa-decarboxylase) is as follows: compartment name; cell counts by split-GAL4 lines, cell counts by PA-GFP tracing (sample size): γ1; 13 (as mixtures of PAM-γ4 and PAM-γ4>γ1γ2 in MB312B), 3.8 ± 1.3 (n = 5). γ2; 13 (as mixtures of PAM-γ4 and PAM-γ4>γ1γ2 in MB312B), 7.8 ± 2.2 (n = 6). γ3; 9 (MB441B), 23.6 ± 6.3 (n = 5). γ4; 13 (as mixtures of PAM-γ4 and PAM-γ4>γ1γ2 in MB312B), 17.3 ± 1.3 (n = 6). γ5; 8 (MB315C), 21.5 ± 1.3 (n = 6). β'1; 14 (as mixtures of PAM-β'1ap and PAM-β'1m in MB025B),
*Figure 5. Continued on next page*

*Figure 5. Continued*

12.8 ± 1.3 (n = 4). β'2; 35 (sum of PAM-β'2a in MB109B, PAM-β'2m in MB032B, PAM-β'2p in MB056B, and PAM-β2β'2a in NP5272), 41.3 ± 6.6 (n = 4). β1; 5 (sum of PAM-β1 in MB063B and at least one PAM-β1 in MB194B), 6.6 ± 1.1 (n = 5). β2; 11 (sum of PAM-β2β'2a in NP5272 and PAM-β2 in MB209B), 18.8 ± 4.3 (n = 5). Thus, by simply summing the number of cells associated with these compartments, PA-GFP experiments identified ~154 DANs in the anterior medial cluster (i.e., PAM cluster) as compared to 121 cells labeled by the split-GAL4 lines. Note these numbers include PAM DANs innervating multiple compartments (PAM-γ4>γ1γ2 and PAM-β2β'2a) and are therefore an overestimate. The split-GAL4 collection identified 98 DANs of 14 types in the PAM cluster, whereas anti-dopamine immunostaining identified 115–135 DANs in the PAM cluster (*Liu et al., 2012*). (**E**) An example of immunostaining for dopa-decarboxylase (DDC) upon photoactivation of a single compartment. The panels represent a single confocal section of the PAM cluster upon photoactivation of the γ5 compartment. (**F**) The γ4γ5 extrinsic neuron identified by the PA-GFP tracing and not in the split-GAL4 screen shown in **D** (arrowhead) is DDC-negative and thus is likely an MBON. Images are maximum intensity projections. Dorsal is to the top and lateral is to the right. The cell body of this γ4γ5 MBON is positioned between the PPL1 and PPL2c DAN clusters. (**G**) Morphology of the γ4γ5 MBON identified by PA-GFP experiments with subsequent dye injection. Diagram of the experiment (top): Texas-Red dye was iontophoresed into the cell body identified by photoactivation (see 'Materials and methods'). A three-dimensional reconstruction of a confocal stack with neuropil labeled by nc82 immunostaining (gray) and the γ4γ5 MBON labeled by anti-Texas-Red immunostaining (red) is shown (bottom). The arrowhead indicates the cell body that is located close to the posterior surface of the brain ventral to the MB calyx. This neuron as well as an additional MBON with its cell body ventral lateral to the antennal lobe observed in PA-GFP experiments was found in a VT-GAL4 line (see *Figure 5—figure supplement 1* for single cell morphologies). Genotypes used: (**A–F**) *yw/yw; MB247-QS,QUAS-C3PA-GFP,QUAS-SPA-GFP/QUAS-C3PA-GFP,QUAS-SPA-GFP,UAS-myr::tdTomato(attP40); NSyb-QF,UAS-myr::tdTomato(attP2)/SplitGAL4DBD(attP2),SplitGAL4AD(VK00027)*. Split-GAL4 lines used: (**A**) *MB399C*; (**B**) *MB078C*; (**C**) *MB312C*; and (**D–F**) *MB315C*. Genotype of the animals in panel (**G**): *yw/yw; MB247-GAL80,UAS-C3PA-GFP/UAS-SPA-GFP; NSyb-GAL4/UAS-C3PA-GFP (attP2),UAS-C3PA-GFP (VK00005),UAS-C3PA-GFP (VK00027)*. Scale bars are 50 μm in (**A–D**) and (**G**), 5 μm in (**E**), and 20 μm in (**F**).

The following figure supplement is available for figure 5:

**Figure supplement 1**. Two MBONs not identified in the split-GAL4 screen.

The MB lobes are divided into 15 distinct compartments containing the segregated dendritic arbors of one or a small number of MBONs (*Figure 1C*, *Figure 12* and *Figure 13*). These compartments tile the MB lobes, revealing a general organizational principle of the MB output. This organization is in accord with an earlier proposal by *Tanaka et al. (2008)* that each of the γ, α'/β', and α/β lobes is divided into five domains. 13 of the 21 MBON types extend dendrites to a single compartment, and 8 MBON types project to two compartments (*Figure 12A–C*). Most of the MBON types innervate KCs from each of the layers within a compartment, but eight types restrict their dendritic arbors to specific layers (*Figure 11D, 12A,C,D*).

The identification of the full complement of 21 MBON types highlights the extensive convergence of 2000 KCs onto just 34 MBONs, a number even smaller than the number of glomeruli in the AL. Thus, the high-dimensional KC representation of odor identity is transformed into a low-dimensional MB output. This suggests that the MBONs do not represent odor identity but instead provide a representation that may bias behavioral responses (see 'Discussion').

## Dopaminergic neurons

Two clusters of dopaminergic neurons (PPL1 and PAM) have previously been shown to project axon terminals to specific regions within the MB lobes and transmit information about reward and punishment to the MB to guide learning (*Schwaerzel et al., 2003*; *Claridge-Chang et al., 2009*; *Mao and Davis, 2009*; *Aso et al., 2010, 2012*; *Burke et al., 2012*; *Liu et al., 2012*). Our split-GAL4 screen identified over 100 DANs of 20 types (*Figure 3C*, *Table 1* and *Video 4*). Each DAN type contains a small number of neurons: DAN types from the PPL1 cluster contain one or two cells per hemisphere and DAN types from the PAM cluster contain up to ~20 cells per hemisphere (*Table 1*, see *Figures 14–16* for each cell type, see 'Materials and methods' for classification).

The axon terminals of the DANs project to specific compartments and, similar to the dendrites of MBONs, tile the entire MB lobes (*Figure 10H,J,L, 14–16* and *Video 4*). 17 of the 20 DAN types project to only a single compartment (*Figure 12*). Two-color labeling and multicolor flp-out of DANs innervating neighboring compartments show clear segregation of their axon termini (*Figure 11E,F* and data not shown). Two-color labeling experiments revealed overlap of the axon termini of DAN types and the dendritic arbors of cognate MBON types that innervate the same compartment (*Figure 11G–I*). Computational alignment to a single reference brain extends these observations to all DAN types and further demonstrates that the DAN axon termini tile the MB lobes (*Figure 10H,J,L*). Thus, different MBON types have access to largely equivalent input from the KCs but are modulated by

**Table 1.** List of cell types in the mushroom body

| Categories* | | Putative transmitter† | Neurites in calyx or lobes‡ | Cell type names§ | Short names | Number of cells | Drivers# | Other names in literature¶ | Reference** |
|---|---|---|---|---|---|---|---|---|---|
| MB-intrinsic neurons | Kenyon cells | | C, L | γd | | ~75 | **MB607B**, MB419B | | Aso 2009 |
| | | | C, L | γmain | | ~600 | MB131B | | Aso 2009 |
| | | | C, L | α'/β'ap | | ~210 | **MB463B** | α'/β'a, α'/β'p | Tanaka 2008 |
| | | | C, L | α'/β'm | | ~140 | **MB418B** | | Tanaka 2008 |
| | | | C, L | α/βp | | ~90 | **MB371B** | | Tanaka 2008, Lin 2007 |
| | | | C, L | α/βs | | ~500 | **MB185B**, MB477B | | Tanaka 2008, Lin 2007 |
| | | | C, L | α/βc | | ~400 | **MB594B** | | Tanaka 2008, Lin 2007 |
| | modulatory neurons | 5HT amn | C, L | MB-DPM | | 1 | | DPM | Waddell 2000 |
| | | GABA | C, L | MB-APL | | 1 | VT43924 | | Tanaka 2008 |
| MB-extrinsic neurons | MB output neurons (MBONs) | glutamate | L | MBON-γ5β'2a | MBON-01 | 1 | **MB011B, MB210B** (MB002B) | MB-M6 | Tanaka 2008 |
| | | | L | MBON-β2β'2a | MBON-02 | 1 | MB399B | | |
| | | | L | MBON-β'2mp | MBON-03 | 1 | **MB002B, MB011B, MB210B** | MB-M4 | Tanaka 2008 |
| | | | L | MBON-β'2mp_bilateral | MBON-04 | 1 | **MB011B** | | |
| | | | L | MBON-γ4>γ1γ2 | MBON-05 | 1 | **MB434B**, MB298B | | |
| | | | L | MBON-β1>α | MBON-06 | 1 | **MB434B**, MB433B | MB-MV2 | Tanaka 2008 |
| | | | L | MBON-α1 | MBON-07 | 2 | **MB310C, MB323B**, MB319C | | |

*Table 1. Continued on next page*

*Table 1. Continued*

| Categories* | Putative transmitter† | Neurites in calyx or lobes‡ | Cell type names§ | Short names | Number of cells | Drivers# | Other names in literature¶ | Reference** |
|---|---|---|---|---|---|---|---|---|
| | GABA | L | MBON-γ3 | MBON-08 | 1 | **MB083C, MB110C** | | |
| | | L | MBON-γ3β′1 | MBON-09 | 1 | **MB083C, MB110C** | | |
| | | L | MBON-β′1 | MBON-10 | 8†† | MB057B, MB078C | | |
| | | L | MBON-γ1pedc>α/β | MBON-11 | 1 | **MB112C**, MB085C, MB262B | MB-MVP2 | Tanaka 2008 |
| | acetylcholine | L | MBON-γ2α′1 | MBON-12 | 2 | **MB077B**, MB051B, MB090C | | |
| | | L | MBON-α′2 | MBON-13 | 1 | **MB018B**, MB091C | MB-V4 | Tanaka 2008 |
| | | L | MBON-α3 | MBON-14 | 2 | MB082C, MB093C, G0239 | MB-V3 | Tanaka 2008, Chiang 2011 |
| | | L | MBON-α′1 | MBON-15 | 2 | MB543B | | |
| | | L | MBON-α′3ap | MBON-16 | 1 | **MB027B, MB549C** | MB-V2α′ | Tanaka 2008, Sejourne 2011 |
| | | L | MBON-α′3m | MBON-17 | 2 | **MB027B** | MB-V2α′ | Tanaka 2008, Sejourne 2011 |
| | | L | MBON-α2sc | MBON-18 | 1 | **MB549C**, MB080C, MB050B | MB-V2α | Tanaka 2008, Sejourne 2011 |
| | | L | MBON-α2p3p | MBON-19 | 2 | MB542B, (MB062B) | | |
| | N.D. | L | MBON-γ1γ2 | MBON-20 | 1 | (VT999036) | | |
| | | L | MBON-γ4γ5 | MBON-21 | 1 | (VT999036) | | |
| | | C | MBON-calyx | MBON-22 | 1 | **MB622B**, MB242A | MB-CP1 | Tanaka 2008 |
| modulatory neurons | dopamine | L | PAM-γ5 | PAM-01 | 8–21## | **MB315C**, MB335C | aSP13, MB-M1? | Tanaka 2008, Keleman 2012 |
| | | L | PAM-β′2a | PAM-02 | 6–9 | **MB109B** | | |

Table 1. Continued

| Categories* | Putative transmitter† | Neurites in calyx or lobes‡ | Cell type names§ | Short names | Number of cells | Drivers# | Other names in literature¶ | Reference** |
|---|---|---|---|---|---|---|---|---|
| | | L | PAM-β2β'2a | PAM-03 | >3 | MB301B, NP5272 | MB-M3, MB-M1? | Tanaka 2008 |
| | | L | PAM-β2 | PAM-04 | 8–19 | MB209B | MB-M8 subset | Perisse 2013 |
| | | L | PAM-β'2p | PAM-05 | 14–17 | **MB056B** | MB-M5? MB-AIM? | Tanaka 2008 |
| | | L | PAM-β'2m | PAM-06 | 12–15 | **MB032B** | MB-M5? MB-AIM? | Tanaka 2008 |
| | | L | PAM-γ4<γ1γ2 | PAM-07 | 13–17 | **MB312B** | subset of MB-AIM? | Tanaka 2008 |
| | | L | PAM-γ4 | PAM-08 | | **MB312B** | subset of MB-AIM? | Tanaka 2008 |
| | | L | PAM-β1ped | PAM-09 | 1–3 | (MB194B) | subset of MB-MVP1 | Tanaka 2008 |
| | | L | PAM-β1 | PAM-10 | 4–6 | **MB063B** | subset of MB-MVP1 and MB-M8 | Tanaka 2008, Perisse 2013 |
| | | L | PAM-α1 | PAM-11 | >6 | MB043B, MB299B | subset of MB-MVP1, MB-VP1 | Tanaka 2008 |
| | | L | PAM-γ3 | PAM-12 | 9–23 | **MB441B** | MB-M2 | Tanaka 2008 |
| | | L | PAM-β'1 ap | PAM-13 | 13–14 | MB025B | subset of MB-AIM? | Tanaka 2008 |
| | | L | PAM-β'1 m | PAM-14 | | MB025B | subset of MB-AIM? | Tanaka 2008 |
| | | L | PPL1-γ1pedc | PPL1-01 | 1–2 | **MB438B** | MB-MP1, MP | Tanaka 2008, Krashes 2009 |
| | | L | PPL1-γ1‡‡ | PPL1-02 | 1 | (TH-GAL4) | | |
| | | L | PPL1-γ2α'1 | PPL1-03 | 1 | **MB296B**, MB439B | MB-MV1 | Tanaka 2008 |
| | | L | PPL1-α'3 | PPL1-04 | 1 | **MB304B** | | Mao 2008 |
| | | L | PPL1-α'2α2 | PPL1-05 | 1 | **MB058B** | MB-V1 | Tanaka 2008 |
| | | L | PPL1-α3 | PPL1-06 | 1 | MB065B, MB060B | | Mao 2008 |
| | | C | PPL2ab§§ | | >1 | (TH-GAL4) | | Mao 2008 |
| | | C | PPL2ab | | >1 | (TH-GAL4) | | Mao 2008 |

Table 1. Continued on next page

*Table 1. Continued*

| Categories* | Putative transmitter† | Neurites in calyx or lobes‡ | Cell type names§ | Short names | Number of cells | Drivers# | Other names in literature¶ | Reference** |
|---|---|---|---|---|---|---|---|---|
| | octopamine | C, L | OA-VPM3 | | 1 | MB022B | | Busch 2009 |
| | | L | OA-VPM4 | | 1 | MB022B, MB021B, MB113C | | Busch 2009 |
| | | C | OA-VPM5 | | 1 | (TDC2-GAL4) | | Busch 2009 |
| | | C | OA-VUM2a | | 2 | (TDC2-GAL4, NP7088) | | Busch 2009 |
| | 5HT | C | CSD | | 1 | **MB465C** | | Roy 2007 |
| | SIFamide | C, L | SIFamide | | 4 | **MB013B** | MB-C2? | Verleyen 2004, Tanaka 2008 |
| | GABA | C | MB-C1 | | >2 | **MB380B** | | Tanaka 2008 |

MB intrinsic and extrinsic cell types are listed. The number of cells in each cell type per brain hemisphere is given. We have made split-GAL4 lines that drive expression in most cell types and these are listed. **Supplementary file 1** lists the cell types in which expression is observed with each split-GAL4 driver. For the remaining cell types, the GAL4 line used to visualize individual neuron(s) within each cell type is given; in nearly all cases, these lines have much broader expression than the split-GAL4 lines. MBON cell types were characterized at the single cell level (see text for details). For ten MBON cell types, the relevant split-GAL4 lines label only one cell per hemisphere. For other MBONs, the relevant split-GAL4 lines expressed in two or more cells per hemisphere and we used MCFO and determined single cell morphologies to classify cell types.

*The MB neurons were categorized into two major groups of cell types: the MB-intrinsic neurons that have all processes in the MB and thus contribute to local signal processing inside the MB, whereas the MB-extrinsic neurons are any neurons that connect the MB with other neuropils irrespective of polarity and transmitter. We defined MBONs as neurons that have dendrites in the MB. The antennal lobe projection neurons are also MB-extrinsic neurons by definition, but not listed. The polarity of neurons was often obvious in high-resolution confocal images of membrane labeled cells (**Figure 8A,B**) in which presynaptic terminals and dendrites exhibit distinctive morphologies. However, in each case we confirmed the polarity using a reporter that localizes to presynapses by virtue of the synaptic vesicle protein Synaptotagmin (pJFRC51-3XUAS-Syt::smGFP-HA in su(Hw)attP1; **Figure 8C**). In addition to the 21 MBON cell types, one neuron (MBON-calyx, or MB-CP1) extends dendrites in the main calyx, the posterior part of the pedunculus and the ventral part of the lateral horn, and thus we classify this as an atypical MB output neuron (see **Figure 3—figure supplement 1**).

†Four of the MBON cell types have been previously described as ChAT immunoreactive (**Sejourne et al., 2011; Placais et al., 2013**), and we observed consistent results. Likewise, our results with anti-dVGluT staining are consistent with conclusions based on the expression pattern of an enhancer trap line in the *dVGluT* locus, OK371 (**Johard et al., 2008**). N.D.; not determined.

‡Neurite distribution inside the MB is indicated: L, lobes; C, calyx.

*Table 1. Continued on next page*

Table 1. Continued

§As described in the text, MB-extrinsic neurons project to specific compartments in the MB lobes with minimal overlap. This allowed us to adopt a nomenclature for MB-extrinsic neurons based on their projection pattern in the MB: for MBONs, we use MBON plus the name of MB lobe compartments in which their dendrites arborize (e.g., MBON-α1 for the MB output neurons that have dendrites in the α1 compartment); for DANs, we use the name of cluster to which they belong, either PPL1 or PAM, followed by the name of the MB compartments in which their axon terminals are found (e.g., PAM-α1 for the dopaminergic neurons that terminate in α1). For the MBONs with axon terminals in the MB lobes, the '>' symbol followed by the compartments, or the lobes, in which the axon terminals innervate is appended to the dendritic compartment (e.g., MBON-β1>α for the MB output neuron with dendrites in β1 and presynaptic terminals in the α lobe). For the DAN type with dendrites in the MB lobes, the '<' symbol followed by the compartments in which its dendritic arbors elaborate is appended to the compartment with axon terminals (e.g., PAM-γ4<γ1γ2 for the PAM neuron that terminates in γ4 and has dendrites in γ1 and γ2). We also provide a short name for each MBON and DAN cell type of the format MBON-01 to MBON-22, PAM-01 to PAM-14, and PPL1-01 to PPL1-06.

#Bold font indicates lines highly recommended for behavioral experiments based on specificity and intensity. The best line for behavioral experiments depends on the UAS effectors. See **Supplementary file 1** for more information.

¶Some of the cell types we define here were previously named or included in a mixture of cell types with a single name. Our judgment of the most likely correspondence is given. Some assignments are uncertain, as indicated by a question mark.

\*\*See reference list in the main text.

††Unlike other MBON cell types, the number of MBON-β'1 cells in MB057B varied between 7 and 9 (8.0 on average, SEM = 0.17, n = 12) across individuals. Also, the distribution of dendrites in the β'1 compartment differed among 24 single cells obtained with MCFO. Some extend dendrites in both ap and m layers, but others extend dendrites only in one of the layers. Three of 24 single cells examined exhibit very sparse dendritic arbors in the β'1 compartment.

‡‡A sixth cell type in the PPL1 cluster that sparsely innervates the γ1 MB compartment has been observed in stochastic labeling experiments using the TH-GAL4 driver (PPL1-γ1) (Igor Siwanowicz, pers. comm.), but we were not able to generate a split-GAL4 for this cell type.

§§Two cell types of DANs in PPL2ab cluster have been identified (**Mao and Davis, 2009**).

##Numbers of cell for each PAM cluster DAN cell type were estimated by counting labeled cells in split-GAL4 (minimal estimate) or PA-GFP experiments (maximal estimate). See **Figure 5** for detail.

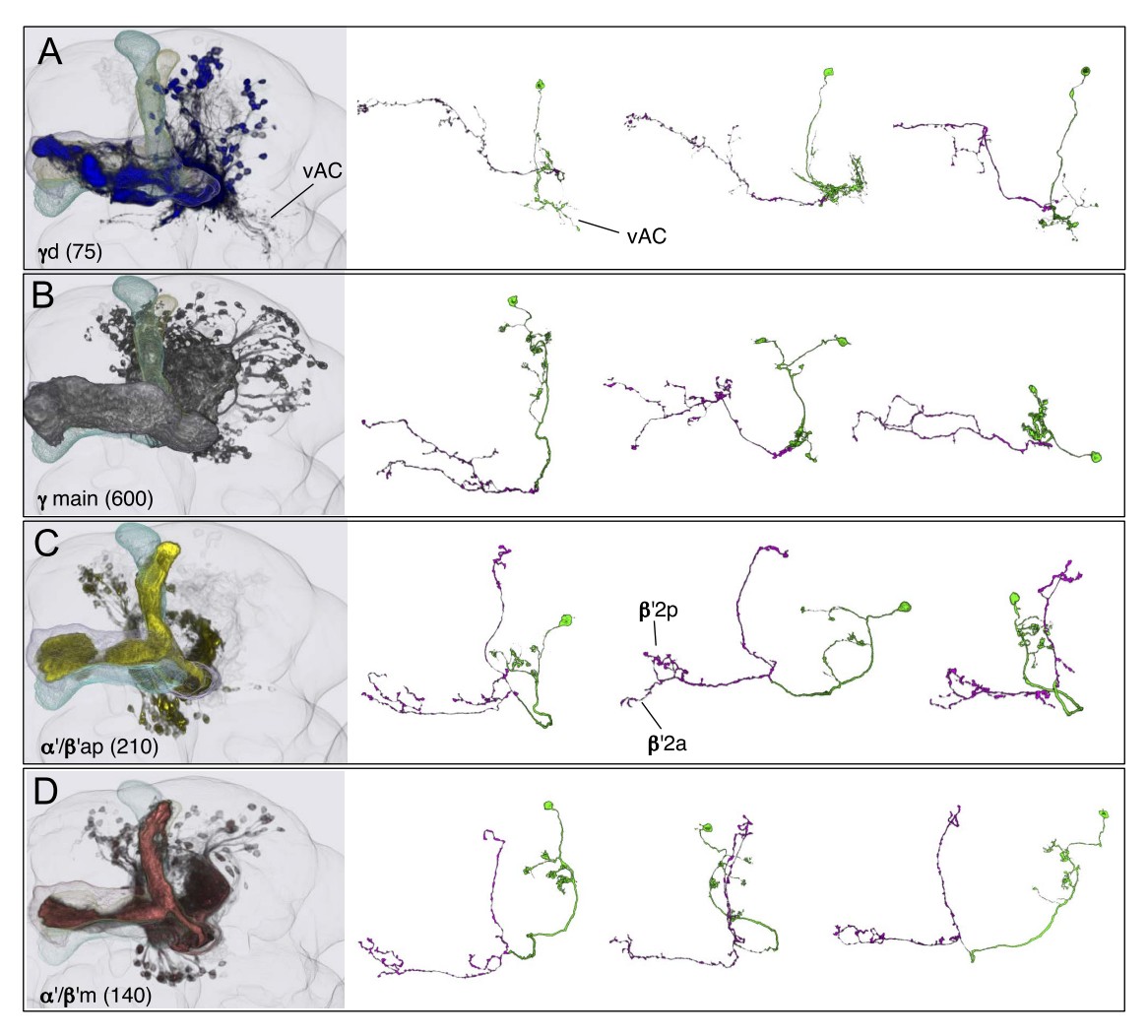

**Figure 6**. KCs of the γ and α'/β' lobes. Seven KC types identified by split-GAL4. Representative aligned images of each KC type (see also *Figure 7*) are shown; the name of the cell type and the approximate number of cells per brain hemisphere are indicated. Three examples of single cell morphologies, segmented from multicolor flip-out experiments, are presented for each cell type illustrating the branching pattern of the KC axons as they project through the lobes. In the single cell images, the cell body, primary neurite, dendrites, and axon in the pedunculus have been false-colored green and axons in the lobes are magenta. Based on co-expression in specific split-GAL4 lines and single cell morphologies, we have divided the KC population into 7 cell types. The number of cells of each type was estimated by counting labeled nuclei in split-GAL4 lines (see 'Materials and methods') and is shown in parentheses. (**A**) The γd KCs are thought to be of embryonic origin, because they are not included in a clonal analysis that visualized all post-embryonic KCs (*Lin et al., 2007*; *Yu et al., 2013*), and they have morphological similarity to the embryonic born KCs of basal cockroaches (*Farris and Strausfeld, 2003*). Their dendritic arbors form a protrusion extending ventral lateral to the main calyx, which we named the ventral accessory calyx (vAC). Their axons occupy the most peripheral layer in the pedunculus, the ventral and anterior layers in the γ1–γ4 compartments, and the dorsal layer in γ5. Single cell morphologies (from MB028B and MB355B) reveal that the γd axons have more branches in γ3–γ5 than in γ1 and γ2. The vAC may be devoted to non-olfactory inputs; the major types of olfactory projection neurons from the antennal lobe do not innervate this structure (*Butcher et al., 2012*). (**B**) The γ main KCs have their dendrites in the main calyx and their axons occupy about 75% of the volume of the γ lobes. Single cell morphologies (from MB369B and MB355B) reveal that each γ main KC branches in all the five compartments of the γ lobe. (**C**) The α'/β'ap KCs have dendrites in the main calyx and project axons to the anterior and posterior layers of the α'/β' lobes. Single cell morphologies (from MB461B and MB463B) reveal that axonal branches from single KCs project to both β'2a and β'2p, where they overlap with distinct sets of output and dopaminergic neurons (see below). (**D**) The α'/β'm KCs have dendrites in the main calyx. In β'2, their axons are located in the area between the bifurcating axons of α'/β'ap neurons shown in panel **C**. In the α' lobe, their axons are medial to those of the α'/β'ap cells. Single cell images are from MB418B and MB369B.

different DANs (*Figures 12 and 13*). In classical learning paradigms, different DAN types respond to different unconditioned stimuli (US) (*Riemensperger et al., 2005*; *Mao and Davis, 2009*; *Burke et al., 2012*; *Liu et al., 2012*). The compartmental organization we observe suggests that the DANs may

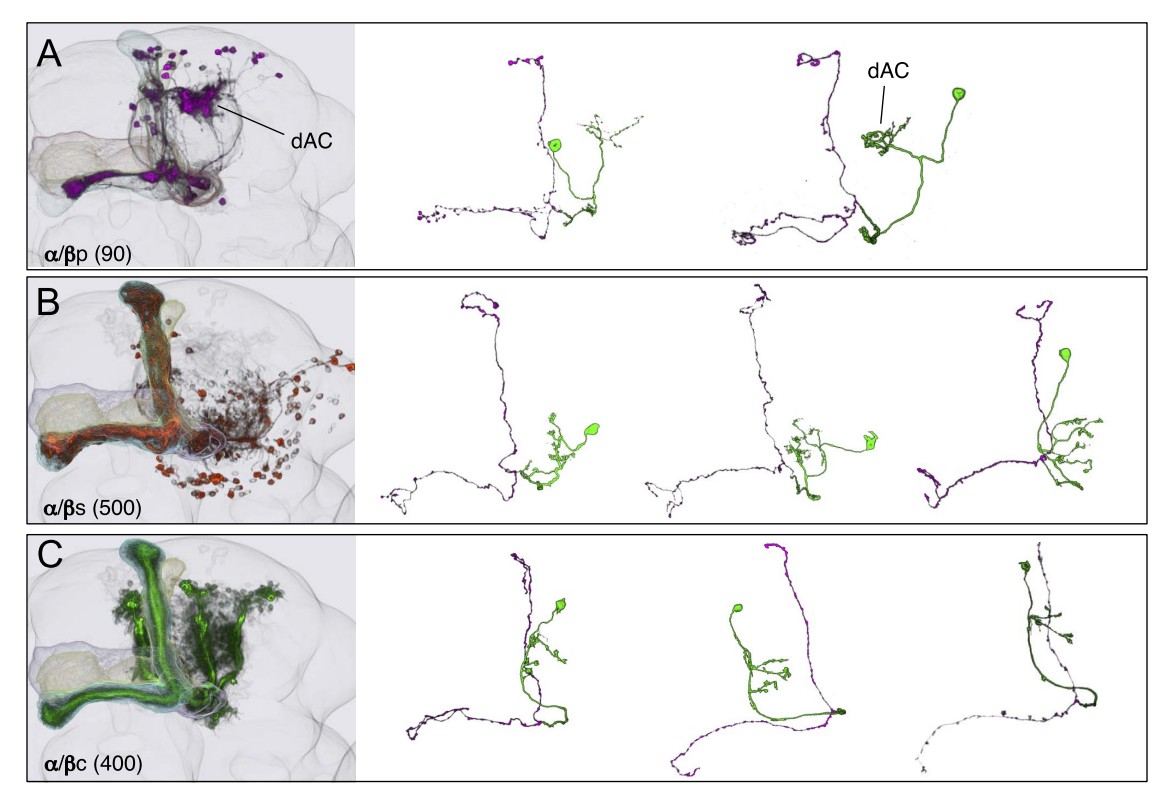

**Figure 7**. KCs of the α/β lobes. (**A**) The dendrites of the α/β posterior (α/βp) KCs form a protrusion extending to the dorsal lateral side of the main calyx. This structure has been called the accessory calyx, but we have renamed it as the dorsal accessory calyx (dAC) to distinguish it from the ventral AC (vAC) (**Figures 1C and 6A**). The α/βp KC axons project to the posterior layer of the α/β lobe. These are the firstborn α/β KCs and are also known as pioneer α/β KCs (**Lin et al., 2007**). The single cell images were segmented from multicolor flp-out (MCFO) brains of MB469B and MB371B. (**B**) The α/β surface (α/βs) KCs have dendrites in the main calyx and project axons to the surface layer of the α/β lobes where they form a continuous layer surrounding the α/β core KCs shown in (**C**). Single cell morphologies of cells (from MB185B) reveal that the α/βs KCs have relatively smooth axonal projections in the lobes. (**C**) The α/β core (α/βc) KCs have dendrites in the main calyx. They are the last born KCs and their axons occupy the core of the pedunculus and the α/β lobes. They can be morphologically subdivided into inner and outer core cells (**Tanaka et al., 2008**), although the border between the inner and outer core is not well defined and we were unable to make a split-GAL4 driver line that labels only the outer core cells. Single cell morphologies (from MB594B) reveal that the axons of the α/βc cells have the fewest branches of the 7 types of KCs.

convey information about the US to specific MBON types. Dopamine release in specific compartments may modify local KC–MBON synapses to bias behavioral output.

## Interactions between the MBON compartments

We identified three MBONs and one DAN type that appear to interconnect different compartments of the MB lobes (**Figure 12A**, **Figure 14—figure supplement 1E**, and **Figure 17A–D**). Each of these three MBONs projects to the compartments of other MBONs but not back to the compartment occupied by its own dendrites. Thus information flows in one direction, and these MBON connections could create a multi-layered feedforward network (**Figure 17J**, see 'Discussion'). 12 of the MBON types receive input from the KCs but not from other MBONs and therefore read out KC activity as a single output layer. Most of the MBONs having dendrites in the α'/β' and γ lobes provide such a single-layer readout. In contrast, the glutamatergic MBON whose dendrites arborize in the γ4 compartment projects axons into the γ1 and γ2 compartments as well as to neuropils outside the MB (**Figure 17A,G–I**). Thus, the γ1 and γ2 MBONs have access to direct KC input as well as to the input provided by the γ4 MBON. The outputs of these two γ lobe compartments thus reflect a two-layer feedforward network. The readout of the α/β lobe is even more elaborate because the γ1 MBON projects to all of the compartments of the α/β lobe (**Figure 17C,E,F**), and the β1 MBON projects to the α1, α2, and α3 compartments (**Figure 17B**). The α1, α2, and α3 MBONs thus represent a four-layer feedforward

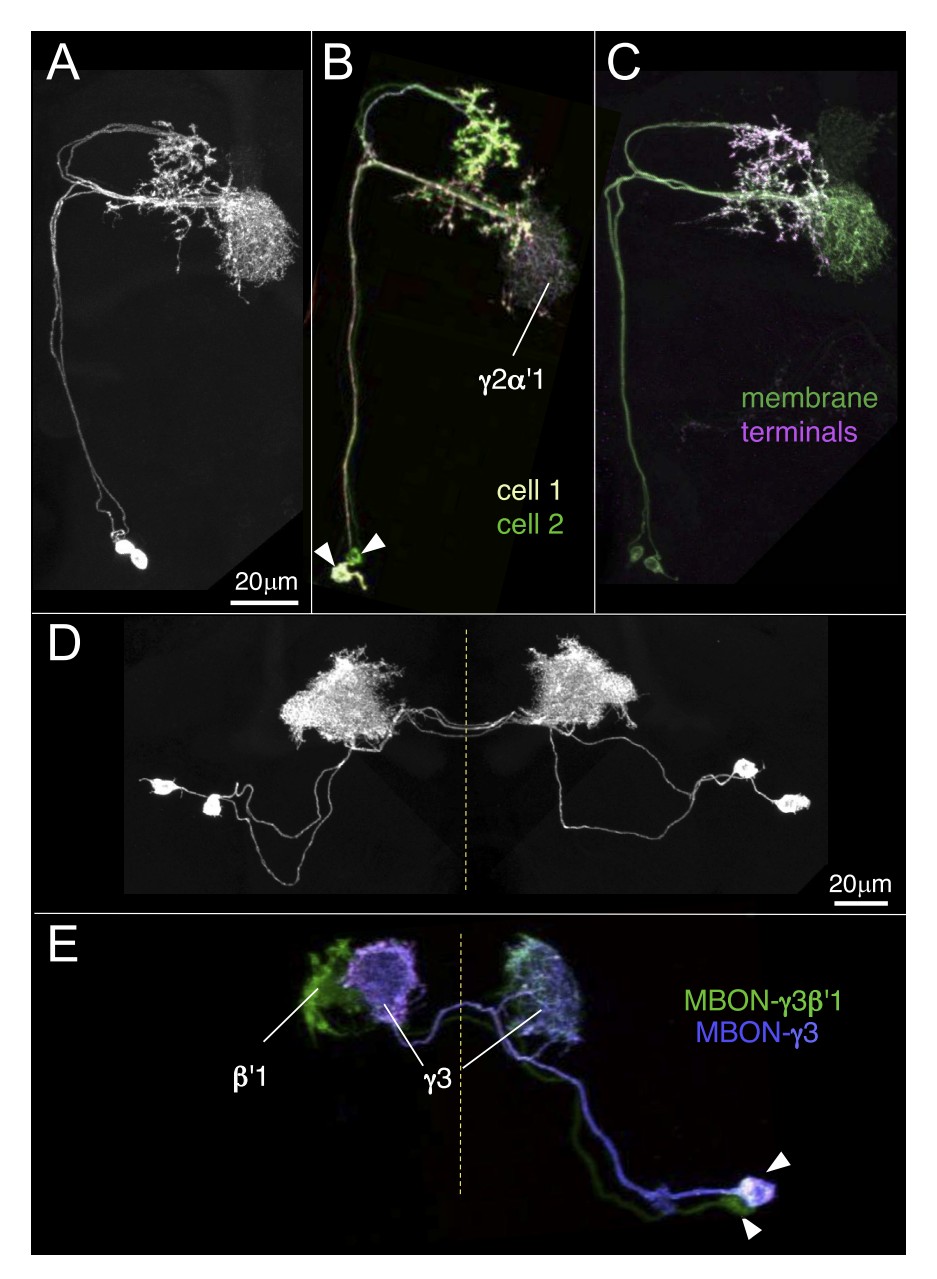

**Figure 8**. Identification of MBONs and visualization of their single cell morphologies. Each MBON is named according to the compartment(s) in the MB lobes where its dendrites arborize (see **Table 1** and below). For example, MBON-γ2α'1 neurons exhibit dendritic arbors in γ2 and α'1 compartments. (**A**) The projection patterns of MBON-γ2α'1 neurons. Maximum intensity projection confocal images of MB077B driven expression in one brain hemisphere are shown. Visualization with *pJFRC225-5XUAS-IVS-myr::smGFP-FLAG* in *VK00005* labels two MBON-γ2α'1 neurons per hemisphere. (**B**) Labeling of the two MBON-γ2α'1 neurons in different colors using multicolor flp-out (MCFO; Nern et al., in preparation; see 'Materials and methods'). The arbors of the two neurons overlap and are indistinguishable, thus these two cells represent a single cell type. Arrowheads indicate cell bodies. (**C**) MB077B driven expression of a membrane targeted epitope (green; *pJFRC225-5XUAS-IVS-myr::smGFP-FLAG* in *VK00005*) and a presynaptically targeted epitope (magenta; *pJFRC51-3XUAS-IVS-Syt::smGFP-HA* in *su(Hw)attP1*). The fine processes in the MB lobes are typical of dendrites, whereas the processes of this neuron that are outside the lobes end with varicosities containing the presynaptic marker. (**D**) The morphologies of MBON-γ3 and MBON-γ3β'1 as identified by MB083C driven expression. Using the *pJFRC225* membrane-targeted reporter, two neurons innervating the γ3 and β'1 MB compartments in each hemisphere can be seen. Dashed vertical line

*Figure 8. Continued on next page*

*Figure 8. Continued*

shows the position of the mid-line. (**E**) Using MCFO, the two cells in one hemisphere were labeled in different colors. Both cells have dendrites in γ3 in both hemispheres but axonal terminals in just the contralateral brain hemisphere. However, one of them (the green cell) also has dendrites in the contralateral β'1 compartment, demonstrating that these two cells represent different cell types, which we named MBON-γ3 and MBON-γ3β'1.

network with access to the KC activity of the α/β and γ lobes, both directly and indirectly through MBON intermediaries (*Figure 17J*).

The DAN cell type that projects axons to the γ4 compartment (*Figure 17D*) provides another form of communication between compartments. This cell type extends dendrites to the γ1 and γ2 compartments, where they appear to receive direct inputs from the axonal termini of MBON-γ4>γ1γ2 (*Figure 17G–I*), creating a recurrent loop involving modulatory dopamine input.

The MBON axons that innervate other compartments within the MB lobes have access to dopamine inputs and thus can potentially be modified by learning. Adaptive multi-layer or 'deep' feedforward networks are known to be capable of more complex readout functions than single-layer readouts (*Bishop, 2006*). Thus, the four-layer α lobe readout system may support more sophisticated neuronal

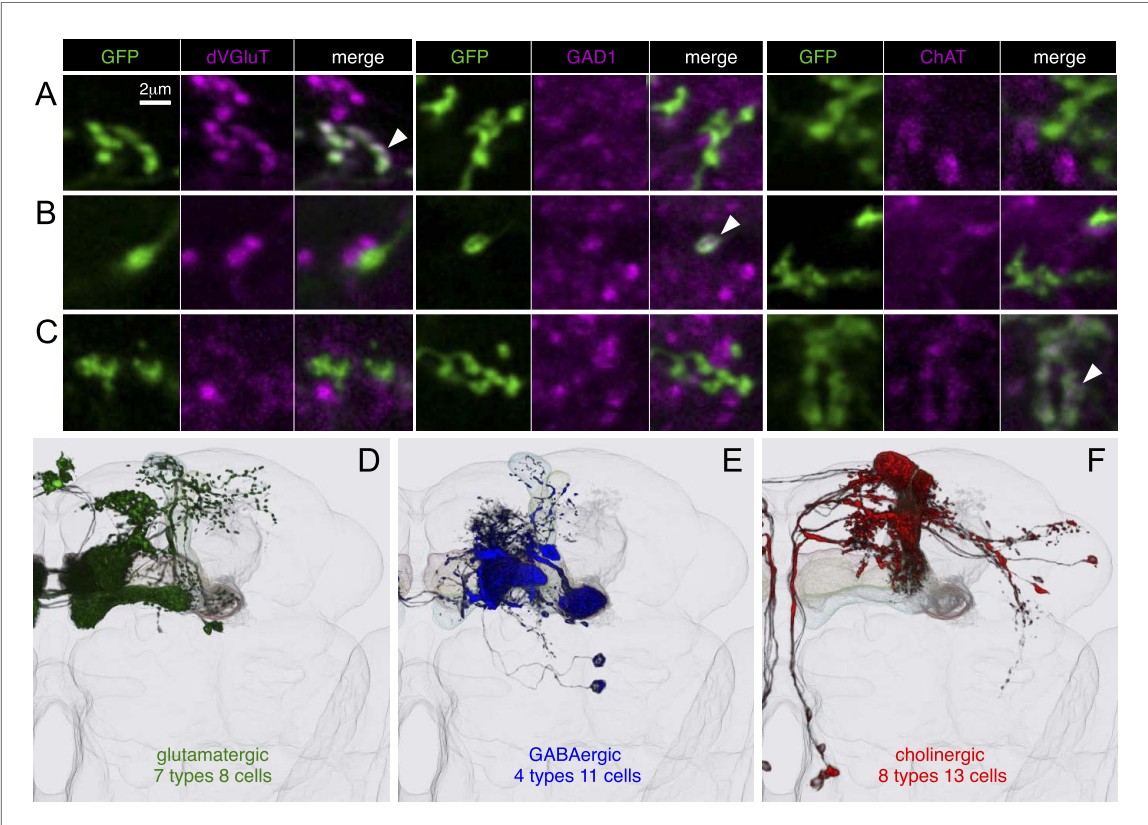

**Figure 9**. Neurotransmitters used by MBONs. The putative neurotransmitters used by MBON cell types were assigned by assessing the immunoreactivity of their axon terminals to antibodies raised against Drosophila vesicular glutamate transporter (dVGluT), Drosophila glutamate decarboxylase 1 (GAD1), and Drosophila choline acetyl transferase (ChAT) (see 'Materials and methods'). Single confocal optical sections; axon terminals of different MBONs are shown in green and antibody staining in magenta. (**A**) Axon terminals of MBON-γ5β'2a (MB210B) were labeled with anti-dVGluT (arrowhead) but not with either anti-GAD1 or anti-ChAT, suggesting that this cell type is glutamatergic. (**B**) Axon terminals of MBON-γ1pedc>α/β (MB112C) were labeled with anti-GAD1 (arrowhead), suggesting that this cell type is GABAergic. (**C**) Axon terminals of MBON-γ2α'1 (MB077B) were labeled with anti-ChAT (arrowhead), suggesting this cell type is cholinergic. (**D–F**) MBONs of the same neurotransmitter type were given the same color and are displayed together. (**D**) Seven types of glutamatergic MBONs. (**E**) Four types of GABAergic MBONs. (**F**) Eight types of cholinergic MBONs. The neurotransmitter of the two MBON types found in the VT-GAL4 collection was not determined (but see *Figure 5—figure supplement 1*). *Video 5* illustrates the relative positions of these neurons in the standard brain.

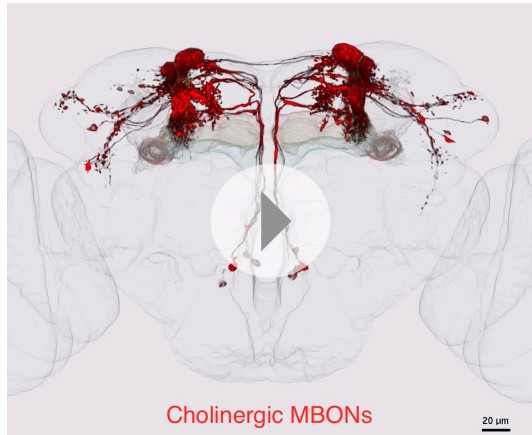

Cholinergic MBONs

20 μm

**Video 5**. Arrangement of MBONs by transmitter type. MBONs was color-coded based on putative neurotransmitter as in *Figure 9*: green, glutamatergic; blue, GABAergic; red, cholinergic.

computations than the one- and two-layer γ or α'/β' lobe systems.

## Inputs to the DANs and outputs of the MBONs

The 34 MBONs represent the sole outputs of the MB lobes. Computational alignment of single-cell images allowed us to localize the axon termini of the MBONs outside the MB. The axon terminals of the MBONs (*Figure 18*) converge onto five neuropils: the crepine (CRE; a region surrounding the horizontal/medial lobes), the superior medial protocerebrum (SMP), the superior intermediate protocerebrum (SIP), the superior lateral protocerebrum (SLP), and the lateral horn (LH) (*Figure 18E*). Most MBON types project axons to several of these neuropils (*Figure 18—figure supplement 1*). Within a neuropil, the axon terminals of each MBON type exhibit distinct and confined projection patterns, suggesting that different MBON types may synapse onto different neurons (*Figure 19*, also see *Figures 14–16* for individual MBON projection patterns). However, we also observed significant overlap between the axon terminals of different MBONs, suggesting that in certain cases MBONs may converge onto the same post-synaptic target neuron (*Figure 20A,D,E*). These results, obtained by computational alignment, were confirmed for a subset of MBONs by two-color labeling experiments (*Figure 20J–L*; *Video 6*).

Computational alignment of single cell images of each DAN type identified the dendritic arbors of the DANs outside the MB (*Figure 18D*). Interestingly, 90% of the dendritic arbors of the DANs reside in four of the neuropils targeted by the MBONs: CRE, SMP, SIP, and SLP (*Figures 18E and 19*). Because DANs are activated by unconditioned stimuli (US), these neuropils are likely targets of the US input (see 'Discussion'). We observed cases where the dendrites of different DAN types in a single neuropil overlap (*Figure 20B,F,G*), suggesting that they may share upstream input. Furthermore, we observed overlap between MBON terminals and DAN dendrites within each neuropil, implying that some MBONs may synapse on specific DANs (*Figure 20C,H,I*). In some cases, the axon terminals of an MBON overlap with the dendrites of the DAN type innervating the same compartment (*Figure 20H*). This may provide feedback that regulates dopamine release and hence learning. In other cases, axon termini of MBONs from one compartment overlap with the dendrites of DAN types projecting to different compartments (*Figure 20I*). This provides a pathway through which the activity of one MBON could modulate the synapses between KCs and MBONs of other compartments.

The MBONs are thought to elicit learned behavioral responses, and it is therefore of interest that four MBON types project to the lateral horn (*Figure 18—figure supplement 1*), a brain region responsible for innate odor responses (*Heisenberg et al., 1985*). These cholinergic MBONs may exploit LH neurons capable of eliciting innate behaviors to generate learned olfactory responses (*Sejourne et al., 2011*). This connection may also serve to modulate innate behavioral responses as a consequence of learning. Most of the MBON outputs, however, do not target the LH but converge onto the neuropils surrounding the MB (*Figure 18*). These convergent loci also receive projections from the antennal lobe and the lateral horn and are likely to be major sites of integration and processing of information within the fly brain (*Aso et al., 2014*).

## Discussion

We have identified the neurons that comprise the MB and characterized their projection patterns. These data provide a comprehensive map describing the potential connections of neurons in the MB lobes. The MB lobes are innervated by approximately 2000 KCs, 34 MBONs of 21 cell types, and about 130 modulatory DANs of 20 cell types that fall into two families, PAM and PPL1. These data, extending the level of detail and completeness of previous studies of MB anatomy (*Tanaka et al., 2008*;

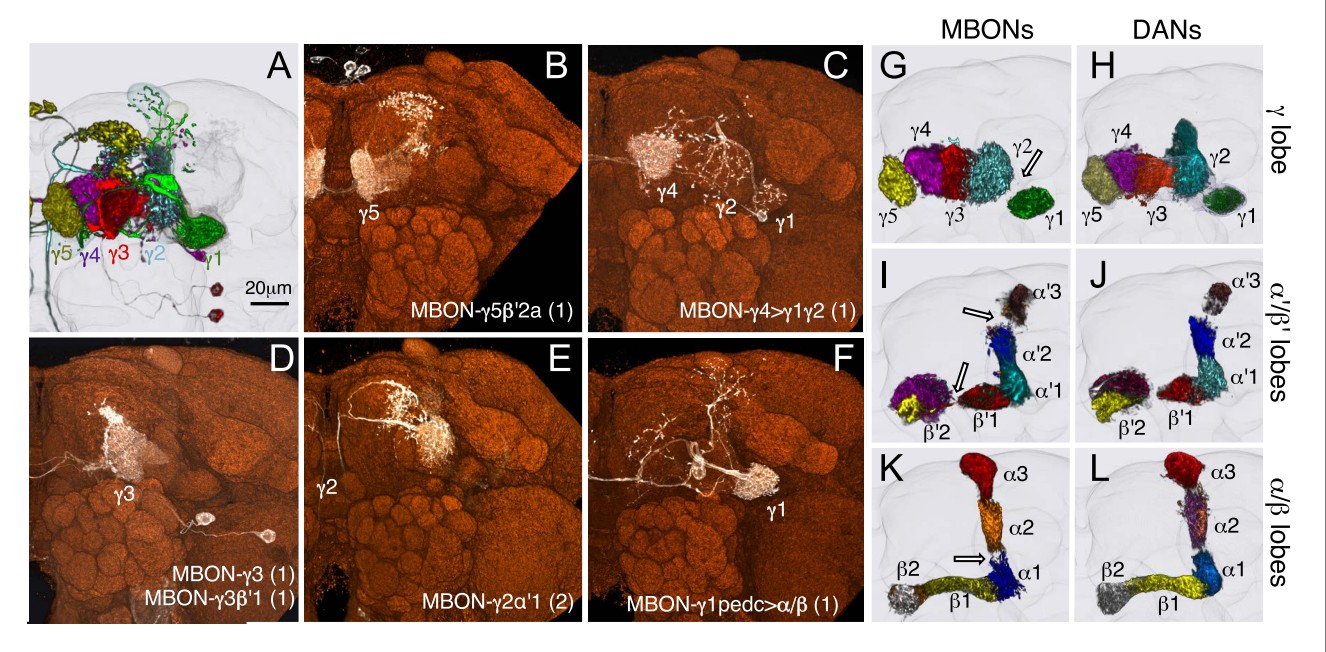

**Figure 10**. Compartmentalization of the MB lobes. (**A**–**F**) Tiling of MBON dendrites in the γ lobe. (**A**) A registered image of a brain hemisphere showing 5 MBON cell types that innervate contiguous compartments of the γ lobe. (**B**–**F**) Confocal images of brains showing expression in the MBONs shown in **A**; *pJFRC225-5xUAS-IVS-myr::smGFP-FLAG* in *VK00005* and the following split-GAL4 lines were used to generate the images: **B**, MB210B; **C**, MB298B; **D**, MB083C; **E**, MB077B; and **F**, MB112C. The cell types are indicated in the panels with the number of neurons of each type in parenthesis; the neurons are shown in white and the nc82 reference stain in orange. (**G**–**L**) The dendrites of MBONs and the axon terminals of DANs tile the MB lobes, defining 15 compartments. Dendrites of MBONs (**G**, **I**, and **K**) and axon terminals of DANs (**H**, **J**, and **L**), aligned to the standard brain, are shown for each lobe. The same false colors were assigned to the DANs and the MBONs of the same compartment. The arrows in (**G**, **I**, and **K**) show the four compartment borders where gaps were routinely seen between the MBON dendrites in adjacent compartments; these gaps correspond to areas of reduced synaptic density (see *Figure 10—figure supplement 1*). Note that the anterior layer of the β'2 compartment contains dendrites of MBON-γ5β'2a (**I**, yellow) and axon terminals of PAM-β2β'2a (**J**, yellow).

The following figure supplement is available for figure 10:

**Figure supplement 1**. Lower density of presynaptic sites at the border between compartments.

*Mao and Davis, 2009*), describe the neuronal architecture of the MB and provide insight into the logic of MB function.

Most of the cell types in the olfactory circuit—the MBONs and the DANs as well as the olfactory sensory neurons, the PNs, and the LH neurons (*Vosshall and Stocker, 2007*; *Masse et al., 2009*)—comprise small groups containing from one to 20 neurons (*Tanaka et al., 2004*; *Tanaka et al., 2008*, *2012*; *Yu et al., 2010*). Furthermore, neurons within these groups share highly stereotyped projection patterns that superimpose across different individuals (this work) (*Marin et al., 2002*; *Datta et al., 2008*). This suggests that the input and output connections of these neurons are genetically determined (*Ruta et al., 2010*; *Kohl et al., 2013*; *Fisek and Wilson, 2014*), although these may be subject to plasticity. In contrast, the KCs of the MB comprise 2000 neurons that define only seven distinct cell types, receive unstructured, rather than stereotyped, inputs (*Murthy et al., 2008*; *Caron et al., 2013*; *Gruntman and Turner, 2013*), and connect to multiple MBONs through plastic synapses (*Cassenaer and Laurent, 2007*, *2012*). These distinctive features of the KCs afford the MB the ability to contextualize novel sensory experiences, consistent with its role in mediating learned olfactory associations and behavior.

## Divergent input and convergent output of the MB

Neural representations of odor exist at multiple stations of the olfactory circuit. In the representation of odors by the antennal lobe projection neurons, every odor can be thought of as a point in an

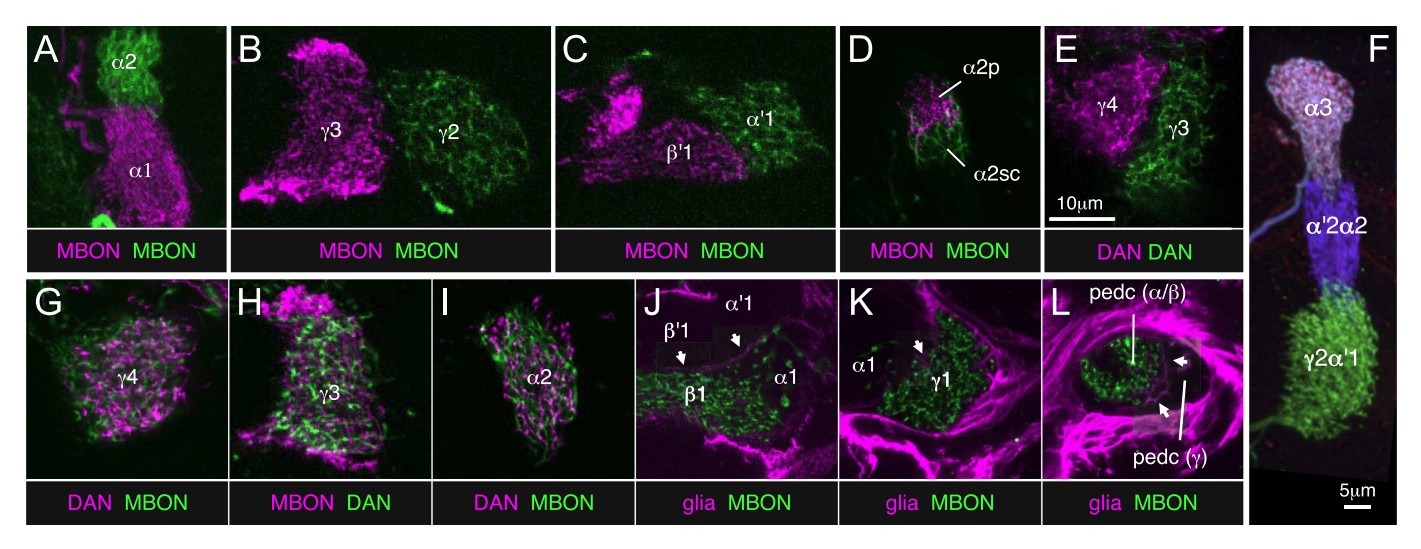

**Figure 11**. Two-color labeling experiments demonstrating compartmentalization of the MB lobes. (**A–E**) Two-color labeling of MBONs (**A–D**) or DANs (**E**) from adjacent compartments, or in the case of (**D**), subdivisions of the same compartment. Neurons were visualized by split-GAL4 and LexA drivers in combination with *pJFRC200-10XUAS-IVS-myr::smGFP-HA* in *attP18* and *pJFRC216-13XLexAop2-IVS-myr::smGFP-V5* in *su(Hw)attP8*, respectively. Substack projections of the compartments are shown. Clear segregation was observed between dendrites of MBONs or axon terminals of DANs from neighboring compartments. (**F**) MCFO labeling of a single brain showing the termini in the MB lobes of three types of PPL1 cluster DANs. The image was generated from line MB060B, which expresses in four types of PPL1 neurons. We were able to confirm all 11 compartment borders tested using either two-color labeling or MCFO experiments; we did not have the required genetic lines to test one of the 12 compartment borders (between β1 and β2). (**G–I**) Two-color labeling of DANs and MBONs from the same compartment. Single confocal slices are shown. The DANs and MBONs coextend, with each densely arborizing in the entire compartment. (**J–L**) Two-color labeling of ensheathing glia and MBONs. Whereas each of the three lobes (i.e., γ, α′/β′, and α/β) is separated clearly by ensheathing glia, we did not observe glia between MBON compartments within each lobe. Single confocal slices are shown. Arrows indicate the ensheathing glia separating the axon bundles of the γ, α′/β′, and α/β neurons in the lobes and pedunculus. The driver lines used are as follows: (**A**) MB310C (magenta), *R34B02-LexA* (green); (**B**) MB083C (magenta), *R25D01-LexA* (green); (**C**) MB083C (magenta), *R25D01-LexA* (green); (**D**) MB062C (magenta), *R34B02-LexA* (green); (**E**) MB316C (magenta), *R48B03-LexA* (green); (**G**) MB312C (magenta), *R53C03-LexA* (green); (**H**) MB083C (magenta), *R48B03-LexA* (green); (**I**) MB058B (magenta), *R34B02-LexA* (green); (**J**) MB434B (green), *R16D08-LexA* (magenta); (**K**) MB112C (green), *R16D08-LexA* (magenta); (**L**) MB112C (green), *R16D08-LexA* (magenta). Scale bar in (**E**) applies to all panels except (**F**).

approximately 50 dimensional space, each dimension corresponding to a particular glomerulus. The KC representation of odor identity has a dimension at least an order of magnitude larger, not only because there are many more KCs than antennal lobe glomeruli but also because the KCs mix projection neuron inputs nonlinearly (*Gruntman and Turner, 2013*). The dimension of the KC representation not only allows for a far greater capacity to respond appropriately to a large number of odors, it can also enhance performance in more complex decision-related tasks. Interestingly, the lack of structure in the input to the KCs, the small number of inputs, and the sparseness of their activity all appear to be tuned to maximize this dimension (*Perez-Orive et al., 2002*; *Huerta et al., 2004*; *Caron et al., 2013*; *Gruntman and Turner, 2013*) (Ann Kennedy, Columbia University Thesis). The fly has therefore evolved an olfactory circuit that exhibits structural and functional features predicted to optimize its ability to contextualize and respond appropriately to a rich array of olfactory experiences.

The convergence of a large number of KCs onto a small number of MBONs indicates that the dimension of the MBON representation is significantly smaller than that of the KCs. The dimension of the MBON can be no greater than their number (34) and is likely to be considerably smaller because there are only 21 different MBON cell types. Thus, rather than providing a general representation of odor identity, the activity of the individual MBONs is more likely to encode a set of 'state variables' that collectively bias behavioral responses to sensory stimuli. This bias is likely to reflect the combined effects of external experiences and the internal state of the fly (see accompanying paper) (*Aso et al., 2014*, *Krashes et al., 2009*; *Bracker et al., 2013*). Consistent with this view, the odor responses of individual MBONs differ between flies and these differences appear to depend on plasticity (Hige et al., unpublished).

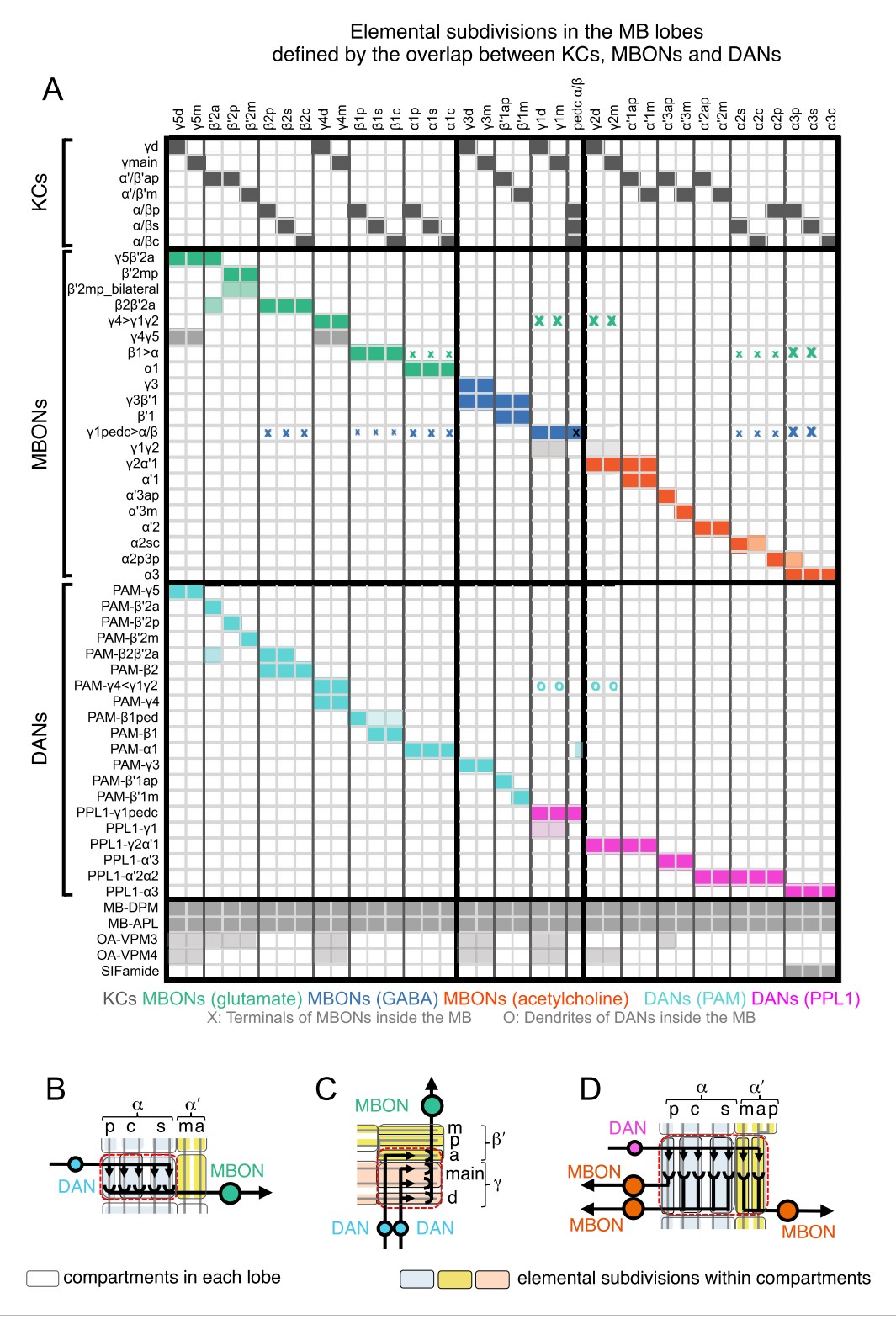

**Figure 12**. The arborization patterns of individual cell types within the MB lobes. (**A**) A matrix summarizing the projection patterns in the MB lobes of the KC axons, the MBONs, the DANs, and other modulatory neurons. The 15 lobe compartments and the core of distal peduculus (pedc) are separated with thick vertical lines, and each compartment is further divided into columns representing 'synaptic units' (***Tanaka et al., 2008***), elemental

*Figure 12. Continued on next page*

*Figure 12. Continued*

subdivisions that are the smallest regions in the MB lobes where a specific set of MBONs, DANs, and KC types overlap. For example, the β1 compartment can be divided into posterior (β1p), surface (β1s), and core (β1c) subdivisions containing the axons of the α/βp, α/βs, and α/βc KCs, respectively. The processes of some MBONs and DANs do not arborize in an entire compartment, but only in a subset of its elemental subdivisions. *Tanaka et al. (2008)* first described this anatomical feature and called such regions 'synaptic units'. An example is shown in *Figure 11D*, which shows a two-color labeling experiment illustrating the subdivision of the α2 compartment by the dendrites of the two MBONs, one of which arborizes in the domain of the posterior KCs (α2p) and the other in the domain of the surface and core KCs (α2sc); this arrangement is diagrammed in Panel **D**. The heavy vertical lines divide subdivisions into three groups representing different MBON neurotransmitter types (also indicated by the colors filling the cells of the matrix). The rows of the matrix correspond to the cell types of KCs, MBONs, and DANs. See *Table 1* for synonyms for cell type names and references. The colors of the cells of the matrix indicate the putative transmitter for MBONs and cluster of origin for DANs. The color correspondence is given below the matrix. The two MBON types of unknown neurotransmitter type are indicated in light gray. The fully-colored cells of the matrix represent processes of neurons (dendrites for MBONs and synaptic terminals for DANs) that are uniformly and densely distributed in that subdivision; fainter colors represent subdivisions that are innervated only sparsely. In three cases, output neurons send axon terminals back into the MB. Such cases are represented by an 'X' with the size representing the density of terminals in that subdivision. One DAN cell type (PAM-γ4<γ1γ2) has dendrites within (as well as outside) the lobes in the subdivisions indicated by an 'O' (see *Figure 14—figure supplement 1E*). The projection patterns of the GABAergic MB-APL and serotonergic MB-DPM are diagramed; these neurons are MB intrinsic neurons that broadly innervate the MB (*Figure 3—figure supplement 1B,C*) (*Waddell et al., 2000*; *Tanaka et al., 2008*). Also diagrammed are the projection patterns of the octopaminergic OA-VPM3 and OA-VPM4 (*Figure 3—figure supplement 1E*) (*Busch et al., 2009*; *Busch and Tanimoto, 2010*) and SIFamide peptidergic neurons (*Figure 3—figure supplement 1F*) (*Verleyen et al., 2004*) within the lobes; these neurons have only a small fraction of their terminals within the MB (*Figure 2—figure supplement 6*) and these sparsely innervate only a subset of the compartments. The following abbreviations are used in the names of the subdivisions: a, anterior; m, middle; p, posterior; d, dorsal; s, surface; c, core; and pedc, pedunculus core. (**B**–**D**) Diagrams illustrating different circuit motifs found in the MB lobes. (**B**) The terminals of a single DAN type (PAM-α1) and the dendrites of a single MBON type (MBON-α1) occupy a compartment. 16 of the 21 MBON types, like MBON-α1, arborize dendrites within just one of the three lobes (i.e., γ, α′/β′, and α/β), indicating that these MBONs receive inputs from only one of the three KC classes. (**C**) Two DANs (PAM-γ5 and PAM-β′2a) innervate the region occupied by one MBON (MBON-γ5β′2a). One DAN fills the γ5 compartment and the other only innervates the anterior elemental subdivision of the β′2 compartment (β′2a); a single MBON has inputs from the areas defined by both DANs. Four MBON types, like MBON-γ5β′2a, extend dendrites spanning lobe boundaries. (**D**) A single DAN (PPL1-α′2α2) innervate two compartments (the α′2 and α2), an area covered by three MBON types (MBON-α′2, -α2sc, and -α2p3p). Eight MBON types, like MBON-α2sc and MBON-α2p3p, arborize dendrites further confined to elemental subdivision(s) within a compartment. This suggests that these MBONs receive input exclusively from subtypes of KCs; for example, MBON-α2p3p receives input from α/βp KCs, which presumably carry non-olfactory information (see *Figure 7A*). Nearly all DANs have their termini confined to a single compartment; however, we identified 3 DAN types that have axon terminals in two compartments.

It is tempting to associate individual MBONs with specific behaviors, but the ultimate bias in behavioral response may be represented across the full set of MBONs as a population code. Thus, the high-dimensional representation of odor identity in the KCs may be transformed into a low-dimensional representation that dictates behavioral bias. Just as an individual odor is represented by an ensemble of KCs, a given behavioral bias is likely to be represented by an ensemble of MBONs. In accord with this view, activation or inactivation of combinations of MBON cell types results in more robust effects on behavior than those observed with individual MBONs (see the accompanying paper) (*Aso et al., 2014*).

## Reactive DANs and predictive MBONs

We have shown that the MB lobes are divided into compartments that receive input from the KCs and specific DANs and transmit information to a small number of MBONs. Each compartment therefore receives specific dopaminergic input capable of modifying the synapses between the KCs and specific MBONs. These compartments may reflect basic computational units of the MB. The specificity of the dopamine input and its ability to direct learning may therefore transform the unstructured KC representation of odor to an ordered MBON representation encoding behavioral bias. Specific subpopulations

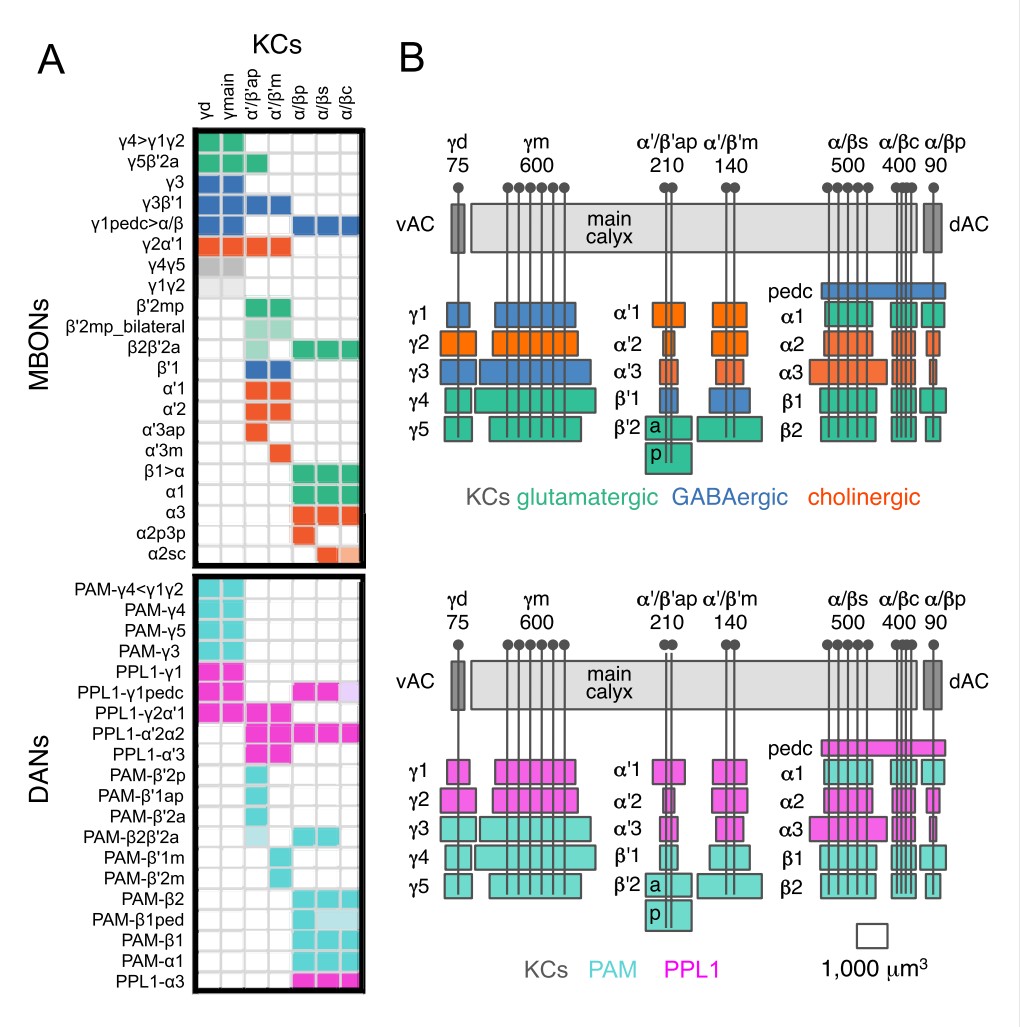

**Figure 13**. Each KC type transmits information to multiple compartments. (**A**) A simplified version of the matrix shown in *Figure 12A* that illustrates the contacts between the individual MBON (top) and DAN cell types (bottom) with the seven types of KCs (columns). Colors of the cells in the matrix indicate neurotransmitter type for MBONs and cluster of origin for DANs. The two MBON types of unknown neurotransmitter type are indicated in light gray. Interestingly, each KC likely synapses with MBONs of all three neurotransmitter types and receives modulatory DAN input from both PAM and PPL1 clusters. (**B**) Diagrams of the MB lobes. The axons of the seven types of KCs are shown as straight vertical lines without branches and the boxes represent each of the 37 elemental subdivisions defined in *Figure 12A*. The number of KCs of each type, based on cell counting (see 'Materials and methods'), is indicated. The size of the box representing each subdivision and calyces indicates its volume, as determined by measurements performed on confocal stacks. The volumes of lobes are not simply proportional to the number of KCs they contain and are not uniform along their lengths, presumably reflecting differences in synaptic density. The convergence ratio from KCs to single MBONs could range from as high as ~2300:1 (for MBON-β2β'2a that arborizes in β2 and β'2a of both hemispheres) to ~90:1 (for MBON-α2p3p), assuming that each KC forms synapses in each elemental subdivision. In the top diagram, the color of the box represents the neurotransmitter used by the MBONs that have dendrites in that subdivision. In the bottom diagram, the color of the boxes represents the cluster of origin of the DANs innervating that subdivision. The following abbreviations are used in the names of the elemental subdivisions: a, anterior; m, middle; p, posterior; d, dorsal; s, surface; c, core; and pedc, pedunculus core.

of DANs may react to different features in the external and internal world (*Mao and Davis, 2009*; *Liu et al., 2012*; *Tomchik, 2013*; *Das et al., 2014*). DAN activity may modify the activity of MBONs through learning to provide a representation that is predictive of the implications of an olfactory stimulus. Learning, in this view (*Sutton and Barto, 1998*), is a transition from a reactive DAN representation to a predictive MBON representation.

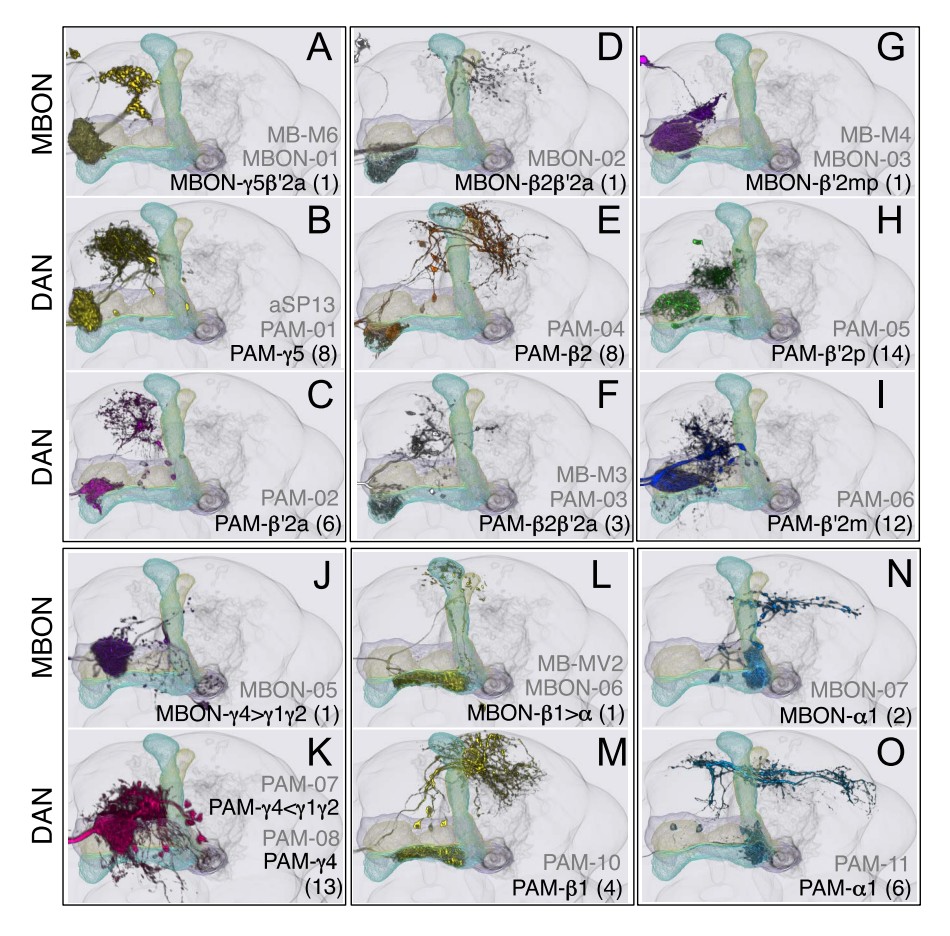

**Figure 14**. Compartments with dendrites of glutamatergic MBONs. Representative images of neurons that have been aligned to the standard brain are shown. Glutamatergic MBONs and DANs that project to the same compartments in the lobes are displayed together. The names of cell types are given in a standard format that includes information about the subdivisions innervated. Short names based on simple numbering, as well the original name in cases where the cell type has been previously described at the single cell level, are also shown (gray font). The number of cells per cell type found in each brain hemisphere is shown in parentheses. Some cell types were not separated by split intersections and images show mixtures of cell types in these cases. Except for PAM-γ4<γ1γ2 and PPL1-γ1, DANs have dendritic branches in the ipsilateral hemisphere and axons that bilaterally innervate the same MB compartments in both hemispheres. The distribution of neurites outside the MB lobes is shown in more detail in **Figure 18—figure supplement 1**. The split-GAL4 drivers for each cell type are listed in **Table 1** and **Supplementary File 1**. (**A**) The dendrites of MBON-γ5β′2a arborize in the contralateral γ5 and β′2a. The major axon of MBON-γ5β′2a projects ipsilaterally to the superior medial protocerebrum (SMP), whereas a very thin axon projects to the CRE and SMP in the other hemisphere (see **Figure 14—figure supplement 1A**). (**B**) PAM-γ5 neurons. Dendrites of these neuron arborize in the same regions of SMP where the MBON-γ5β′2a neurons terminate, suggesting a possible recurrent loop (see text and **Figure 20C**). (**C**) PAM-β′2a neurons. (**D**) The dendrites of MBON-β2β′2a bilaterally arborize in the β2 compartment and β′2a subdivision and its axon projects ipsilaterally to the superior intermediate protocerebrum (SIP) and superior lateral protocerebrum (SLP) (see also **Figure 14—figure supplement 1B**). (**E**) PAM-β2 neurons. (**F**) PAM-β2β′2a neurons have sparse terminals in the anterior layer of β′2a and even sparser terminals in the core layer of β2. The dendrites of PAM-β2β′2a and PAM-β2 are spatially segregated, suggesting that distinct upstream neurons regulate their activity. (**G**) MBON-β′2mp arborizes in the contralateral β′2mp; its main axon projects to the CRE and SMP on the same side and its minor axon to the ipsilateral side (**Figure 14—figure supplement 1C**). A second output neuron, MBON-β′2mp-bilateral, sparsely arborizes its dendrites in the β′2mp compartment and projects dense axons bilaterally (see **Figure 14—figure supplement 1D**). (**H**) PAM-β′2p neurons. (**I**) PAM-β′2m neurons. (**J**) The MBON-γ4>γ1γ2 dendrites arborize in the contralateral γ4 and its axon projects both within the lobes to γ1 and γ2 and to regions outside the lobes. (**K**) PAM-γ4 neurons and PAM-γ4<γ1γ2 neurons. The image shows a mixture of two cell types; PAM-γ4<γ1γ2 is

*Figure 14. Continued on next page*

*Figure 14. Continued*

unusual, in which it has some of its dendrites within the MB (in the γ1 and γ2 compartments; see *Figure 14—figure supplement 1E*). (**L**) The MBON-β1>α dendrites arborize in the contralateral β1 and its axon innervates the α1, α2, and α3 compartments within the lobes as well as areas outside the lobes (see *Figure 14—figure supplement 1F*); the terminals in α3 are concentrated in the surface and posterior layers. (**M**) PAM-β1 neurons (bottom). The posterior layer of β1 is more densely innervated by a second PAM cluster cell type, PAM-β1ped, that also projects to the posterior end of the pedunculus (see *Figure 14—figure supplement 1G*). (**N**) The dendrites of MBON-α1 arborize in α1 and its axons project to the posterior SIP and SLP. Two neurons with identical morphology are present in each hemisphere. Although we observed this cell type in MCFO analysis of the dVGlut-GAL4 line OK371 (*Mahr and Aberle, 2006*), its terminals showed much weaker immunoreactivity to the anti-dVGluT than the other putative glutamatergic neurons. (**O**) PAM-α1 neurons (bottom) have terminals that extend slightly outside the area arborized by the dendrites of MBON-α1 to the distal end of the pedunculus.

The following figure supplement is available for figure 14:

**Figure supplement 1**. Neurons of the glutamatergic compartments.

## Feedforward and feedback processing in the MBON/DAN network

The dendritic and axonal projection patterns of the MBONs and DANs suggest feedforward and feedback circuit motifs in the MB lobes. The interactions among the different MBON types may form a

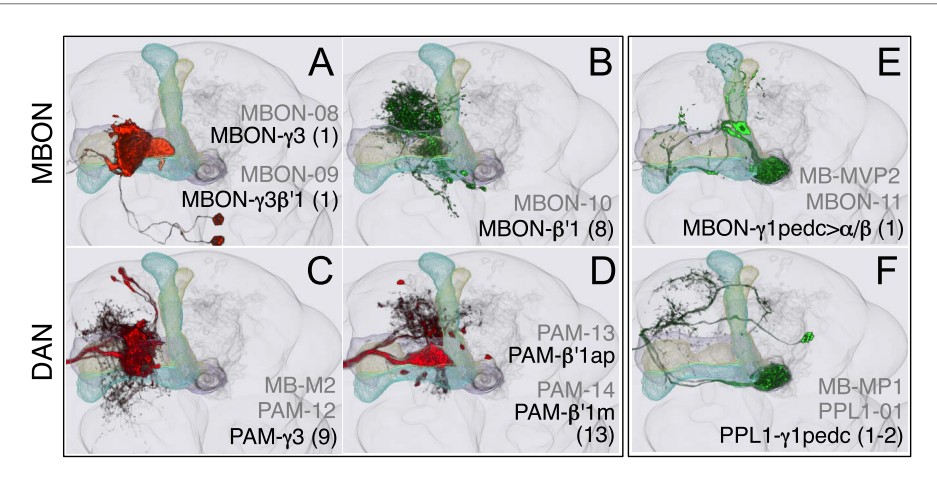

**Figure 15**. Compartments with dendrites of GABAergic MBONs. (**A**) MBON-γ3 and MBON-γ3β'1; the morphologies of these cell types are described in more detail in *Figure 8D,E*. (**B**) MBON-β'1 (top right); this MBON type is unusual in a number of ways. First, there are, on average, 8 cells per hemisphere (i.e., the number of cells fluctuate between 7–9 in MB057B) compared with one or two for the other MBON cell types. Second, they are the only MBONs from the lobes whose dendrites arborize, in addition to the lobes, in neighboring neuropils (CRE and SMP). Since CRE and SMP contain zones where the terminals of other MBONs converge (see below), these neurons may sum the outputs from a number of MB compartments. Finally, this is the only MBON cell type that projects to the lateral accessory lobe, an output region of the central complex. (**C–D**) Three types of PAM cluster DANs innervate the γ3 and β'1 compartments: PAM-γ3 neurons (**C**); PAM-β'1ap neurons; and PAM-β'1m neurons (**D**). (**E**) The dendrites of MBON-γ1pedc>α/β (top) arborize in the ipsilateral γ1 and the core of the pedunculus, where the axons of the α/β KCs are found. Its axon projects bilaterally to the α/β lobes and contralaterally to the core of the pedunculus and, to a lesser extent, outside the MB lobes; in α3, its terminals are enriched in the surface layer (similar to those of MBON-β1>α, see *Figure 17E*). (**F**) PPL1-γ1pedc; a DAN of the PPL1 cluster with terminals that overlap with the dendrites of MBON-γ1pedc>α/β. One additional PPL1 cluster DAN innervates γ1 sparsely (PPL1-γ1; not shown, see *Table 1*).

The following figure supplement is available for figure 15:

**Figure supplement 1**. The MBON-γ1pedc>α/β and PPL1-γ1pedc innervates the γ1 lobe compartment as well as the core of the distal pedunculus.

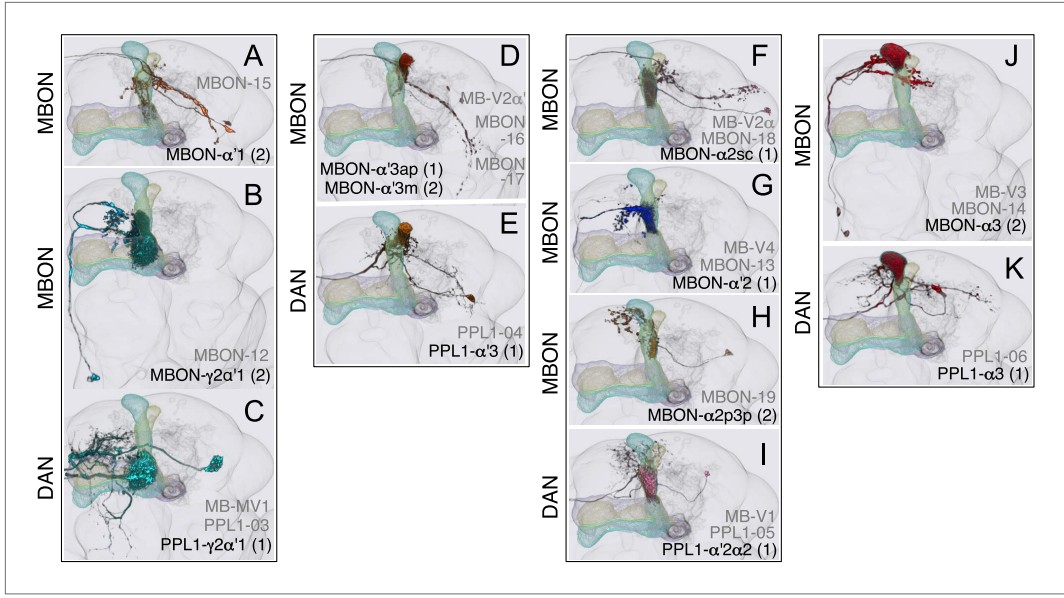

**Figure 16**. Compartments with dendrites of cholinergic MBONs. (**A**) The dendrites of the two MBON-α′1 cells arborize unilaterally in α′1. Their thin dendritic branches project through the edge of the α′2 and α′3 compartments. The MBON-α′1 neurons share an axon tract with MBON-α′3ap and MBON-α′3m and terminate in similar region of SIP, SLP, and LH (see panel **D**). (**B**) MBON-γ2α′1. This cell type consists of two morphologically identical cells (see **Figure 8B**) that project to the CRE and SMP. (**C**) PPL1-γ2α′1 neuron. (**D**) The dendrites of MBON-α′3ap and MBON-α′3m arborize in the ipsilateral α′3m or α′3ap, respectively, and their axons project bilaterally to the same regions of the LH, SIP, and SLP. The axonal branches of the MBON-α′3m stop at the LH, whereas the MBON-α3′ap axons extend to a region ventral to the LH. (**E**) PPL1-α′3 neuron. (**F**) The dendrites of MBON-α2sc arborize unilaterally in α2sc, and its axon projects bilaterally to the SIP, the SLP, and the dorsal LH. (**G**) MBON-α′2 projects to the CRE and SMP. A few terminals are also found in SIP. (**H**) The two MBON-α2p3p cells arborize their dendrites unilaterally in the posterior layer of the α2 and α3 compartments (α2p and α3p) and project to the SMP. The KCs in the posterior layer of the lobes (the α/βp neurons) have their dendrites in the dorsal accessory calyx, a region spatially segregated from the dendritic arbors of other KCs (see **Figure 7A**), and their activity is decreased in response to odors (**Perisse et al., 2013**), suggesting a distinct role for these output neurons. (**I**) PPL1-α′2α2 neuron. (**J**) Two morphologically identical MBON-α3 neurons have dendrites in α3 and terminals in the SMP, SIP, and SLP. (**K**) PPL1-α3 neuron.

multi-layer feedforward readout network (**Figure 17**). Processing through such a network significantly expands the computational capacity of a readout system, which can be valuable for more complex learning strategies. Consider a learning scenario in which a fly associates an odor with a strongly aversive US. A conditioned aversive response to the initial odor could occur through plastic changes affecting an ensemble of output neurons, and a low response threshold could allow these responses to generalize to related odors. Upon further experience, some odors within this category might be identified as 'safe' or even appetitive. Ethologically important exceptions could be learned if one of the neurons that interconnect compartments (MBON-γ4>γ1γ2, MBON-γ1pedc>α/β, and MBON-β1>α) became responsive to a 'safe' odor and inhibited the original trained ensemble of aversive MBONs. Thus, the layered MBON network provides an efficient mechanism for modifying and updating previous learning.

Interestingly, the dendrites of the DANs overlap with MBON axons in four of the five MBON projection zones (**Figures 19 and 20**). Thus, MBONs may modify the activity of the DANs that modulate their own activity and plasticity, resulting in a recurrent loop. This could provide positive or negative feedback to a specific compartment. Positive feedback might enhance learning to particularly salient stimuli, whereas negative feedback might suppress dopamine release once the correct response has been learned. These recurrent connections may also allow output from one compartment to modulate learning in other compartments, as some MBONs appear to target DANs that innervate non-cognate compartments.

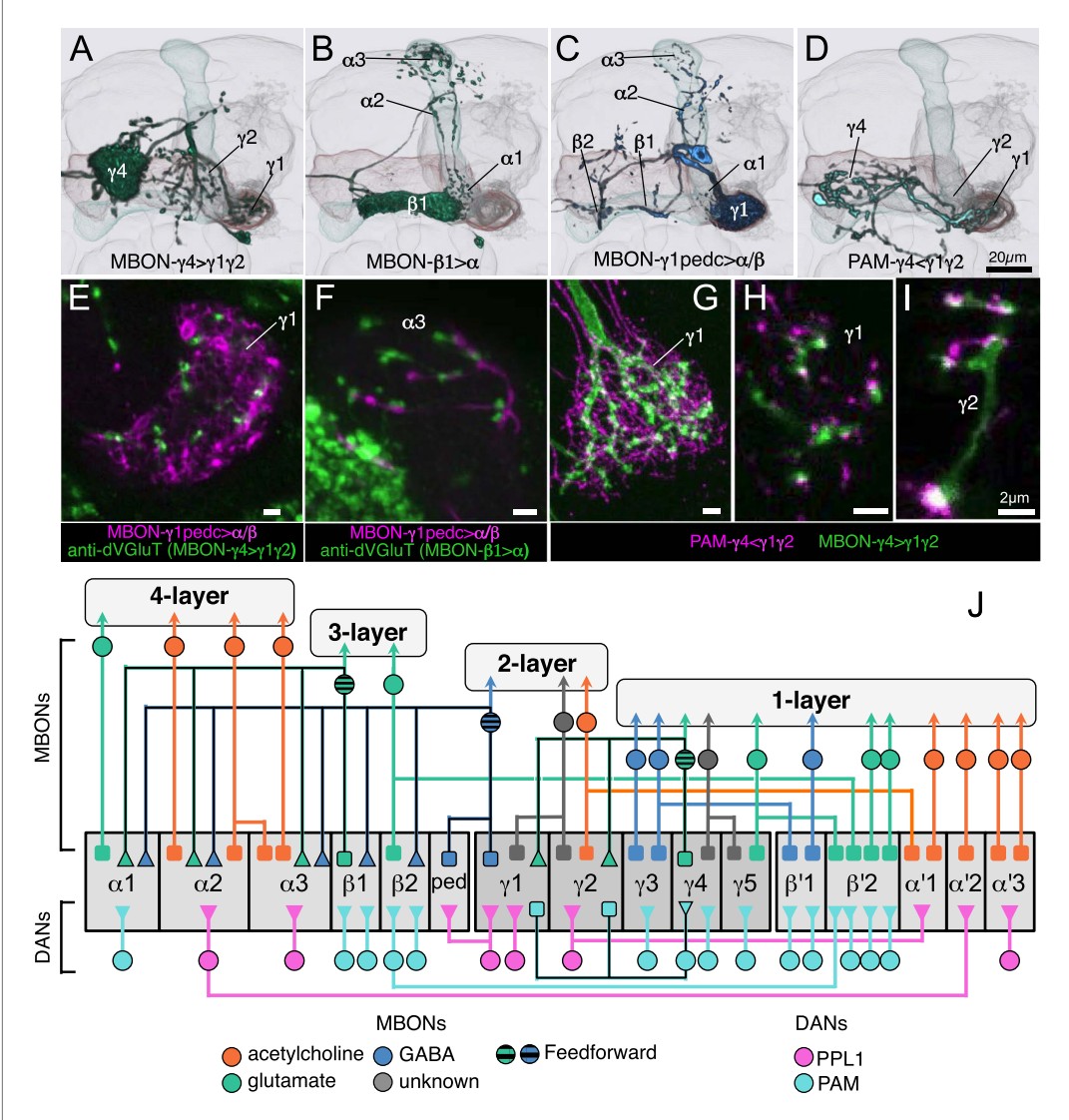

**Figure 17**. Three types of MBONs and one type of DAN interconnect multiple compartments. (**A–C**) Registered images of the three MBON cell types whose axons project to other compartments within the MB lobes. (**A**) A glutamatergic output (MBON-γ4>γ1γ2) from γ4 sends axons back into the lobes that terminate in γ1 and γ2. (**B**) A glutamatergic output from β1 (MBON-β1>α) terminates in α1, α2, and α3. (**C**) A GABAergic output from γ1 and the core of distal peduncle (MBON-γ1pedc>α/β) terminates in the α/β lobes in both hemispheres and in the peduncle, where α/β KCs bifurcate, in the contralateral side of the brain. (**D**) A DAN cell type (PAM-γ4<γ1γ2) that has dendrites in γ1 and γ2 and terminals in γ4. This cell type provides potential feedback signals to the MBON-γ4>γ1γ2 (see panels **G–I**). (**E**) Two-color labeling of the GABAergic output neuron shown in panel C (MB112C, *pJFRC225-5XUAS-IVS-myr::smGFP-FLAG* in *VK00005*) and an antibody against vesicular glutamate transporter. The dendrites of MBON-γ1pedc>α/β overlap with glutamatergic terminals in γ1. The likely source of these glutamatergic terminals is MBON-γ4>γ1γ2 shown in panel **A**. (**F**) The GABAergic terminals of MBON-γ1pedc>α/β and glutamatergic terminals presumably of MBON-β1>α, the neuron shown in panel **B**, are in close proximity in the surface layer of α3. The green signal in the lower left corner of the panel is from neurons outside the MB. This overlap of glutamatergic and GABAergic feedforward projections suggests that they may cooperatively or competitively regulate the excitatory output of MBON-α3. (**G**) A substack projection showing two-color labeling of the dendrites of the only DAN (MB312C) that has dendrites in the MB lobes and axon terminals of an MBON in the γ1 compartment (*R53C03-LexA*). (**H** and **I**) Single confocal slices in the γ1 and γ2 compartment showing two-color labeling of the cell types shown in (**G**). (**J**) Schematic of the circuits within the MB lobes. DAN inputs (bottom) and MBON outputs (top) from 15 MB lobe compartments and the core of the distal peduncle (ped) are shown

*Figure 17. Continued on next page*

*Figure 17. Continued*

(gray rectangles, middle). Colors indicate neurotransmitter types for MBONs and cluster of origin for DANs as indicated. Dendrites are represented as squares and presynaptic terminals as triangles. Three MBONs, indicated by the heavier outline and striped cell bodies, send axonal terminals (triangles) back into the MB lobes creating a 4-layer feedforward network. See text for details. Eight types of MBONs receive input from more than one of the 15 compartments in the lobes (or a compartment plus the ped) and in five cases those compartments reside in different lobes. As different functions of learning and memory, such as acquisition and retrieval, have been attributed to different KC classes (*Isabel et al., 2004*; *Krashes et al., 2007*), MBONs integrating across lobes may function in coordinating different phases of learning and memory. MBON-γ1pedc>α/β and PPL1-γ1pedc innervate the core of the distal pedunculus (ped) as well as the γ1 compartment. Three DAN cell types innervate multiple lobe compartments.

## Convergence zones of the MB outputs

The axonal terminals of the MBONs are largely confined to five discrete neuropils in the brain, the SIP, SMP, CRE, SLP, and the LH (*Figures 18 and 19*). The CRE, SMP, SIP, and SLP may be sites of convergence for all of the signals relevant for classical conditioning. These convergence zones are major sites of arborization for the dendrites of the DANs and axons of the MBONs. Since DANs are activated in response to an aversive or appetitive US, these neuropils are also likely to receive input from neurons transmitting information about the nature of the US. A US, by definition, elicits an innate behavioral response consistent with its valence, suggesting that the outputs from these neuropils may convey motor commands. Interestingly, dendrites of neurons projecting to the fan-shaped body in the central complex, a brain region coordinating motor actions (*Strauss, 2002*), arborize in these neuropils (*Hanesch et al., 1989*; *Young and Armstrong, 2010*). We therefore postulate an evolutionary primitive circuit in the MBON convergence zones, in which US inputs activate motor command neurons to elicit innate responses. Evolution may have built upon this simple reflex circuit, incorporating pathways from the MB that generate learned behaviors in response to a CS (*Figure 21*).

## Conclusion

The vertebrate brain consists of interconnected structures comprised of large collections of equivalent neurons whose number often increases with evolutionary complexity. This is in sharp contrast to most brain structures in invertebrates that consist of small number of neurons with stereotyped projections that suggest determined connections. The MB represents an exception. The MB lobes are formed by the axons of a large number of equivalent neurons, the KCs, and as with vertebrate cortical neurons, the number of KCs increases in species with more complex behaviors (*Strausfeld, 2012*). Moreover, the inputs to the KCs from olfactory projection neurons are not determined or stereotyped but appear random. Thus, the MB diverges from the highly ordered neural architecture typical of the invertebrate nervous system. The MB may therefore represent an evolutionary primitive brain structure homologous in form and function to structures in the vertebrate brain (*Schurmann, 1974*; *Laurent, 2002*; *Tomer et al., 2010*; *Farris, 2011*). The elucidation of the inputs and outputs of the MB may now permit an understanding of how learning links an abstract representation to a specific behavior, and this may provide insight into higher associative functions in both invertebrate and vertebrate brains.

# Materials and methods

## Molecular and genetic methods

Split-GAL4 and LexA transgenes used enhancers, selected based on GAL4-line expression patterns (*Jenett et al., 2012*), and were constructed as previously described (*Pfeiffer et al., 2010*). VT999036 was from the Vienna Tiles collection and a gift of Barry Dickson. Promoter regions corresponding to the following GAL4 constructs were amplified by PCR: *TH-GAL4* (*Friggi-Grelin et al., 2003*), *Ddc-GAL4* (*Li et al., 2000*), *HL9-GAL4* (*Claridge-Chang et al., 2009*), and *Tdc2-GAL4* (*Cole et al., 2005*). All fragments were amplified from genomic DNA except for the upstream region of *HL9*, which was amplified from the *HL9-GAL4* plasmid in order to conserve the mutated exon B start site. 5'-*Xba*I and 3'-*Fse*I sites were added to the fragments upstream of GAL4. Downstream fragments were amplified with added 5'-*Spe*I (TH) or *Nhe*I (Tdc2, Ddc and HL9) sites and 3'-*Not*I sites. These fragments were then cloned into the corresponding sites on pBPp65ADZpUw and pBPZpGAL4DBDUw vectors

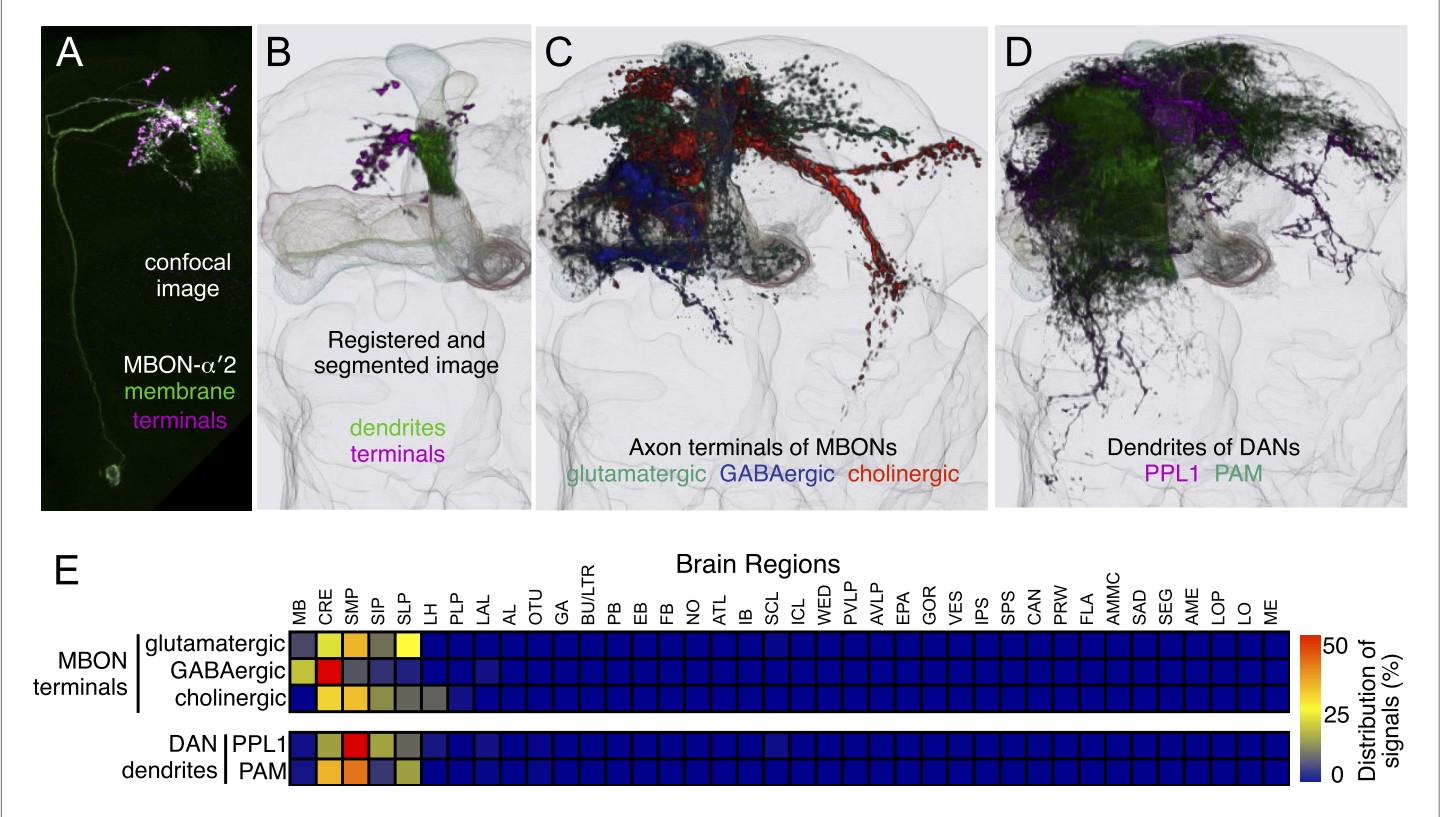

**Figure 18**. Projection patterns of the MBON axons and DAN dendrites outside the MB lobes. (**A**) To analyze the projection patterns of the MBONs outside the MB lobes, we first segmented their axon terminals based on the localization of a presynaptic marker. A maximum intensity projection of a confocal stack showing MBON-α'2 neurons labeled with a presynaptic reporter (magenta) and a general membrane marker (green). The split-GAL4 line MB018B was used to drive the expression of two constructs: a reporter targeted to membranes (*pJFRC225-5XUAS-IVS-myr::smGFP-FLAG* in *VK00005*) and a reporter targeted to presynapses (*pJFRC51-3XUAS-IVS-Syt::smGFP-HA* in *su(Hw)attP1*). (**B**) An image of the segmented terminals and dendrites of a MBON-α'2 neuron aligned to the standard brain. After registration of the image, synaptic terminals were segmented based on the preferential labeling by Syt::smGFP-HA. Dendrites were segmented based on their morphology and localization within the MB lobe. (**C**) A registered image of the segmented presynaptic terminals of MBONs showing the areas of the brain innervated by MBONs. The segmented terminals were false-colored based on their putative neurotransmitter. (**D**) A registered image of the segmented dendrites of DANs. The segmented dendrites were false-colored based on their cluster of origin. (**E**) Distribution of MBON terminals and DAN dendrites as quantified in each neuropil found in the adult brain (see ***Ito et al., 2014*** for the location and abbreviation for each neuropil [***Ito et al., 2014***]). After normalizing the signal intensity from different cell types, we separately summed signals from the terminals of glutamatergic, GABAergic, or cholinergic MBONs and dendrites of PPL1 and PAM cluster DANs. 99% of the MBON terminals are distributed in the MB lobes and five neuropils (CRE, SMP, SIP, SLP, and LH). Colors represent the percent of the processes found in each of 38 brain regions. See 'Materials and methods' for a description of the quantification method.

The following figure supplement is available for figure 18:

**Figure supplement 1**. Projection patterns of the individual MBON and DAN cell types.

(***Pfeiffer et al., 2010***) that had been modified to add new restriction sites as follows: the p65ADZp and ZpGAL4DBD segments of these vectors were amplified with the addition of a 5'-*Xba*I and a 3'-*Avr*II site and then cloned into pBDP (***Pfeiffer et al., 2008***) at 5'-*EcoR*I and 3'-*Not*I. Downstream fragments were cloned into the modified AD and DBD vectors at *Avr*II *Not*I (TH) or *Nhe*I *Not*I (Tdc2, Ddc and HL9) sites. To generate *Trh-p65ADZp* and *Trh-ZpGAL4DBD*, the *Trh* promoter region was amplified from genomic DNA using primers SM(1A) and BI(1S) (***Alekseyenko et al., 2010***) and cloned into pBPp65ADZpw and pBPZpGAL4DBDw using the Gateway system as previously described (***Pfeiffer et al., 2008***, ***2010***).

 *pJFRC225-5xUAS-IVS-myr::smGFP-FLAG* in *VK00005* and *pJFRC200-10xUAS-IVS-myr::smGFP-HA* in *attP18* are described by Viswanathan et al. (unpublished); smGFP is a non-fluorescent, mutated GFP fused with multiple copies of an epitope tag (either HA, V5, or FLAG) for immunolabeling with

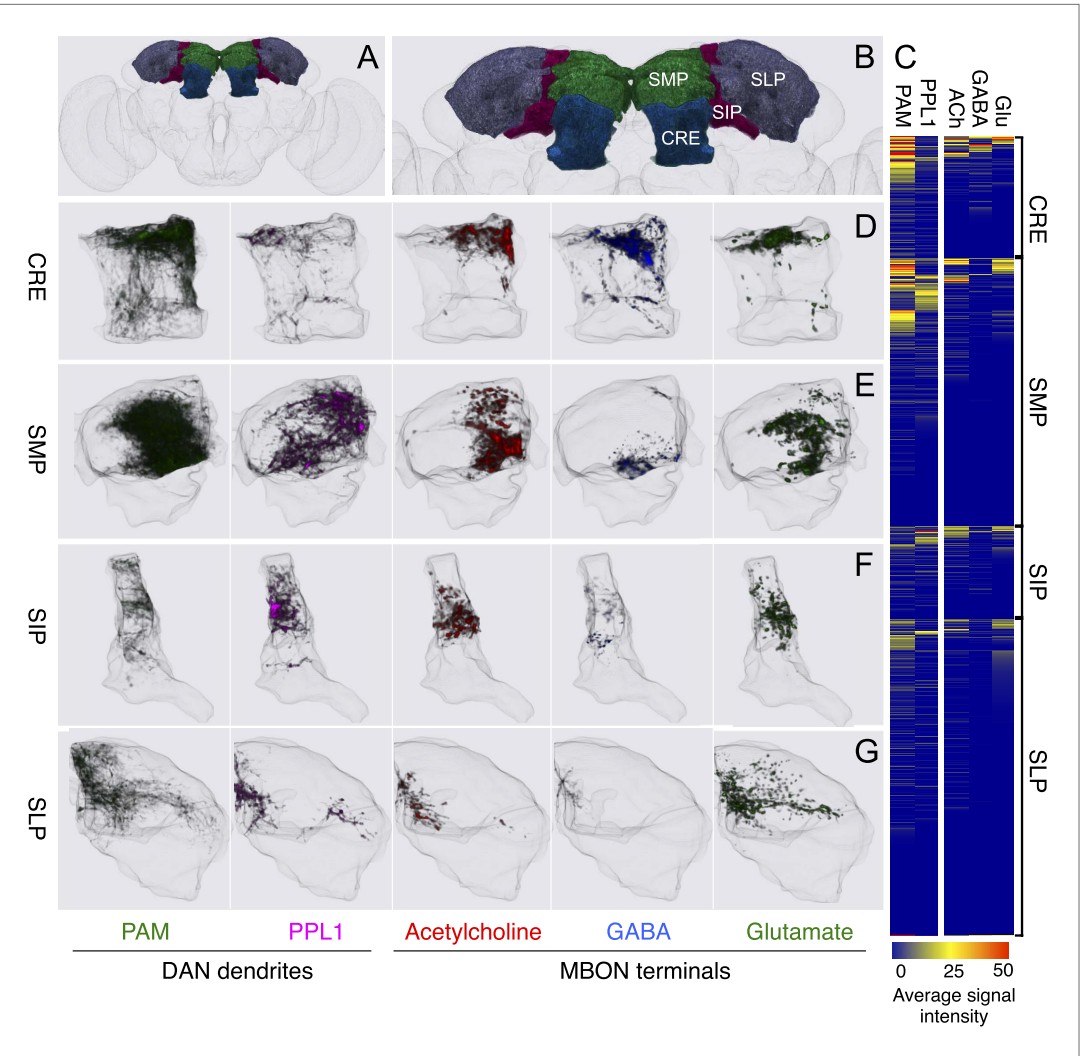

**Figure 19**. Distribution of MBON terminals and DAN dendrites in four brain areas. (**A**) Schematic of the brain highlighting the areas that exhibit the highest density of MBON terminals and DAN dendrites. (**B**) A magnified view of a portion of the brain shown in panel **A** with the CRE, SMP, SIP, and SLP neuropils indicated. (**C**) Distribution of MBON terminals and DAN dendrites showing their further clustering within the CRE, SMP, SIP, and SLP. Heat map representations of neurite density were computed separately for PAM and PPL1 cluster DAN dendrites and for cholinergic (ACh), GABAergic, and glutamatergic (Glu) MBON terminals. Each horizontal row shows mean intensities in a 10 × 10 × 10 voxel (3.8 × 3.8 × 3.8 μm) cube for each of the indicated neuronal types (columns), represented in an 8-bit scale (0–255). The rows have been sorted based on the sum of the intensity in each volume for all neuronal types. (**D**–**G**) Distributions of MBON terminals and DAN dendrites in the CRE (**D**), SMP (**E**), SIP (**F**), and SLP (**G**) are shown.

various fluorescent dyes. *pJFRC51-3xUAS-Syt::smGFP-HA* in *su(Hw)attP1* and *pJFRC216-13xLexAop-myr::smGFP-V5* in *su(Hw)attP8* were generated by standard methods using vectors described in *Pfeiffer et al. (2010)*. Using a Syt::smGFP-HA construct with only three copies of the UAS sequence was necessary to decrease expression to a level that generated a >5-fold enrichment of signal in presynaptic boutons relative to other cellular subdomains. *UAS-nuclearLacZ* (UAS-nlsLacZ) for cell counting was previously described (*Baker et al., 1996*).

Multicolor flp-out (MCFO) is a stochastic method that labels individual cells in different colors using a set of three UAS-STOP-epitope constructs that each expresses a different epitope when the STOP cassette is removed. The STOP cassettes in these constructs are each flanked by FRT sites that are removed in a stochastic way by limited expression of flp recombinase (*Struhl and Basler, 1993*). Reagents for MCFO are described in Nern et al., in preparation.

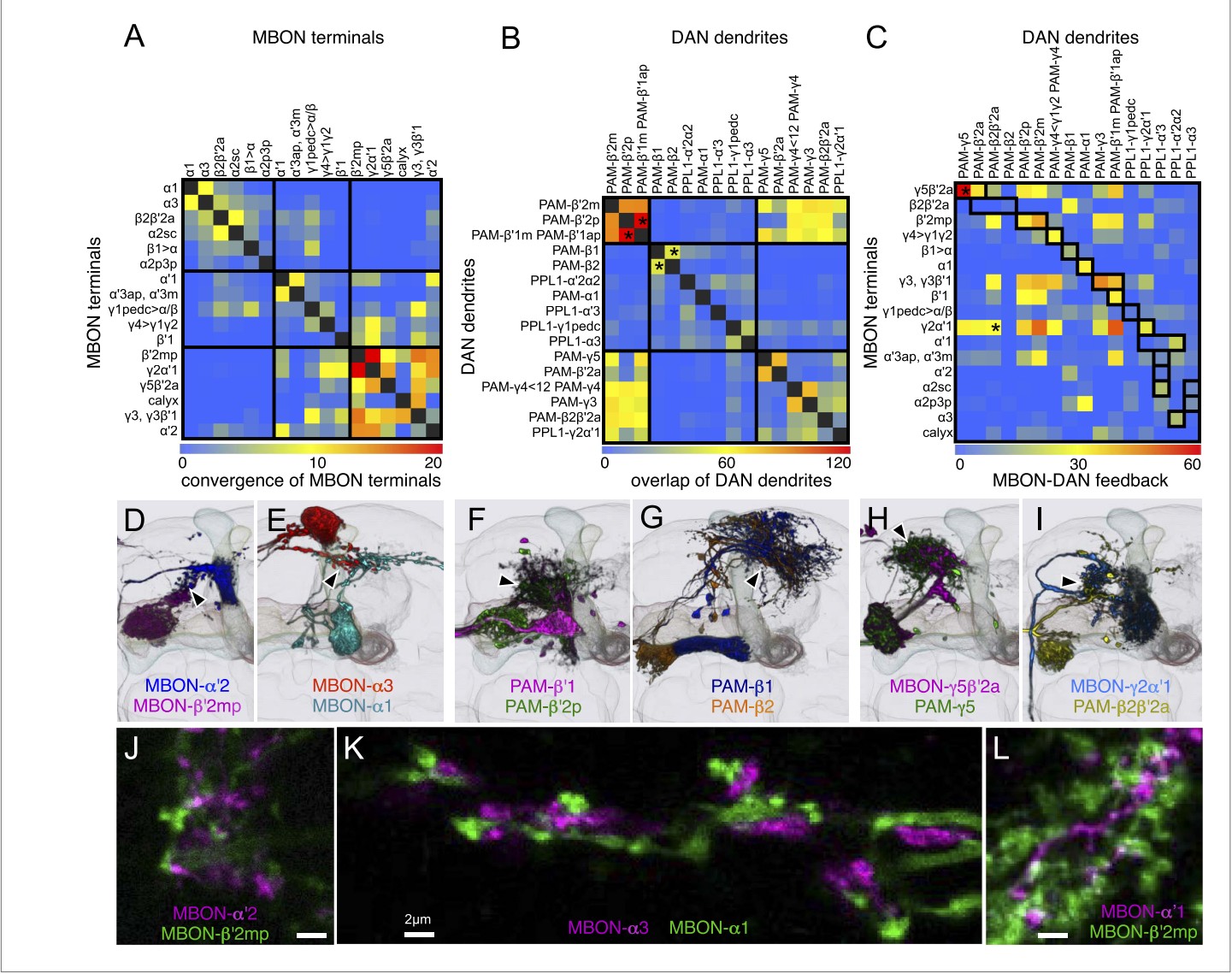

**Figure 20**. Convergence of MBON and DAN processes outside the MB lobes. (**A–C**) Matrices showing co-localization of the processes of MB extrinsic neurons outside the MB. Axon terminals and dendrites from each cell type were first segmented and the degree of overlap calculated (see 'Materials and methods') and displayed in the form of a linear heat map with arbitrary units. For those cells in the matrices marked with an asterisk, images of the overlapping neurons are shown in (**D–I**). The heavy black lines in (**A**) and (**B**) show the three major groups as determined by hierarchical clustering (Ward's method). (**A**) Overlap of the MBON terminals. (**B**) Overlap of DAN dendrites. (**C**) Overlap of the MBON terminals and DAN dendrites. The cells of the matrix outlined by the heavy black lines indicate potential feedback loops where MBON terminals from a compartment lie in close proximity to the dendrites of DANs that innervate that same compartment. Note that the atypical neuron with dendrites in the calyx (MB-CP1, see **Table 1**) is also shown in panels A and C (labeled 'calyx'). (**D** and **E**) Convergence (arrowhead) of the terminals of MBONs from α'2 and β'2mp (**D**) and of the terminals of MBONs from α3 and α1 (**E**). The overlap of the axon terminals (also shown in two-color labeling experiments, panels **J** and **K**) suggests that different MBONs may target the same post-synaptic neurons. It is interesting that MBONs of different neurotransmitter types exhibit overlapping axon terminals (e.g., cholinergic MBON-α3 and glutamatergic MBON-α1, **E**), suggesting that the target neuron may be modulated differently by each of the MBONs; for example, receiving excitatory input from cholinergic MBONs and inhibitory inputs from glutamatergic MBONs. (**F**) Overlap (arrowhead) of dendrites of PAM cluster DANs that innervate β'1 and the β'2p. (**G**) Overlap (arrowhead) of dendrites of PAM cluster DANs that innervate β1 and β2. (**H**) Overlap (arrowhead) of the terminals of the MBON-γ5β'2 and dendrites of PAM-γ5. (**I**) Overlap (arrowhead) of the terminals of the MBON-γ2α'1 and dendrites of PAM-β2β'2a. (**J–L**) Single confocal slices showing two-color labeling of the axonal terminals of the indicated MBONs. The following split-GAL4 and LexA driver lines were used: (**J**) MB081C and *R25D01-LexA*; (K) G0239 (*Chiang et al., 2011*) and *R71C03-LexA*; (**L**) MB026B and *R12C11-LexA*.

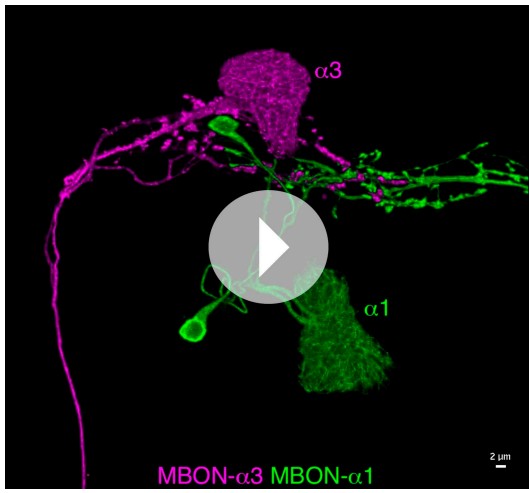

α3

α1

2 µm

MBON-α3 MBON-α1

**Video 6**. Convergence of MBON-α3 with MBON-α1. Two-color labeling of MBON-α3 and MBON-α1 by G0239 and *R71C03-LexA* as in *Figure 20K*. MBON-α1 was segmented from the pattern of *R71C03-LexA*. The video displays the projection patterns of both MBON types, then zooms to the SIP, where terminals of MBON-α3 and MBON-α1 are located, often within submicron distances of one another.

*NSyb-QF* was generated by PCR amplifying the 1.9 kb *Eco*RI fragment of NSyb from pGWB-NSyb (gift of Julie Simpson), adding the Drosophila Synthetic Core Promoter (DSCP) (*Pfeiffer et al., 2008*) by overlap-extension PCR and cloning the resulting fragment between the *Mlu*I and *Eco*RI sites of pQUAST (*Potter et al., 2010*) replacing the QUAS and *hsp70*-promoter sequences. A QF coding sequence (QFrco; gift of Christopher Potter) was then cloned into this vector using *Eco*RI and *Bam*HI. Transgenic flies were obtained by standard P-element mediated transgenesis (Genetic Services, Inc.). A total of four independent transformants were identified and in all cases most neurons were positive for QF as assessed by multiple lines bearing *QUAS-PA-GFP*; we used the line with the highest expression levels. A small group of neurons, including a subset of neurons within the PPL1 cluster, however, did not produce detectable QF activity in any of the *NSyb-QF* inserts (data not shown). To make *MB247-QS*, a 247 bp sequence of *Mef2* encoding the MB247 enhancer was PCR amplified from genomic DNA, the DSCP sequence added by overlap-extension PCR, and the MB247–DSCP fragment cloned between the *Mlu*I and *Eco*RI sites of pQUAST. The coding sequence of QS was then added using *Eco*RI and *Not*I. Five independent transformants were recovered, and all exhibited suppression of QF activity in α/β and γ KCs as assessed with *NSyb-QFrco*. To generate *QUAS-C3PA-GFP* and *QUAS-SPA-GFP*, the C3PA-GFP and SPA-GFP coding sequences, flanked with *Eco*RI-CAAC (a *Drosophila* Kozak sequence) at the 5′-end and *Not*I at the 3′-end, were cloned into pQUAST (*Potter et al., 2010*). To generate *10xUAS-C3PA-GFP* and *10xUAS-SPA-GFP*, the C3PA-GFP and SPA-GFP coding sequences were blunt-end cloned into *pJFRC-MUH* (*Pfeiffer et al., 2008*) that had been digested with *Not*I and *Xho*I. Transgenic flies were obtained by phiC31-integrase mediated transgenesis (Genetic Services, Inc.) with insertion in *attP40*, *attP2*, *VK00027*, *VK00005* for C3PA-GFP and in *attP40* for SPA-GFP.

## Screening of split-GAL4 intersections

To identify enhancer fragments that drive expression in the MB cell types, we screened a database of the adult brain expression patterns of 7000 GAL4 driver lines (*Pfeiffer et al., 2008*; *Jenett et al., 2012*). We then generated approximately 400 transgenic lines that express either the transcription activation domain (p65ADZp) or the DNA binding domain (ZpGAL4DBD) of GAL4 under the control of one of the selected enhancers using the vectors described in *Pfeiffer et al., (2010)*. We also generated p65ADZp and ZpGAL4DBD lines using the control regions from the genes encoding the enzymes for synthesizing monoamine neurotransmitters: tyrosine hydroxylase, dopamine decarboxylase, tryptophan hydroxylase, and tyrosine decarboxylase. To assay the expression pattern produced by an intersection of two enhancers, we visualized GAL4 activity in the progeny of a cross between a line expressing p65ADZp under one enhancer and a line expressing ZpGAL4DBD under the other enhancer.

We screened the expression patterns observed in female brains of more than 2500 different p65ADZp-ZpGAL4DBD combinations, each chosen based on our anatomical analyses of the original GAL4 lines as likely sharing expression in a particular cell type. For screening expression patterns generated by p65ADZp and ZpGAL4DBD combinations, we crossed males carrying *pJFRC200-10XUAS-IVS-myr::smGFP-HA* in *attP18*; the ZpGAL4DBD transgene in *attP2* with virgin females carrying the p65ADZp transgene in either *su(Hw)attP8*, *attP40*, or *VK00027* and examined expression in 3- to 10-day old female progeny. For screening, we performed immunohistochemistry as described below in Terasaki 60-well microtiter plates (Thermo Scientific, Waltham, MA) containing 8 µl of solution. To obtain

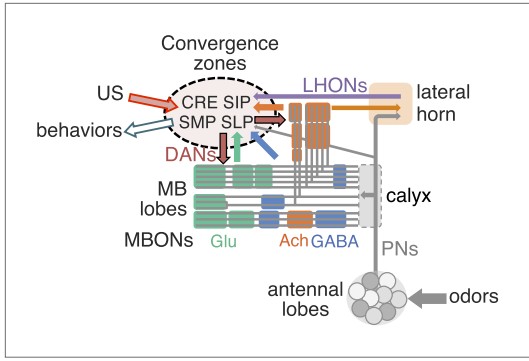

**Figure 21**. Schematic of the proposed convergence zone. MB lobes consist of three groups of compartments based on the putative transmitter of MBONs (glutamate, GABA, and acetylcholine, shown color-coded). These compartments are interconnected inside the lobes and the MBONs send converging outputs (color-coded arrows) to small subregions within five neuropils: the lateral horn (LH), CRE, SMP, SIP, and SLP. The dendrites of DANs are also confined mostly in the convergence zones within the CRE, SMP, SIP, and SLP. These neuropils, therefore, must receive information encoding the unconditioned stimuli (US), such as sugar and shock, which recruit DANs for memory formation. In the simplest model, these same sensory inputs would activate not just the DANs but also other output neurons of these neuropils that elicit appropriate unconditioned behaviors. MBON, conveying the learned valence of stimuli, might then terminate onto these same output neurons. A subset of projection neurons from the antennal lobe and LH output neurons (LHONs) also converge with MBONs in these zones (see accompanying paper **Aso et al., 2014**). In this way, learned and innate responses could use a common set of downstream circuits, originating in the LH, CRE, SMP, SIP, and SLP, to drive behavior.

polarity and higher resolution information on selected lines, split-GAL4 lines were crossed to *pJFRC51-3xUAS-Syt::smGFP-HA* in *su(Hw) attP1*; *pJFRC225-5xUAS-IVS-myr::smGFP-FLAG* in *VK00005* and four females brains plus two ventral nerve cords (VNCs) were dissected per line and immunolabeled in 2.0 ml tubes as described below. We further characterized lines we intended to use in behavioral experiments. First, we compared the intensity and specificity of lines that drive expression in the same cell types by immunostaining in parallel, mounting on the same glass slide, and imaging with identical confocal settings. Second, we screened for expression in non-neuronal tissues by crossing lines with *pJFRC2-10xUAS-mCD8::GFP* in *VK00005* and examining the bodies of F1 progeny by stereo fluorescence microscopy; we observed fluorescence in non-neuronal tissues in 17 out of the 176 split-GAL4 lines (see **Figure 2—figure supplement 7** for examples).

**Figure 2—figure supplements 2–6** and **Supplementary file 1** document the lines used in this and the accompanying paper (**Aso et al., 2014**). These include KCs (**Figure 2—figure supplement 2**), DANs (**Figure 2—figure supplements 3 and 4**), and MBONs (**Figure 2—figure supplement 5**). We also made split-GAL4 lines for a variety of other modulatory input cell types that are putatively serotonergic, GABAergic, octopaminergic, and peptidergic (**Figure 2—figure supplement 6**), but we did not characterize these lines further in this study as their projections are not limited to localized areas of the MB lobes. Confocal image stacks documenting the expression patterns of the 92 selected lines in adult female brains and VNCs are available online (http://www.janelia.org/split-gal4).

Because the observed expression pattern depends to some extent on the reporter used, and in particular on its site of genomic insertion, we assayed the expression of the 92 selected lines (**Supplementary file 1**) independently with reporter constructs inserted at each of the two chromosomal sites (*attP18* and *VK00005*) that we intended to employ in future behavioral experiments. Consistent expression patterns were observed when using different UAS reporters; but for some lines, the intensity of expression observed in both the targeted MB neurons and off target cells varied significantly with the reporter used. *pJFRC2-10xUAS-IVS-mCD8::GFP* and *pJFRC225-5xUAS-IVS-myr::smGFP-FLAG* in *VK00005* expressed more strongly in the targeted MB neurons than *pJFRC200-10XUAS-IVS-myr::smGFP-HA* in *attP18*, but the two reporters (*pJFRC2* and *pJFRC225*) in *VK00005* occasionally visualized off-targeted cells that were not visible with *pJFRC200* in *attP18*.

## Cell counting

The small number of MB extrinsic neurons expressing split-GAL4 in most lines, generally 1 to 14 cells per brain hemisphere, permitted the use of simple visual inspection to judge the completeness and reproducibility of the expression pattern. However, for lines expressing in subsets of the KCs, often several hundred cells, we also employed computer-assisted cell counting. KC nuclei were visualized with *UAS-nlsLacZ* and their membranes with *pJFRC225-5xUAS-IVS-myr::smGFP-FLAG*. The cell body cluster of the KCs was imaged at high-resolution (0.1 μm × 0.1 μm × 0.1 μm voxels). To improve

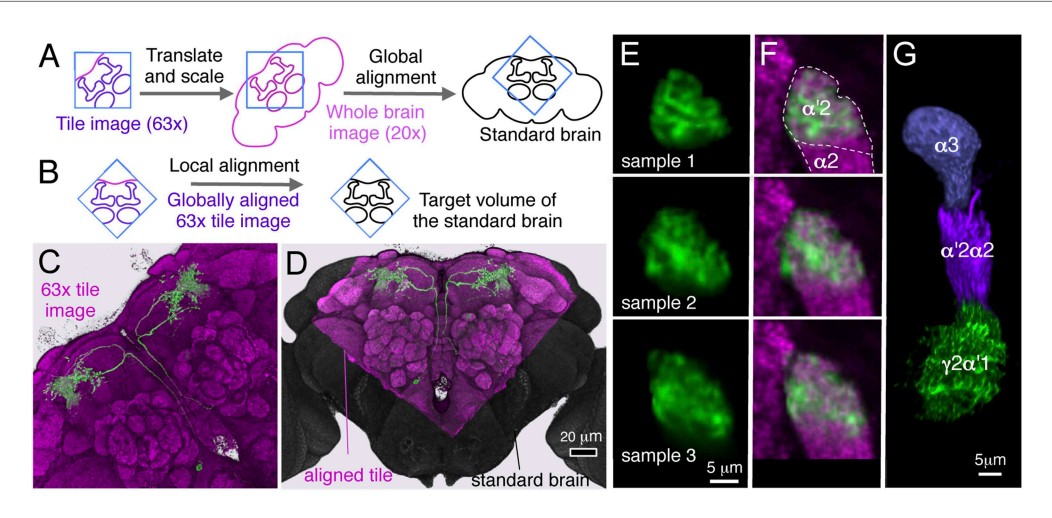

**Figure 22**. Alignment of partial brain images to a standard brain model. (**A**) A low-resolution (20×) confocal stack covering the entire brain and optic lobes (magenta diagram) and a single high-resolution (63×) confocal stack of the portion of the brain containing the MB (blue square) are collected for each specimen. The 63× tile image is first aligned to a whole brain image of the same brain by scaling and rigid translation using labeling of the presynaptic active zone protein Bruchpilot by the mouse monoclonal antibody nc82 (*Laissue et al., 1999*; *Hofbauer et al., 2009*) as the reference. The whole brain image is then globally aligned to the JFRC2013 standard brain (black diagram) by affine registration. In this way the high-resolution tile image is globally aligned to the standard brain and corresponding target volume in the standard brain is identified. (**B**) The nc82 pattern of the high-resolution tile image is non-rigidly aligned to that of the standard brain. See 'Materials and methods' for more detail. (**C**) A 63× image tile of a brain (green, MB018B-driven expression pattern showing MBON-α'2 neurons; magenta, nc82 staining). (**D**) The same tile after alignment (gray; nc82 pattern of the standard brain). (**E**) Portions of optical slices from confocal stacks of three brains from the MB018B driver line. (**F**) After alignment, the dendrites of MBON-α'2 neurons in all three samples can be seen to be confined within the α'2 compartment of the standard brain. (**G**) An alignment to the standard brain of three separately imaged brains, each of which visualized a different PPL1 cluster dopaminergic neuron; compare to *Figure 11F*, where the same three neurons were imaged in a single brain using MCFO. DOI: 10.7554/eLife.04577.044

separation of neighboring nuclei, myr::smGFP-FLAG signals were subtracted from the nlsLacZ channel. Nuclei were segmented using the 3D component analyzer plugin of the Fluorender (Voxelpress; http://www.voxelpress.com/); first, we counted nuclei-sized (mean ± 2SD) nlsLacZ-stained objects at low threshold, deleted the detected volumes, and then counted the remaining nuclei at higher threshold. The counting results were visualized by randomly assigning colors and numbers to each counted object. This procedure gave results consistent with previous manual counting of c739 and c305a drivers (*Aso et al., 2009*). The signal intensity of nlsLacZ varied between nuclei in the same sample; we counted all detectable objects.

## Photoactivatable GFP tracing

Brains of 3- to 7-day old females were dissected in saline (108 mM NaCl, 5 mM KCl, 5 mM HEPES, 5 mM trehalose, 10 mM sucrose, 4 mM NaHCO₃, 1 mM NaH₂PO₄, 2 mM CaCl₂, 1 mM MgCl₂, pH-7.3) and mounted dorsal-anterior up in a Sylgard-coated Petri dish. Photoactivation was performed using a two-photon laser scanning microscope (Ultima, Prairie Technologies, Middleton, WI) with an ultra-fast Ti:S laser (Chameleon Vision, Coherent, Santa Clara, CA) modulated by Pockels Cells (Conoptics, Danbury, CT). A water immersion objective (60X/1.0 NA, Olympus, Japan) was used for both visualization and photoactivation. Emitted photons were detected by a GaAsP detector (Hamamatsu Photonics, Japan) for green fluorescence and a PMT for red fluorescence. The laser was tuned to 925 nm for visualizing the samples (with an intensity of ~0.7–2 mW as measured after the objective) and 710 nm for photoactivation. The Pockels Cells bias voltage was adjusted to obtain maximum signal-to-noise ratio when tuned at 710 nm, and photoactivation laser intensity at 710 nm was adjusted to be between 2 and 3 mW as measured after the objective. The photoactivation scan was performed with a pixel size of 0.24–0.39 μm and pixel dwell time of 2 μs. Each pixel was scanned four times successively using the

frame-averaging function of the microscope software (PrairieView, Prairie Technologies); this was repeated with each repetition separated by 10–30 s. A target volume, such as the MB lobes (*Figure 4*) or a single MB lobe compartment (*Figure 5*), was divided into 8–15 z-slices with 2–5 μm steps, depending on the size of the target as well as the orientation of the sample. The number of repetitions of the photoactivation scan depended on the expression levels of PA-GFP as well as the depth of the photoactivation target. For photoactivation of the MB lobes, the photoactivation scan was repeated 30–60 times. For photoactivation of a single compartment, the photoactivation scan was repeated 90–120 times. All photoactivated samples were prepared for confocal microscopy as follows: fixed for 45 min at RT using 2% PFA/PBL (2% paraformaldehyde in 75 mM lysine, 37 mM sodium phosphate buffer, pH 7.4), washed multiple times in PBS containing 0.3% Triton X-100 (PBST), blocked with 10% normal goat serum in PBST, incubated in primary antibodies overnight at 4°C, washed multiple times in PBST, and incubated in secondary antibodies overnight at 4°C (or for more than 3 hr at RT) before final washes with PBST. The samples were mounted using either VECTASHIELD (Vector Labs, Burlingame, CA) or SlowFade Gold (Life Technologies, Grand Island, NY) and confocal imaging was performed using an LSM510 with a Plan-Neofluar 40X/1.3 objective (Zeiss, Germany). The following primary and secondary antibodies were used: rabbit-anti-DsRed (1:1000, Clontech), nc82 (1:10, Developmental Studies Hybridoma Bank), mouse anti-tyrosine hydroxylase (1:100, EMD Millipore, Germany), rat anti-Dopa decarboxylase (1:400, a gift from Jay Hirsch), rabbit anti-TexasRed (1:500, Invitrogen, Life Technologies), Alexa Fluor 568 goat anti-rabbit (1:200, Life Technologies), Alexa Fluor 633 goat anti-rat (1:200, Life Technologies), and Alexa Fluor 633 goat anti-mouse (1:200, Life Technologies).

We used flies expressing PA-GFP pan-neuronally with the exception of the KCs, generated with a *Synaptobrevin-GAL4* driver (*NSyb-GAL4 2-1*; gift of Julie Simpson) in combination with *MB247-GAL80*, to identify the MB extrinsic neurons by photoactivation of the MB lobes (*Figure 4*). Generation of the photoactivation mask for the MB lobes was guided by a red fluorescent protein expressed in KCs (*MB247-DsRed*, a gift of Andre Fiala). In initial experiments, a volume of ~800 μm × ~600 μm × ~220 μm covering most of the central brain was imaged at a pixel size of 0.39 μm × 0.39 μm, with z-step size of 3 μm before and after photoactivation of the MB lobes in a hemisphere. This identified five clusters of MB extrinsic neuron cell bodies, each of which reproducibly contained more than two cells (n = 3 brains, data not shown, see *Figure 4*). Neurons found reproducibly, but not within these clusters, include MB-APL, MB-DPM, two MBONs located by the contralateral spur (MBON-γ4>γ1γ2 and MBON-β1>α; MV2), and two MBONs located in the contralateral anterior lateral region (MBON-γ3 and MBON-γ3β′1). We examined the number of neurons in each of the five clusters except the anterior PAM cluster (see below) by comparing higher resolution images of areas containing each cluster taken before and after photoactivation. Two or three clusters were examined per sample, and a volume of ~200 μm × ~200 μm × ~100 μm was imaged at 0.39 μm × 0.39 μm × 1 μm resolution for each cluster. Images taken before and after photoactivation were aligned using a subpixel registration algorithm (*Guizar-Sicairos et al., 2008*) and correlation between registered images. Slight shifts in sample orientation precluded the use of a simple image calculation to identify photoactivated neurons. Instead, the photoactivated cell bodies were visually identified from aligned images assisted by a custom MATLAB interface. This analysis identified two neurons with their cell bodies located in regions that do not contain any of the neurons identified in the split-GAL4 lines. One was located ventral to the calyx (MBON-γ4γ5) and the other was located ventral lateral to the antennal lobe (MBON-γ1γ2) (see below, and *Figure 5G* and *Figure 5—figure supplement 1*). They represent the only two cell types identified by PA-GFP tracing that were not found in the split-GAL4 lines.

It is important to note that we were not able to reproducibly visualize neurons with elaborate arbors over multiple neuropils and only diffuse and sparse processes in the MB lobes, such as octopaminergic neurons and SIFamide peptidergic neurons (data not shown). This indicates a limitation for the PA-GFP tracing experiments, at least using the parameters described above, in detecting neurons with a large cytoplasmic volume by just photoactivating PA-GFP molecules within a small fraction of the total volume.

We performed photoactivation of individual compartments to more specifically visualize the MB extrinsic neurons innervating each compartment (*Figure 5*). PA-GFP was expressed pan-neuronally using the Q-system (*Potter et al., 2010*) except in the α/β and γ KCs (*NSyb-QF, MB247-QS, QUAS-PA-GFP*) and each compartment was demarcated by a red fluorescent protein (myr::tdTomato) expressed using a split-GAL4 driver. The photoactivation mask was generated using this red marker, and in some cases where the split-GAL4 driver labels two compartments, this was further restricted using basal fluorescence of PA-GFP in the α′/β′ KCs (for example, to demarcate the β2 compartment

using MBON-β2β'2a, *Figure 5A*). Upon photoactivation, the brains were fixed and immunostained for tdTomato and tyrosine hydroxylase or dopa-decarboxylase as described above. The confocal images were processed by a custom MATLAB code to identify photoactivated cells. For each image, the fluorescent intensity of the photoactivated GFP signals was measured by drawing a region-of-interest around tdTomato positive neurons (i.e., neurons whose processes had been used as target for the photoactivation), and the green channel of the image was thresholded by a pixel intensity value representing mean minus 2× standard deviations of all the pixels within the regions-of-interest. A cell was counted as photoactivated if most of the pixels within the cell, as determined visually, were over this threshold value. Finally, we determined whether these cells are dopaminergic by examining co-localization with tyrosine hydroxylase or dopa-decarboxylase. Photoactivation of each of the 15 compartments identified all the MBONs included in the split-GAL4 lines (*Figure 5*, and data not shown). We observed the same number of PA-GFP positive cell bodies for each MBON cell type as were labeled by the corresponding MBON split-GAL4 line (e.g., *Figure 5A,B*) and therefore these experiments confirmed that the split-GAL4 lines label all neurons within each MBON type. For the MBONs with dendrites in α'1 and α'3 compartments and cell bodies near KC cell bodies, it was not possible to assess whether there are more neurons of these types, because the cell bodies of photoactivated α'/β' KCs were not distinguishable from those of the MBONs.

Photoactivation of the MB lobes resulted in labeling of over 100 cells in the anterior medial cluster including the PAM-DANs (cluster 5 in *Figure 4*) and it was not feasible to accurately count them by the methods described above. We therefore employed two independent approaches to examine the MB neurons in this cluster. We first performed photoactivation of the MB lobes with flies carrying *R58E02-GAL80* (*Liu et al., 2012*) transgene that suppresses *NSyb-GAL4*-mediated PA-GFP expression in all PAM-DANs. We observed a dramatic reduction in the number of photoactivated cells in the anterior medial cluster ipsilateral to the photoactivated lobes as compared to flies without *R58E02-GAL80* (data not shown). This suggests that many of the MB extrinsic neurons in the anterior medial cluster are PAM-DANs. Moreover, we observed no photoactivated cells in the contralateral anterior medial cluster when using *R58E02-GAL80*, indicating that the MB extrinsic neurons in this region are all PAM-DANs. We then characterized the MB extrinsic neurons in the anterior medial cluster by photoactivation of individual compartments as described above. We identified all the MBONs in this cluster and confirmed their numbers. We observed that there are more photoactivated GFP positive neurons in this cluster than those labeled by the split-GAL4 lines expressed in PAM-DANs (see for example *Figure 5C,D*). These are likely dopaminergic as confirmed by the immunostaining (*Figure 5E* and data not shown).

It is important to note that the identification of neurons using PA-GFP tracing critically depends on the signal-to-noise ratio of photoactivated fluorescence relative to basal fluorescence of PA-GFP molecules. Neurons may not be detected if they express insufficient levels of PA-GFP, as observed in some cases using NSyb-enhancer, or if the photoactivated volume contains only a small fraction of a neuron's processes. For example, we observed little expression of *NSyb-QF* in a subset of DANs in the PPL1 cluster as well as in PAM-α1 DANs (data not shown). Thus, we could not assess whether the split-GAL4 lines label all the PPL1-DANs and PAM-α1 DANs.

We used dye-filling to visualize one of the MB extrinsic neurons identified by PA-GFP that was not included in the split-GAL4 collection (*Figure 5G*). Flies were generated in which PA-GFP was expressed pan-neuronally except in the KCs (genotype in *Figure 5G* legend). No red fluorescent protein was expressed. Photoactivation was targeted to the medial lobes using the absence of basal PA-GFP fluorescence in the KCs to demarcate the MB lobes. Upon photoactivation, the brain was treated with collagenase (2 mg/ml, Sigma) for 30 s, the glial sheath removed using fine forceps, and the brain was mounted again in Sylgard-coated Petri dish in saline. A fire-polished, pulled glass pipette (0.5 mm ID, 1.0 mm OD, Sutter) was backfilled with a TexasRed dye (lysine-fixable 3000 MW, Invitrogen) dissolved in saline. A two-photon microscope was used to guide the pipette to the photoactivated cell body, and the dye was injected into the cell body by iontophoresis using over one hundred 3 ms pulses of 20 Vdc applied every 0.5 s. The dye was allowed to diffuse for an additional 10 min and then the brain was fixed and immunostained with anti-TexasRed antibody and nc82. Four brains were examined and in all cases we identified one neuron with the same morphology.

## Immunohistochemistry

Dissection and immunohistochemistry of fly brains were done as previously described with minor modifications (*Jenett et al., 2012*). Brains and VNCs of 3- to 10-day old female flies were dissected in

Schneider's insect medium and fixed in 2% paraformaldehyde in Schneider's medium for 55 min at room temperature (RT). After washing in PBT (0.5% Triton X-100 in PBS), tissues were blocked in 5% normal goat serum (or normal donkey serum, depending on the secondary antibody) for 90 min. Subsequently, tissues were incubated in primary antibodies diluted in 5% serum in PBT for 2–4 days on a nutator at 4°C, washed three times in PBT for 30 min or longer, then incubated in secondary antibodies diluted in 5% serum in PBT for 2–4 days on a Nutator at 4°C. Tissues were washed thoroughly in PBT four times for 30 min or longer and mounted on glass slides for imaging (see below for the mounting protocol).

The following antibodies were used: rabbit anti-GFP (1:1000; Invitrogen; A11122), mouse anti-nc82 (1:33.3; Developmental Studies Hybridoma Bank, Univ. Iowa) (*Hofbauer et al., 2009*), rabbit anti-HA (1:300; Cell Signaling Technology, Danvers, MA), rat anti-FLAG (1:200; Novus Biologicals, Littleton, CO), mouse anti-*Drosophila* ChAT (ChAT4B1; 1: 100; Developmental Studies Hybridoma Bank, Univ. Iowa) (*Takagawa and Salvaterra, 1996*), rabbit anti-GABA (1:500; A2052, Sigma-Aldrich, Switzerland), rabbit anti-5HT antiserum (1:1000; Sigma-Aldrich, catalog no. S-5545), mouse anti-tyrosine hydroxylase (LNC1, Millipore), rat anti-DDC (1:400; a gift from Dr J Hirsh) (*Beall and Hirsh, 1987*), mouse anti-beta galactosidase (1:200; Abcam, Cambridge, MA), mouse anti-V5-TAG (1:1000; AbD Serotec, UK), Dylight-549 conjugated mouse anti-V5 (1:500; AbD Serotec), rabbit anti-*Drosophila* GAD1 (1:1000; a gift from Dr. FR Jackson), rabbit anti-DvGluT (1:5000; a gift from Dr. A DiAntonio) as primary antibodies, and cross-adsorbed secondary antibodies to IgG (H+L): AlexaFluor-488 donkey anti-mouse (1:400; Jackson Labs), AlexaFluor-594 donkey anti-rabbit (1:500; Jackson Labs, Sacramento, CA), Cy3 donkey anti-rabbit (1:500; Jackson Labs), AlexaFluor-647 donkey anti-rat (1:300; Jackson Labs), AlexaFluor-488 goat anti-rabbit (1:800; Invitrogen A11034), and AlexaFluor-568 goat anti-mouse (1:400; Invitrogen A11031).

For determining the likely transmitter used by each MBON cell type, we immunolabeled brains from flies carrying the appropriate split-GAL4 drivers and *pJFRC225-5xUAS-IVS-myr::smGFP-FLAG* in *VK00005*. Tissues were first incubated in primary antibody against GABA, GAD1, or DvGluT for 2–3 days at 4°C, washed, incubated in secondary antibody for 2–3 days, and washed overnight. To visualize the MBON, we then incubated tissues in either rabbit anti-GFP or rat anti-FLAG (depending on the host species of other primary antibody) for 2–3 days, washed, and then incubated in secondary antibody for 2–3 days. Mixtures of 40–60 brains from 17 split-GAL4 drivers were stained in the same tube and mounted and imaged on the same glass slide to enable an unbiased comparison of immunoreactivity across MBONs. The expression pattern of the myr::smGFP-FLAG was used to genotype the brains. For other protocols, tissues were incubated in mixtures of multiple primary or secondary antibodies.

## Clearing and mounting

After immunohistochemistry, tissues were post-fixed with 4% PFA in PBS for 4 hr at RT followed by four, 15 min washes in PBT. To improve adhesion during mounting, tissue were washed in PBS (15 min) to remove the Triton and then placed on poly-L-lysine-coated cover slips to which they electrostatically adhere. Tissues were then dehydrated through a series of ethanol baths (30%, 50%, 75%, 95%, and 3 × 100%) for 10 min each and then 100% xylene three times for 5 min each in Coplin jars. Samples were embedded in a xylene-based mounting medium (DPX; Electron Microscopy Sciences, Hatfield, PA), and the DPX was allowed to dry for 2 days before imaging. For comparing expression intensities, up to 60 brains and VNCs were mounted on the same cover slip. Because tissues were attached to the flat surface of the cover slip in the same orientation, the same brain structures were located at the same depth during confocal imaging, facilitating a fair comparison of signal intensity across samples.

## Image acquisition and analysis

Imaging was done on an LSM710 confocal microscope (Zeiss). Brains and VNCs were imaged first at low-resolution using a Plan-Apochromat 20x/0.8 M27 objective (voxel size = 0.56 × 0.56 × 1.0 μm; 1024 × 1024 pixels per image plane). The region including the neurons of interest was then imaged at higher resolution by using a Plan-Apochromat 63x/1.40 oil immersion objective (voxel size = 0.19 × 0.19 × 0.38 μm; 1024 × 1024 pixels). For cell types too large to fit in a single image, regions of interests were scanned separately with multiple tiles (a maximum of five tiles was required to cover the entire brain and optic lobes) that were then stitched (*Yu and Peng, 2011*).

Confocal images were analyzed using the Janelia Workstation, a suite of tools for viewing and analyzing image data (S Murphy; K Rokicki; C Bruns; Y Yu; L Foster; E Trautman; D Olbris; T Wolff;

A Nern; Y Aso; N Clack; P Davies; S Kravitz; T Safford, unpublished), ImageJ (http://imagej.nih.gov/ij/), Fiji (http://fiji.sc/), and Fluorender (http://www.sci.utah.edu/software/13-software/127-fluorender. html; [*Wan et al., 2009*]). Figure panels with black backgrounds are single slices or maximum intensity projections of confocal stacks or substacks. Figure panels with white or gray backgrounds show neurons that have been segmented using FluoRender.

## Brain alignment and neuropil masks

Making an anatomical atlas from confocal images of many different split-GAL4 lines depends on being able to align data collected from individual specimens onto the framework of a standard brain. In order to minimize the amount of deformation required in the brain alignment process, we prepared a standard brain (JFRC2013) using fixation and dehydration steps identical to those used to prepare our experimental samples. After stitching (*Yu and Peng, 2011*) five tiles of high-resolution confocal image stacks covering the entire brain and optic lobes, debris on the brain surface were removed from the image using Fluorender. We then generated a downscaled (0.38 µm isotropic voxels) version of the JFRC2013 standard brain for use in alignment.

Because confocal imaging time was a limiting factor, we sought to develop a method that enabled alignment of a single 63× confocal image stack that covered only a portion of the brain to a standard model of the entire brain. Our alignment strategy is outlined in *Figure 22*. We imaged each brain twice: the entire brain at low-resolution (20× objective lens; voxel size, 0.56 × 0.56 × 1.0 µm) and a confocal stack (or stacks) covering the region of interest at high-resolution (63× objective lens; voxel size, 0.19 × 0.19 × 0.38 µm). The two images were scaled to have isotropic voxels; because the low-resolution images were obtained with an air objective, they were optically flattened compared to the high-resolution images and, thus, required scaling in the z-axis. We then aligned the high-resolution image tile to the low-resolution whole brain image, using the reference nc82 channel, by means of image stitching (*Yu and Peng, 2011*), which obtains translations through searching the maximum normalized cross correlation using the fast Fourier transform (FFT). Then affine registration of the low-resolution whole brain image to the JFRC2013 standard brain was used to provide a global alignment of the partial brain image to the standard brain. Finally, a non-linear transformation was applied to locally register the high-resolution tile image to the JFRC2013 standard brain using a symmetric diffeomorphic registration algorithm (*Avants et al., 2008*) with a combination of mutual information and normalized cross correlation as the similarity metric in the local alignment.

For generating the atlas, we used only ~25% of the specimens that showed the best alignment to the standard brain based on the reference nc82 marker. In this way we were able to obtain very high-quality alignments as judged by two criteria. First, the alignments to the standard brain obtained from multiple specimens of the same GAL4 line were very similar (*Figure 22E,F*). Second, we observed the same relative arrangement of cell types when assessed using images from a single brain (*Figure 11F*) and by alignment of images of separate brains (*Figure 22G*). We routinely obtained alignment data of a given cell type from multiple brains to allow us to assess biological and alignment variability between samples.

Neuropil masks were generated in the JFRC2013 standard brain by alignment to the binary masks of the neuropils defined in *Ito et al. (2014)*, followed by manual editing of the neuropil borders guided by the nc82 staining of the JFRC2013 brain. To generate masks for each MB compartment, we averaged the registered intensities of MB extrinsic neurons projecting to the same compartment and samples representing the same KC cell type; we manually defined the borders between neighboring compartments after applying a Gaussian blur filter in 3D (sigma = 2).

## Distribution of terminals and dendrites in the brain

After normalization of intensities between images, terminals and dendrites were segmented based on morphology and Syt::smGFP-HA distribution using FluoRender (*Wan et al., 2009*, *2012*). To determine the distribution of projections in different brain regions shown in *Figure 18E*, signals from the cell types making up each of five groups—glutamatergic, GABAergic and cholinergic MBONs and the PPL1 and PAM DANs—were averaged within a group. Then the total signal within the volume of each neuropil mask for each of the five groups was divided by total signal observed in all neuropils. To calculate signal intensities shown in *Figure 19C*, the average of signals in 10 × 10 × 10 voxel volumes (3.8 µm in each dimension) were used.

To estimate degree of overlap between processes of MBONs and DANs in *Figure 20*, we used the method previously described by *Cachero et al. (2010)*. We had multiple images for each cell type and

we treated the two brain hemispheres separately, giving us on average 17.4 image pairs per cell type combination. We computed the overlap for each image pair separately; each cell in the matrices shown in *Figure 20A–C* represents the mean value.

## Acknowledgements

We thank Yichun Shuai (Cold Spring Harbor Laboratory) for pointing out the presence of MB extrinsic neurons in the GAL4 line *VT999036*, Barry Dickson for providing *VT999036*, and Christopher Potter (Johns Hopkins) for proving the QFrco vector and sharing unpublished information. F Rob Jackson (Tufts University), Aaron DiAntonio (Washington University), Jay Hirsh (University of Virginia), and the Developmental Studies Hybridoma Bank (created by the NICHD of the NIH and maintained at the University of Iowa) provided antibodies and Yong Wan and Hideo Otsuna (University of Utah) added new functions to Fluorender needed for this work. The Janelia Scientific Computing Software group (Sean Murphy, Konrad Rokicki, Christopher Bruns, Les Foster, Eric T Trautman, Donald J Olbris, Nathan Clack, Pete Davies, Saul Kravitz, Todd Safford, Charlotte Weaver, and Rob Svirskas) generated software tools, the Janelia Fly facility (Amanda Cavallaro, Todd Laverty, and others) helped in fly husbandry, and the FlyLight Project Team (Nick Abel, Gina DePasquale, Adrianne Enos, Joanna Hausenfluck, Phuson Hulamm, Reeham Motaher, Omatara Ogundeyi, Allison Sowell, Susana Tae, and Rebecca Vorimo) performed brain dissections, histological preparations, and confocal imaging. Margaret Bezrutczyk and Camille Rogine assisted in generating reagents for the PA-GFP experiments. Phyllis Kisloff assisted in manuscript preparation. We thank members of the Rubin lab (Aljoscha Nern, Chris Murphy, Barret D Pfeiffer, Tanya Wolff, Arnim Jenett, and Ming Wu) and Axel lab for materials and discussion. YA and HT initiated this work while participants in the Janelia Farm Visiting Scientist Program.

## Additional information

### Funding

| Funder | Grant reference number | Author |
|---|---|---|
| Howard Hughes Medical Institute | | Yoshinori Aso, Daisuke Hattori, Yang Yu, Rebecca M Johnston, Nirmala A Iyer, Teri-TB Ngo, Heather Dionne, Richard Axel, Gerald M Rubin |
| Howard Hughes Medical Institute | Janelia Visiting Scientist Program | Yoshinori Aso, Hiromu Tanimoto |
| Gatsby Charitable Foundation | Gatsby Initiative Fund in Brain Circuitry at Columbia University | LF Abbott |
| Swartz Foundation | | LF Abbott |
| Bundesministerium für Bildung und Forschung | Bernstein Focus Neurobiology of Learning 01GQ0932 | Hiromu Tanimoto |
| Max-Planck-Gesellschaft | | Hiromu Tanimoto |
| Deutsche Forschungsgemeinschaft | TA 552/5-1 | Hiromu Tanimoto |
| Japan Society for the Promotion of Science | MEXT/JSPS KAKENHI 25890003, 26120705, 26119503 and 26250001 | Hiromu Tanimoto |
| Naito Foundation | | Hiromu Tanimoto |
| Jane Coffin Childs Memorial Fund for Medical Research | | Daisuke Hattori |
| G Harold and Leila Y. Mathers Foundation | | LF Abbott |
| Simons Foundation | | LF Abbott |

The funders had no role in study design, data collection and interpretation, or the decision to submit the work for publication.

## Author contributions

YA, Conceived and designed the study, conducted split-GAL4 screening, acquired confocal images, analyzed and interpreted data and wrote the article; DH, Designed and carried out the PA-GFP experiments, analyzed data and wrote the article; YY, Developed image alignment software; RMJ, Prepared samples and acquired confocal images; NAI, Prepared samples and acquired confocal images; T-TBN, Conducted split-GAL4 screening and acquired confocal images; HD, Made molecular constructs; LFA, Analyzed and interpreted data and wrote the article; RA, Analyzed and interpreted data and wrote the article; HT, Conceived and designed the study, analysis and interpretation of data, drafting or revising the article; GMR, Conceived and designed the study, analyzed and interpreted data and wrote the article

# Additional files

### Supplementary file

• Supplementary file 1. List of split-GAL4 driver lines for the neurons in the mushroom body. For each driver line, the source of the enhancer fragments used to make the p65ADZp and ZpGAL4DBD constructs are given. Most enhancer fragments are from the collection of GAL4 lines described in *Jenett et al. (2012)*; see 'Materials and methods' for the details of the TH and TDC2 constructs. All ZpGAL4DBD constructs are inserted into *attP2*; the insertion sites of the p65ADZp constructs are indicated by the letter at the end of the driver name as follows: A, *su(Hw)attP8*; B, *attP40*; C, *VK00027*. The cell types in which expression is seen with each driver are indicated in the matrix; the level of expression in each cell type is indicated by the gray scale (see 'Materials and methods'). Note that observed expression depends to some extent on the reporter construct used (see *Figure 2—figure supplement 1* for an example). The data in this matrix were based on imaging with *pJFRC225-5xUAS-IVS-myr::smGFP-FLAG* in *VK00005*. The *VK00005* insertion site supports high-expression levels (Pfeiffer et al., 2012); many of the split-GAL4 drivers show more cell type specificity using reporters in *attP18* (see the accompanying paper for examples) (*Aso et al., 2014*). We did not observe expression outside the central nervous system in any of these lines except MB062C; see 'Materials and methods' and *Figure 2—figure supplement 7* for a description of the assay used.

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
