## [Decision Letter]

Thank you for sending your work entitled "The neuronal architecture of the mushroom body provides a logic for associative learning" for consideration at *eLife.* Your article has been favorably evaluated by K VijayRaghavan (Senior editor), Leslie Griffith (Reviewing editor), and 2 reviewers, one of whom is also a member of our Board of Reviewing Editors, and one of whom, Ann-Shyn Chiang, has agreed to reveal her identity.

The Reviewing editor and the reviewers discussed their comments before we reached this decision. Instead of summarizing the reviews, the Reviewing editor has appended all the comments from the two reviewers. As you see, the reviewers are impressed by the study, and most comments are meant to help further improve/clarify the presentation. These can be addressed by textual changes. We look forward to receiving the revised version of this paper.

Reviewer #1:

Aso et al. reported stunning advances in the anatomical organization of the mushroom bodies, a key center for learning and memory in insects. Together with the accompanying behavioral analysis paper, these represent major breakthrough in understanding the structure and function of the mushroom bodies, their output, and their modulatory input. These studies have certainly laid down groundwork for years to decades of future investigations, and have already provided interesting insights on the logic of information processing principles through these intriguing structures. I am extremely enthusiastic in supporting the publication of these papers. Below I provide critiques to the anatomy paper mostly to ensure that the paper is accessible to as wide an audience as it can.

1) My strongest critique is the nomenclature, in particular the classes of MBONs. As far as I understand, most of these neurons are identified for the first time, at least to the resolution described, in this paper. So, the authors are in a position to name these neurons. I understand the rationales of the authors' choice using the connectivity with specific mushroom body lobes, and it may seem rational within this paper as it builds from Kenyon cell subtypes to MBONs. But even when reading the companion paper, only a slight distance away, it is already very difficult to remember these names (e.g., MBON-gamma4>gamma1gamma2, MBON-beta'2mp_bilateral, MBON-gamm1pedc>alpha/beta, etc.) or the rationale behind without constantly checking this paper. Consider many papers that will follow this paper in many years to come! Often complex names can turn off young scientists wishing to enter an otherwise very exciting field. In my view, these names should be greatly simplified. I have a specific suggestion. Since the authors found that MBONs neatly fall within three neurotransmitter phenotypes, and that MBONs that belong to a particular neurotransmitter category intersect with specific MB lobes at specific locations (both very interesting new findings!), why not name the MBONs after neurotransmitters (which is one of the most used convention for classifying neurons), and then within a neurotransmitter category, give a number, perhaps based on the proximal-distal axis of the mushroom body axons (e.g., MBON-ACh1, ACh2...; GABA1, GABA2...; Glu1, Glu2...). An alternative is to use consecutive numbers, for instance, MBON-ACh1-ACh8, GABA9-GABA12, GLU13-19, etc. With a modified Table 1, readers can quickly check which lobe MBON-GABA3 receives input from, etc. The authors may also consider renaming the DANs but this may have more historical constraint and DANs don't neatly match MBONs. Still, some simplification also seems to be useful.

2) Definition of cell type: I would add the following "We operationally define a cell type..." at the beginning, and "The screen was near saturation in our largeGAL4 collection."

3) I may have missed among the large amount of data, but can the authors describe MBON subtypes that are modulated by previously reported appetitive vs. aversive DANs?

4) A few critiques on Figure 21 (the summary figure). It is probably better to draw a more anatomically correct mushroom body schematics (with a bifurcation), with Glu, ACh, and GABA mapped to the correct anatomical loci (right now Glu is the most proximal to the cell body, which is misleading as in reality it is the most distal at the tip of medial lobes). Also, why is the lateral horn left out of the scheme since it is the target of some MBONs and is thought to mediate innate olfactory medicated behavior?

5) Discussion of what is known about the inputs to CRE, SMP, SIP, and SLP neuropils will be helpful for readers to appreciate how the US leads to behavior through the action of the DA-MBON loops.

6) The movies are very useful to educate readers about the 3D organization of neuronal composition and their projections. However, the names stay too transient for readers to learn/remember. The authors could color code the names for each subclass of neurons and leave them on the screen after the neuronal classes are introduced, at least during segments when different classes of neurons are presented at the same time, to help readers to learn the relative positions of different classes of neurons and to remember one thing or two after viewing the movies.

Reviewer #2:

The study is a milestone in Drosophila neuroanatomy. Using state-of-art genetic tools, authors identified a comprehensive list of single neurons comprising the mushroom body and constructed a map of potential neural connections dividing MB lobes into 15 functional compartments. Moreover, they generated 85 split-GAL4 lines expressing exclusively in few MB related neurons that allow genetic manipulation of specific target neurons. Most major claims, except one which can be easily addressed (see below), are supported by thorough and convincing evidence. The study has great impact to the field and will transform the way of future study for understanding how genes and circuits control complex insect behaviors. I support the publication of the study and urge the authors make the generated tools immediately available to the entire field to further accelerate our understanding of brain functions.

It has been reported previously that "KC dendrites are segregated into 17 complementary domains according to their neuroblast clonal origins and birth orders (56). The authors stated that "The five KC types that receive olfactory information are each represented by hundreds of neurons per hemisphere and have their dendrites in the main calyx. Although their axons project to spatially segregated layers in the lobes, their dendritic arbors are intermingled in the calyx and KCs within a given cell type exhibit variable dendritic projection patterns (Figures 6 and 7). Moreover, the KCs receive input from an apparently random collection of glomeruli (67, 33, 18)". The statement is vague and implies a random spatial distribution of KC dendrites in the main calyx. In fact, the main calyx is clearly subdivided into 4 paralleled divisions, each derived from dendrites of descendent neurons of one of the four neuroblasts (Technau and Heisenberg, 1982; [41]; [53]; [115], 2006; [56]). Within each division, dendritic arbors of each KC type segregate at specific and separate spatial domain (56). Authors agree with that the pioneer alpha/beta (the posterior alpha/beta) KCs have dendrites exclusively in the dorsal accessory calyx and show that the newly identified embryonic gamma KCs have dendrites exclusively in the ventral accessory calyx. Consistently, in the main calyx, dendritic arbors of 5 other KC types also appear to distribute differently from each other (Figures 6 and 7). It remains unclear how the segregated KC dendrites account for the random PN-KC connections. Authors should either clarify their statement and cite the previous report indicating segregated dendrite distribution of each KC type in the main calyx or provide additional evidences of detailed anatomical analysis if otherwise.

---

## [Author Response]

Reviewer #1:

*Aso et al. reported stunning advances in the anatomical organization of the mushroom bodies, a key center for learning and memory in insects. Together with the accompanying behavioral analysis paper, these represent major breakthrough in understanding the structure and function of the mushroom bodies, their output, and their modulatory input. These studies have certainly laid down groundwork for years to decades of future investigations, and have already provided interesting insights on the logic of information processing principles through these intriguing structures. I am extremely enthusiastic in supporting the publication of these papers. Below I provide critiques to the anatomy paper mostly to ensure that the paper is accessible to as wide an audience as it can*.

*1) My strongest critique is the nomenclature, in particular the classes of MBONs. As far as I understand, most of these neurons are identified for the first time, at least to the resolution described, in this paper. So, the authors are in a position to name these neurons. I understand the rationales of the authors' choice using the connectivity with specific mushroom body lobes, and it may seem rational within this paper as it builds from Kenyon cell subtypes to MBONs. But even when reading the companion paper, only a slight distance away, it is already very difficult to remember these names (e.g., MBON-gamma4>gamma1gamma2, MBON-beta'2mp_bilateral, MBON-gamm1pedc>alpha/beta, etc.) or the rationale behind without constantly checking this paper. Consider many papers that will follow this paper in many years to come! Often complex names can turn off young scientists wishing to enter an otherwise very exciting field. In my view, these names should be greatly simplified. I have a specific suggestion. Since the authors found that MBONs neatly fall within three neurotransmitter phenotypes, and that MBONs that belong to a particular neurotransmitter category intersect with specific MB lobes at specific locations (both very interesting new findings!), why not name the MBONs after neurotransmitters (which is one of the most used convention for classifying neurons), and then within a neurotransmitter category, give a number, perhaps based on the proximal-distal axis of the mushroom body axons (e.g., MBON-ACh1, ACh2...; GABA1, GABA2...; Glu1, Glu2...). An alternative is to use consecutive numbers, for instance, MBON-ACh1-ACh8, GABA9-GABA12, GLU13-19, etc. With a modified*
Table 1*, readers can quickly check which lobe MBON-GABA3 receives input from, etc. The authors may also consider renaming the DANs but this may have more historical constraint and DANs don't neatly match MBONs. Still, some simplification also seems to be useful*.

We recognize that anatomical nomenclature always represents a compromise between “simple” names and names that convey greater meaning. We chose our naming system, after much thought and consultation with many individuals working in the field, for two main reasons: (1) We felt our anatomical data provided a very solid foundation, making it unlikely that the MBON names would have be revised due to future studies. In contrast, we were not able to assign neurotransmitter types to all MBONs and even for those that we could assign, the possibility of additional co-transmitters remains open. (2) We preferred names that provided more anatomical information, allowing the name itself to provide the reader with an immediate connection to the existing literature that often refers to particular anatomical domains of the MB lobes.

As a practical matter, changing names now would be problematic, as our lines are currently in use in about a dozen other laboratories (most of whom were collaborators in the work described in these two papers), that have been doing behavioral and physiological studies extending the current work. We are aware of five papers currently under review from those laboratories that use our current nomenclature.

However, we do appreciate the point raised by the reviewer and so we have included a short name as a column in Table 1. We also introduce this in the second paper where Figure 1 serves to diagram the morphology of these cells. For the reason mentioned above, we are not including transmitter information, but just a simple numbering scheme; MBON-01, MBON-02, etc. We also provide a short name for each DAN cell type of the format PAM-01 to PAM-14 and PPL1-01 to PPL1-06. We also show these short names in Figures 14, 15 and 16, where we also show previously used names from the literature; in this way the reader can see all alternative names for a cell type together with a visualization of the cell types morphology.

In these two papers (also for the reasons mentioned above) we otherwise continue to use the “full names”.

*2) Definition of cell type: I would add the following "We operationally define a cell type..." at the beginning, and "The screen was near saturation in our largeGAL4 collection*.*"*

We modified the text as suggested.

*3) I may have missed among the large amount of data, but can the authors describe MBON subtypes that are modulated by previously reported appetitive vs*. *aversive DANs?*

Direct experimental evidence for modulation (that is, electrophysiological measurement of MBON output before and after dopamine signaling by specific DANs) has not yet been reported. What we now have is anatomical correlations between the MB compartments containing the dendrites of specific MBON and the synaptic terminals of subsets of DANs that have been implicated in behavioral assays of aversive or appetitive memory. This large body of work is summarized diagrammatically in Figure 1 (and the text) of the accompanying paper we now refer the reader to that summary. While we recognize that the two papers need to stand on their own, we have tried to minimize redundancy between them in the review of published background information.

*4) A few critiques on*
Figure 21
*(the summary figure). It is probably better to draw a more anatomically correct mushroom body schematics (with a bifurcation), with Glu, ACh, and GABA mapped to the correct anatomical loci (right now Glu is the most proximal to the cell body, which is misleading as in reality it is the most distal at the tip of medial lobes). Also, why is the lateral horn left out of the scheme since it is the target of some MBONs and is thought to mediate innate olfactory medicated behavior*?

We modified Figure 21 as suggested. (For consistency we also modified Figure 15 of the accompanying paper.)

*5) Discussion of what is known about the inputs to CRE, SMP, SIP, and SLP neuropils will be helpful for readers to appreciate how the US leads to behavior through the action of the DA-MBON loops*.

In modified the Figure 21 legend to include this information.

*6) The movies are very useful to educate readers about the 3D organization of neuronal composition and their projections. However, the names stay too transient for readers to learn/remember. The authors could color code the names for each subclass of neurons and leave them on the screen after the neuronal classes are introduced, at least during segments when different classes of neurons are presented at the same time, to help readers to learn the relative positions of different classes of neurons and to remember one thing or two after viewing the movies*.

These movies are meant to give a quick overview on the morphology of each MB neurons. We believe that the main objective of the reviewer, more time to study each labeled image, can be achieved by pausing the playing of the movies to allow extended viewing time for any frame. We agree that these types of visualizations can be very useful for viewing the combinations of selected subsets of neurons. The movies were generated using the freely available 3D rendering software, Fluorender ((http://www.sci.utah.edu/software/13-software/127-fluorender.html.) We will make the data set and template we used for making movies available on the same website where we have made the original confocal imaging data available (http://www.janelia.org/split-gal4). Using that information, in combination with the Fluorender software, one should be able to visualize any selected combination of MB cell types and freely rotate them in 3D.

Reviewer #2:

*The study is a milestone in Drosophila neuroanatomy. Using state-of-art genetic tools, authors identified a comprehensive list of single neurons comprising the mushroom body and constructed a map of potential neural connections dividing MB lobes into 15 functional compartments. Moreover, they generated 85 split-GAL4 lines expressing exclusively in few MB related neurons that allow genetic manipulation of specific target neurons. Most major claims, except one which can be easily addressed (see below), are supported by thorough and convincing evidence. The study has great impact to the field and will transform the way of future study for understanding how genes and circuits control complex insect behaviors. I support the publication of the study and urge the authors make the generated tools immediately available to the entire field to further accelerate our understanding of brain functions*.

We can assure the reviewer that the split-GAL4 driver lines described in the paper will all be made freely available at the time of publication, and that a representation subset of the original confocal data used to characterize each line will be downloadable from http://www.janelia.org/split-gal4.

*It has been reported previously that "KC dendrites are segregated into 17 complementary domains according to their neuroblast clonal origins and birth orders (*[56]*). The authors stated that "The five KC types that receive olfactory information are each represented by hundreds of neurons per hemisphere and have their dendrites in the main calyx. Although their axons project to spatially segregated layers in the lobes, their dendritic arbors are intermingled in the calyx and KCs within a given cell type exhibit variable dendritic projection patterns (*Figures 6 and 7*). Moreover, the KCs receive input from an apparently random collection of glomeruli (*[67]*,*
[33]*,*
[18]*)". The statement is vague and implies a random spatial distribution of KC dendrites in the main calyx. In fact, the main calyx is clearly subdivided into 4 paralleled divisions, each derived from dendrites of descendent neurons of one of the four neuroblasts (Technau and Heisenberg, 1982;*
[41]*;*
[53]*;*
[115]*,*
*2006**;*
[56]*). Within each division, dendritic arbors of each KC type segregate at specific and separate spatial domain (*[56]*). Authors agree with that the pioneer alpha/beta (the posterior alpha/beta) KCs have dendrites exclusively in the dorsal accessory calyx and show that the newly identified embryonic gamma KCs have dendrites exclusively in the ventral accessory calyx. Consistently, in the main calyx, dendritic arbors of 5 other KC types also appear to distribute differently from each other (*Figures 6 and 7*). It remains unclear how the segregated KC dendrites account for the random PN-KC connections. Authors should either clarify their statement and cite the previous report indicating segregated dendrite distribution of each KC type in the main calyx or provide additional evidences of detailed anatomical analysis if otherwise*.

We added the following sentences and citations to more clearly explain the neuroblast origins of the KCs and the segregation of their dendrites in the calyx:

“Each MB contains ∼2,000 KCs, which are sequentially generated from four neuroblasts (41, 53, 115, 56).”

“The split-GAL4 screen and the analysis of the axonal projection patterns of single cells revealed that these three classes of KCs divide into seven cell types (Figure 3 and Video 2). Each of the four neuroblasts contributes to each of the seven cell types and the dendrites of the KCs generated from the different neuroblasts remain segregated in the main calyx (56).”

Also we modified the sentence to explain that dendrites of KCs from the same type show spatial bias in the calyx.

“Each KC cell type sends axonal projections to a spatially segregated layer in the lobes. The dendritic arbors of each KCs type also tend to be found in the same regions of the calyx (54, 56), but those dendritic zones are largely overlapping and individual.”